# Tables2Traces: Distilling Tabular Data to Improve LLM Reasoning in Healthcare

## Abstract

Large language models (LLMs) excel at reasoning when fine-tuned on curated text corpora, but many domains, such as medicine, primarily store knowledge in structured tabular data. Despite its richness, tabular data has been largely overlooked as a source of reasoning supervision. Interpreting such data requires structured, relational reasoning across features and outcomes, not just surface-level pattern matching. In practice, this mirrors clinical decision making, where doctors often compare patients with similar characteristics and reason about why their outcomes diverge. We introduce **Tables2Traces**, the first framework to enable improved domain-grounded reasoning from raw tabular data by generating contrastive, case-based reasoning traces for model fine-tuning. This establishes a new supervision paradigm: converting tabular records, traditionally used only for prediction, into structured reasoning signals that can serve as an effective new source of supervision for LLMs. Crucially, this paradigm is orthogonal to text-based QA supervision: rather than competing with curated corpora, it unlocks an abundant and low-cost modality that complements existing approaches. Using only cardiovascular patient records, Tables2Traces yields relative gains of 17.2% on in-domain MedQA questions and 8.4% out-of-domain, improving accuracy in 15 of 17 clinical categories. On MedMCQA, it achieves a 7.2% relative improvement and outperforms the base model in 16 of 21 specialties. These gains are driven by a lightweight and general pipeline that elicits structured reasoning via contrastive and counterfactual prompts. Compared to training on narrative patient descriptions, Tables2Traces generalizes more effectively across question types and medical specialties, showing that even limited tabular data can serve as a scalable and complementary source of reasoning supervision for LLMs.

## 1 Introduction

Large language models (LLMs) have achieved remarkable performance across reasoning tasks, from multi-step mathematics (Cobbe et al., 2021) to medical question answering (Singhal et al., 2023). These advances are typically attributed to large-scale pretraining followed by supervised fine-tuning on datasets already structured as text-based reasoning tasks (Ouyang et al., 2022; Wei et al., 2022).

In many domains, however, knowledge is stored in *structured, non-linguistic formats* such as electronic health records, lab results, insurance claims, finance spreadsheets, or scientific measurements. Such datasets encode complex relationships and decision logic but lack the textual form required for LLM fine-tuning (Yin et al., 2020; Liu et al., 2021). Closing this modality gap would unlock the reasoning signals latent in these data sources.

Consider a clinician assessing cardiovascular risk from patient records. A row of clinical features (e.g, age, LDL, diabetes, blood pressure) supports reasoning such as: "patients over 60 with high LDL and diabetes are at elevated risk, even if blood pressure is normal." Clinicians perform this reasoning intuitively, yet there is currently no systematic way to pass this knowledge to an LLM. Existing strategies rely on curated QA corpora (Puri et al., 2020), table-embedding models (e.g., TaBERT (Yin et al., 2020), TAPEX (Liu et al., 2021)), or lightweight adapters (Li & Liang, 2021; Hu et al., 2022), but none directly translate raw tabular data into reasoning supervision.

This raises two challenges. **(C1) Representation**: how to represent each row of features into a coherent format suitable for reasoning while preserving feature relationships.

**(C2) Trace elicitation:** how to automatically generate reasoning traces that capture the latent knowledge contained in the structured data.

To address C1-C2, we introduce **Tables2Traces**, the first end-to-end framework to transform the latent knowledge embedded in tabular data to reasoning traces that can then be used to fine-tune and improve an LLM. Our goal is to strengthen domain-grounded clinical reasoning rather than general-purpose reasoning; the cross-specialty gains we observe reflect transfer of this domain-grounded signal rather than an attempt to improve reasoning broadly. More specifically, Tables2Traces addresses the following key research question:

> Can the latent knowledge embedded in structured tabular data be reformulated into reasoning tasks that LLMs can learn from-and does fine-tuning on such synthetic examples improve reasoning in both in-domain and out-of-domain settings?

Addressing this question offers dual benefits. **For domain experts**, it offers a path to adapt LLMs using structured datasets to which they already have access. This is especially valuable in fields like medicine, where data privacy and regulatory constraints often prevent data sharing. Practitioners can extract reasoning supervision directly from their own tabular data, effectively imbuing LLMs with local contextual knowledge.

**For the LLM research community**, this approach provides a new and complementary modality that contains rich domain knowledge for supervision. Although most fine-tuning datasets are human-annotated or LLM-synthesized from existing text corpora (Gururajan et al., 2024), we demonstrate that abundant tabular data can also provide useful reasoning supervision. This reframes tabular data as a rich and domain-specific supervision source and contributes to the growing data-centric shift in LLM development. Crucially, to the best of our knowledge, we are the first to demonstrate that latent knowledge encoded in tabular datasets can be reformulated into reasoning tasks that improve LLMs.

In doing so, we make the following contributions:

**Contributions.** ① **Conceptually:** We introduce a novel LLM supervision paradigm of transforming tabular datasets into structured reasoning traces. This introduces a new pathway for domain adaptation using data that has traditionally been excluded from LLM training pipelines. Our contribution is data-centric: we show that structured tabular records can be transformed into reasoning supervision that improves medical QA performance, without introducing new model architectures or optimization methods. ② **Methodologically:** We present **Tables2Traces**, a lightweight and modular pipeline that generates contrastive questions and multi-step reasoning traces from labeled tabular data, using only a small QA subset for output-format alignment. ③ **Empirically:** We evaluate on `MedQA` and `MedMCQA`, using 105k synthetic traces and 10k QA examples. Tables2Traces yields strong in-domain gains (+17.2%) and generalizes out-of-domain (+8.4% on `MedQA`, +7.2% on `MedMCQA`), showcasing the effectiveness of tabular supervision. ④ **Analytically:** We find that Tables2Traces closes part of the gap to a state-of-the-art medical QA model, *Aloe*, while using only around 1.3% as much medical QA data. Aloe serves as a natural reference point as it exemplifies the standard paradigm for medical QA, where performance is driven by large-scale QA-format supervision across many curated medical datasets. We show how contrastive supervision improves generalization across question types and embedding space regions.

**Disclaimer.** The models produced in this work are intended solely for research use; they are not designed or validated for clinical decision-making and must not be used for diagnostic or treatment purposes.

## 2 RELATED WORK

This work engages with works on LLM fine-tuning and LLMs for tabular data.

**LLM Fine-Tuning.** Prior work demonstrates that fine-tuning LLMs on structured reasoning datasets (e.g. GSM8K) can significantly improve problem-solving capabilities, which can then be enhanced via chain-of-thought prompting (Wei et al., 2022). Instruction-tuning (e.g. Self-Instruct (Wang et al., 2023b)) further show that training with human-style prompts and reasoning traces enhances generalization across unseen tasks. More recently, DeepSeek-R1 (Guo et al., 2025) introduced a large-scale framework for fine-tuning LLMs on curated reasoning traces using the Generalized Reinforcement Preference Optimization (GRPO). By combining diverse, high-quality reasoning

traces with fine-tuning, DeepSeek-R1 demonstrated strong improvements. Our work builds on this paradigm by generating structured reasoning traces not from existing text corpora, but from raw tabular datasets, enabling fine-tuning in domain-specific settings.

In parallel, alignment methods such as reinforcement learning from human feedback (RLHF) (Ouyang et al., 2022), direct preference optimization (DPO) (Rafailov et al., 2023), and reward modeling (Christiano et al., 2017) have shown that models benefit from being tuned on outputs aligned with human preferences. However, all of these methods assume access to a large corpus of natural language examples. In contrast, we synthesize supervision from structured data, which is abundant but underutilized in current LLM pipelines. Our work is orthogonal to these and serves to highlight the potential of structured tabular data as a new source of supervision.

**LLMs for tabular data.** Research on modeling structured tables with language models has largely focused on two paradigms: semantic table understanding and supervised prediction. For the former, models like TaBERT (Yin et al., 2020), TAPEX (Liu et al., 2021), and TURL (Deng et al., 2022) learn joint text–table representations for question answering and schema reasoning. For the latter task of prediction, architectures such as TabNet (Arik & Pfister, 2021) and FT-Transformer (Gorishniy et al., 2021) are optimized for predictive modeling over tabular features.

More recently, LLM-based approaches such as TabLLM (Hegselmann et al., 2023), UniPredict (Wang et al., 2023a), LLaMA-GTL (Yan et al., 2024) and TP-BERTa (Zhang et al., 2023) explore serialized tabular inputs or tabular pretraining to improve transferability in tabular prediction tasks.

However, these methods tackle the fundamentally different problem of LLMs understanding tables or LLMs being used as tabular predictors. In contrast, we focus on answering the question of how to use tabular structured data (and the knowledge contained therein) to enhance the reasoning capabilities of LLMs within the relevant problem domain. i.e. how can we use tabular data on cardiovascular patients to improve an LLMs capabilities to reason about cardiovascular problems or even more general medical questions.

## 3 METHOD

### 3.1 PROBLEM FORMULATION

We assume a tabular dataset $\mathcal{D} = \{x_1, \ldots, x_N\}$ where each row $x_i \in \mathbb{R}^d$ is a structured record (e.g., a clinical case) and is associated with a binary label $y_i \in \{0, 1\}$. While such tabular datasets encode rich domain knowledge, they do not naturally align with the data formats LLMs are typically trained on. We hence seek a mapping that converts this structure into contrastive prompts and structured reasoning traces suitable for supervised fine-tuning of LLMs:

$$\Pi : \mathcal{D} = \{(x_i, y_i)\}_{i=1}^N \longrightarrow \mathcal{C} = \{(P_i, R_i)\}_{i=1}^M \tag{1}$$

where each prompt $P_i$ describes a clinical scenario in natural language and each trace $R_i$ is a structured reasoning trace generated by an LLM.

Ultimately, the goal is that fine-tuning a target LLM with parameters $\Theta$ on $\mathcal{C}$ (derived from tabular data), can teach the model to learn high-level reasoning behaviors from the structure of the data, without requiring domain-specific logic or annotation. In doing so, this can improve LLM performance on related text-based tasks such as Q&A. To anchor ideas, we hypothesize that eliciting reasoning traces from tabular medical data and then fine-tuning an LLM on these traces should improve an LLM's capabilities on medical Q&A tasks.

In this work, we primarily focus on the role of medical tabular data, hence our focus is on eliciting clinical reasoning capabilities from a structured medical dataset.

### 3.2 TABLES2TRACES

We propose Tables2Traces as a framework that realizes this mapping function $\Pi$. The underlying algorithm is outlined in Algorithm 1. In particular, it allows us to provide solutions to overcome the challenges of (C1) Representation and (C2) Trace elicitation.

**(C1) Representation.** Our first challenge is how to represent the tabular data in a suitable format, prior to eliciting reasoning traces. A deterministic encoder $\phi$ translates each tabular row $x_i$ into a compact textual patient description. Column headers are normalized into human-interpretable phrases (e.g. ldl_chol→low–density lipoprotein cholesterol), numerical values are rendered with units, and missing entries are declared explicitly. Ultimately, $\phi$ (operationalized with an LLM), transforms tabular rows into fluent text-based summaries $\phi(x_i) = s_i$. The result is a corpus of textual case descriptions $\mathcal{C}_{\text{simple}} = \{s_i\}_{i=1}^N$, which we use as training data for the Tables2Traces (simple) variant.

**(C2) Trace Elicitation.** Once we have the data in a suitable, represented textual format, we wish to elicit the appropriate knowledge and reasoning from the data. We do so as follows:

▶ *Contrastive Neighbor Selection.* Clinical reasoning frequently involves comparative analysis, i.e., why did this patient die, whereas a similar patient survived? To elicit a similar contrastive reasoning for each anchor example $x_i$, we retrieve its nearest survivor $x_i^{(0)}$ and nearest deceased $x_i^{(1)}$ based on the Gower distance [1], as it respects heterogeneous feature types. We use this to form a narrative triplet $\tau_i = \left( \phi(x_i^{(0)}), s_i, \phi(x_i^{(1)}) \right)$, which corresponds to a contrastive decision used as supervision for the LLM.

▶ *Reasoning extraction via prompt design.* We wrap each summary $\tau_i$ in a prompt template $P_i$ (Appendix C.3) designed to elicit deeper reasoning about structured differences in the tabular data rather than surface-level pattern matching. The design is inspired by ideas from contrastive learning, consistency checking, and counterfactual reasoning, adapted into natural-language form so that they can serve as supervision for an LLM:

1. **Differential reasoning.** Compare the TARGET to each neighbour and list decisive feature differences. This step encourages differential reasoning about relative risk patterns and structured comparisons between similar cases.

2. **Label Plausibility.** State whether the recorded outcome is clinically plausible. This step promotes a general form of critical evaluation and teaches the model to reason about internal consistency rather than simply accept labels at face value.

3. **Counterfactual Planning.** We prompt the model to suggest one *minimal* feature edit that would reverse the outcome and justify why. The model must isolate an actionable feature (e.g. reduce LDL to $< 100$ mg/dL) and explain its physiological effect. This mirrors clinical reasoning in which a clinician asks, "What intervention would have saved this patient?". This encourages a higher-level causal reasoning heuristic that goes beyond pattern matching and requires identifying features with directional influence.

▶ *Trace Extraction and Corpus Assembly.* We pass the constructed prompt $P_i$ to the frozen LLM $\mathcal{L}$ (e.g. Deepseek-R1) and extract the generated reasoning trace $R_i$. Collecting every pair yields the corpus $\mathcal{C}$, wherein we have converted the tabular data into structured reasoning traces.

**Supervised Fine-tuning.** We then fine-tune a downstream target LLM using standard supervised learning on the dataset $\{(P_i, R_i)\}$. Each training example consists of: **Input:** the contrastive prompt $P_i$ (including the three case descriptions) and **Output:** the generated reasoning trace $R_i$.

---

**Algorithm 1** TABLES2TRACES: From Tabular Data to Reasoning Corpus

---

**Require:** Tabular dataset $D$, frozen LLM $\mathcal{L}$
1: **Output:** Reasoning corpus $\mathcal{C}$
2: **for each** $(x_i, y_i) \in D$ **do**
3:    $s_i \leftarrow \phi(x_i)$                    ▷ Representation
4:    **for** $y \in \{0, 1\}$ **do**
5:       $x_i^{(y)} \leftarrow \arg\min_{x:y_x=y} \text{Gower}(x, x_i)$
6:    **end for**
7:    $\tau_i \leftarrow \left( \phi(x_i^{(0)}), s_i, \phi(x_i^{(1)}) \right)$   ▷ Contrastive triple
8:    $P_i \leftarrow \pi(\tau_i)$                  ▷ Compose prompt
9:    $R_i \leftarrow \text{POSTPROCESS}\left(\mathcal{L}(P_i)\right)$ ▷ Trace elicitation
10:    $\mathcal{C} \leftarrow \mathcal{C} \cup \{(P_i, R_i)\}$
11: **end for**
12: **return** $\mathcal{C}$

---

[1]The neighbor-selection step is agnostic to the specific choice of metric, and alternative or learned distance functions can be substituted without modifying the overall framework - see Appendix M.

We fine-tune the model on a dataset $\mathcal{R}' = \{(P_i, R_i)\}_{i=1}^{M}$ consisting of $M$ prompt–response pairs, where each $P_i$ is a contrastive input prompt and $R_i$ is the corresponding reasoning trace. 90% of these samples are synthetic traces generated from tabular data and 10% are multiple-choice QA-format examples (e.g., MedQA). The QA examples were not contained in the evaluation datasets. Training on the QA subset alone performs on par with the base model (Appendix L).

We hold out 5% of the overall dataset for evaluation. Prompt templates and representative traces are shown in Appendices C and D.

Let $\mathcal{L}_{\mathrm{LM}}$ denote the language modeling loss. The fine-tuning objective is then:

$$\min_{\theta} \sum_{(P,R) \in \mathcal{R}'} \mathcal{L}_{\mathrm{LM}}(R \mid P; \theta),$$

where $\theta$ are the parameters of the language model. As Tables2Traces uses standard supervised fine-tuning with a fixed architecture, the computational cost scales linearly with the number of reasoning traces, and trace generation is a one-time preprocessing step that parallelizes trivially across rows.

**Extensibility.** Although we apply the method in a clinical setting with a binary outcome, the framework is not restricted to this domain. Tables2Traces assumes only that each row represents a coherent and interpretable entity and that a meaningful way to select similar and dissimilar rows exists. Clinical datasets naturally satisfy these conditions because each row corresponds to a patient, but other domains, such as finance or education, may require different design choices about what constitutes an entity and how similarity should be defined. The framework also supports alternative label structures in principle, including multi-class or continuous outcomes, by adjusting how neighbors are selected. Exploring these extensions is an interesting direction for future work, and our current implementation should be viewed as a proof of concept in a domain with well-defined row semantics.

## 4 EXPERIMENTS

We evaluate Tables2Traces as a mechanism for transforming structured tabular data into effective domain-grounded reasoning supervision for LLMs. Our goal is to assess whether this supervision improves medical QA performance, and to analyze where performance differences across question types, medical domains, and benchmarks.

**Data.** We use a subset of the UK Biobank (Sudlow et al., 2015) comprising 105,299 individuals aged 40 and above, all diagnosed with cardiovascular disease. Each patient is represented by 32 variables spanning demographics, medication usage, lab results, and comorbidities. All data were collected under appropriate ethical approvals and informed consent (Palmer, 2007).

**Setup.** We evaluate two tabular supervision variants: (1) **Tables2Traces (simple)**, which converts each row into a standalone patient narrative; and (2) **Tables2Traces**, which adds contrastive prompts using nearest-neighbor pairs. Each variant uses 90% synthetic reasoning traces and 10% of synthetically generated QA-format examples from `HPAI-BSC/MedQA-Mixtral-CoT` (Gururajan et al., 2024), included only for aligning the model to the multiple-choice answer format. Both Tables2Traces variants rely on this same small subset, and ablations confirm that it does not contribute meaningfully to performance (Appendix L).

We fine-tune both 8 billion parameter (8B) (`DeepSeek-R1-Distill-LLaMA-8B`) and 7 billion (7B) parameter (`DeepSeek-R1-Distill-Qwen-7B`) models[2] using the Open-R1 framework (HuggingFace, 2025). Training configurations are detailed in Appendix B and results for 7B models are provided separately in Appendix H.

**Evaluation.** We evaluate on two medical QA benchmarks: `MedQA` (Jin et al., 2021) and `MedMCQA` (Pal et al., 2022). Accuracy is reported under four aggregation schemes (average, best-of-$n$, majority vote, worst-of-$n$). Results are averaged over 10 stochastic runs; error bars show the

---

[2]We consider these smaller model sizes as these are realistic LLM sizes for fine-tuning in clinical settings given compute limitations.

standard error of the mean. We compare both fine-tuned variants against a **Base** model without fine-tuning.

We also report **Aloe** (Gururajan et al., 2024), a strong medical QA system trained with resource-intensive, heavily curated supervision across more than twenty datasets (e.g., synthetic chain-of-thought, guideline-based answers, adversarial/preference tuning). Importantly, Aloe is *not a reasoning model*, so it is not aligned with our supervision signal. Our approach is orthogonal: Tables2Traces derives reasoning supervision automatically from structured tabular data without manual labels. For completeness, we fine-tuned Aloe with Tables2Traces; performance did not improve (Appendix G), consistent with this misalignment.

All 8B models use the same `LLaMA-8B` backbone and identical inference settings, and we apply identical chain-of-thought prompts at test time (Appendix C.7-C.8). Differences therefore reflect supervision rather than prompting. As a qualitative sanity check of the supervision signal, two cardiologists independently reviewed 10 randomly sampled traces and confirmed that none received a *Concerning* safety rating (Appendix O; protocol in Appendix N).

We assess performance across the following dimensions:

1. **Performance: Are gains consistent across clinical subdomains and benchmarks?**
   Section 4.1 evaluates performance across 18 clinical categories in MedQA and 21 in MedMCQA to identify where training is most effective and whether performance generalizes.
2. **Domain generalization: What types of questions benefit from tabular supervision?**
   Section 4.2 examines both *domain transfer* (e.g., cardiovascular $\rightarrow$ neurology) and *format transfer* (e.g., patient-specific $\rightarrow$ abstract) to determine what kinds of questions benefit most.
3. **Locating successes and failures: Where do models succeed or fail in embedding space?**
   Section 4.3 uses UMAP visualizations to localize model performance across semantic regions of the question embedding space.
4. **Upper bound comparison: How does performance compare to a QA-optimized model?**
   Sec. 4.4 compares our method to *Aloe*, a model trained on large-scale QA data, including the training set from both MedQA and MedMCQA. This contextualizes how far Tables2Traces can go with primarily tabular supervision compared to a task-optimized upper bound.

### 4.1 ARE GAINS CONSISTENT ACROSS CLINICAL SUBDOMAINS AND BENCHMARKS?

**Goal.** Assess whether improvements from tabular supervision generalize across diverse clinical categories, and whether these gains hold across both `MedQA` and `MedMCQA` benchmarks.

**Setup.** For `MedQA`, we assign each question to one of 18 clinical categories using `DeepSeek-R1`. For `MedMCQA`, we use the dataset assigned category labels and evaluate on the public validation set (as the test set is not accessible). We restrict evaluation to questions with a single correct answer to ensure consistency with the `MedQA` setup and to allow for accurate, per-question performance analysis.

**Results.** As shown in Table 1 and Appendix E, Tables2Traces consistently outperforms the base model across a majority of clinical categories in both datasets. On `MedQA`, we observe improvements in 16 out of 18 categories (89%), with the largest relative gains in *Renal/Genitourinary* (+29.65%), *Hematologic* (+18.98%), and *Cardiovascular* (+17.21%). On `MedMCQA`, Tables2Traces improves performance in 16 out of 21 categories (76%), including strong gains in *Psychiatry* (+31.71%), *ENT* (+16.56%), and *Anatomy* (+11.91%). Categories showing drops in performance e.g., *Skin* (–24.39%) and *Orthopaedics* (–15.00%) have very few test samples ($N = 11$ and $N = 15$), making these estimates statistically unreliable.

**Takeaway.** Tables2Traces yields robust, cross-domain improvements on both benchmarks, improving in over 80% of clinical categories. The few observed declines are isolated to small and noisy subsets.

### 4.2 WHAT TYPES OF QUESTIONS BENEFIT FROM TABULAR SUPERVISION?

**Goal.** To assess whether supervision derived from a single clinical domain (cardiovascular) and a single input style (patient-specific reasoning traces) can improve performance on both out-of-domain specialties and abstract medical knowledge.

Table 1: Per-category evaluation metrics on the MedQA benchmark for Base and Tables2Traces.

| Category | Model Type | Avg Accuracy | Best-of-n | Majority Vote | Worst-of-n | % Change |
|---|---|---|---|---|---|---|
| **Cardiovascular** | Base | $0.40 \pm 0.03$ | $0.86 \pm 0.03$ | $0.31 \pm 0.04$ | $\mathbf{0.06 \pm 0.02}$ | |
| *(N = 130)* | Tables2Traces | $\mathbf{0.47 \pm 0.03}$ | $\mathbf{0.91 \pm 0.03}$ | $\mathbf{0.42 \pm 0.04}$ | $\mathbf{0.06 \pm 0.02}$ | +17.21% ↑ |
| **Dermatologic** | Base | $0.59 \pm 0.08$ | $\mathbf{0.94 \pm 0.06}$ | $0.53 \pm 0.12$ | $0.06 \pm 0.06$ | |
| *(N = 17)* | Tables2Traces | $\mathbf{0.60 \pm 0.08}$ | $0.88 \pm 0.08$ | $\mathbf{0.59 \pm 0.12}$ | $\mathbf{0.12 \pm 0.08}$ | +0.99% ↑ |
| **Endocrine/Metabolic** | Base | $0.49 \pm 0.03$ | $0.89 \pm 0.02$ | $0.45 \pm 0.04$ | $\mathbf{0.13 \pm 0.03}$ | |
| *(N = 179)* | Tables2Traces | $\mathbf{0.51 \pm 0.02}$ | $\mathbf{0.91 \pm 0.02}$ | $\mathbf{0.46 \pm 0.04}$ | $0.10 \pm 0.02$ | +4.71% ↑ |
| **Gastrointestinal** | Base | $0.47 \pm 0.04$ | $0.87 \pm 0.04$ | $0.40 \pm 0.05$ | $\mathbf{0.12 \pm 0.04}$ | |
| *(N = 86)* | Tables2Traces | $\mathbf{0.50 \pm 0.04}$ | $\mathbf{0.91 \pm 0.03}$ | $\mathbf{0.47 \pm 0.05}$ | $0.08 \pm 0.03$ | +6.72% ↑ |
| **Hematologic** | Base | $0.40 \pm 0.04$ | $0.84 \pm 0.04$ | $0.34 \pm 0.06$ | $0.04 \pm 0.03$ | |
| *(N = 68)* | Tables2Traces | $\mathbf{0.48 \pm 0.04}$ | $\mathbf{0.91 \pm 0.04}$ | $\mathbf{0.43 \pm 0.06}$ | $\mathbf{0.07 \pm 0.03}$ | +18.98% ↑ |
| **Immunologic** | Base | $0.51 \pm 0.04$ | $0.85 \pm 0.04$ | $\mathbf{0.47 \pm 0.06}$ | $\mathbf{0.22 \pm 0.05}$ | |
| *(N = 81)* | Tables2Traces | $\mathbf{0.54 \pm 0.04}$ | $\mathbf{0.94 \pm 0.03}$ | $0.46 \pm 0.06$ | $0.17 \pm 0.04$ | +6.80% ↑ |
| **Infectious** | Base | $0.48 \pm 0.03$ | $0.92 \pm 0.02$ | $0.41 \pm 0.04$ | $\mathbf{0.11 \pm 0.02}$ | |
| *(N = 176)* | Tables2Traces | $\mathbf{0.53 \pm 0.02}$ | $\mathbf{0.94 \pm 0.02}$ | $\mathbf{0.45 \pm 0.04}$ | $\mathbf{0.11 \pm 0.02}$ | +9.73% ↑ |
| **Musculoskeletal** | Base | $0.49 \pm 0.05$ | $0.89 \pm 0.05$ | $\mathbf{0.49 \pm 0.07}$ | $0.04 \pm 0.03$ | |
| *(N = 45)* | Tables2Traces | $\mathbf{0.51 \pm 0.04}$ | $\mathbf{0.96 \pm 0.03}$ | $0.40 \pm 0.07$ | $\mathbf{0.07 \pm 0.04}$ | +4.07% ↑ |
| **Neurological** | Base | $0.47 \pm 0.04$ | $0.86 \pm 0.04$ | $0.42 \pm 0.06$ | $0.09 \pm 0.03$ | |
| *(N = 77)* | Tables2Traces | $\mathbf{0.54 \pm 0.04}$ | $\mathbf{0.95 \pm 0.03}$ | $\mathbf{0.48 \pm 0.06}$ | $\mathbf{0.14 \pm 0.04}$ | +15.15% ↑ |
| **Obstetrics/Gynecology** | Base | $0.46 \pm 0.04$ | $0.90 \pm 0.04$ | $0.39 \pm 0.06$ | $\mathbf{0.09 \pm 0.03}$ | |
| *(N = 70)* | Tables2Traces | $\mathbf{0.47 \pm 0.03}$ | $\mathbf{0.94 \pm 0.03}$ | $\mathbf{0.40 \pm 0.06}$ | $0.03 \pm 0.02$ | +2.80% ↑ |
| **Oncology** | Base | $0.53 \pm 0.04$ | $0.92 \pm 0.03$ | $0.47 \pm 0.06$ | $0.11 \pm 0.04$ | |
| *(N = 72)* | Tables2Traces | $\mathbf{0.56 \pm 0.04}$ | $\mathbf{0.93 \pm 0.03}$ | $\mathbf{0.53 \pm 0.06}$ | $\mathbf{0.14 \pm 0.04}$ | +5.82% ↑ |
| **Other** | Base | $\mathbf{0.53 \pm 0.07}$ | $0.77 \pm 0.08$ | $\mathbf{0.45 \pm 0.09}$ | $\mathbf{0.23 \pm 0.08}$ | |
| *(N = 31)* | Tables2Traces | $0.50 \pm 0.07$ | $\mathbf{0.87 \pm 0.06}$ | $0.42 \pm 0.09$ | $0.19 \pm 0.07$ | -4.88% ↓ |
| **Pediatric** | Base | $\mathbf{0.39 \pm 0.09}$ | $0.77 \pm 0.12$ | $\mathbf{0.39 \pm 0.14}$ | $\mathbf{0.00 \pm 0.00}$ | |
| *(N = 13)* | Tables2Traces | $\mathbf{0.39 \pm 0.05}$ | $\mathbf{1.00 \pm 0.00}$ | $0.31 \pm 0.13$ | $\mathbf{0.00 \pm 0.00}$ | -1.96% ↓ |
| **Psychiatric** | Base | $0.59 \pm 0.05$ | $\mathbf{0.94 \pm 0.03}$ | $0.54 \pm 0.07$ | $\mathbf{0.23 \pm 0.06}$ | |
| *(N = 52)* | Tables2Traces | $\mathbf{0.62 \pm 0.05}$ | $0.90 \pm 0.04$ | $\mathbf{0.61 \pm 0.07}$ | $0.21 \pm 0.06$ | +5.57% ↑ |
| **Renal/Genitourinary** | Base | $0.37 \pm 0.04$ | $0.85 \pm 0.05$ | $0.26 \pm 0.06$ | $0.04 \pm 0.03$ | |
| *(N = 54)* | Tables2Traces | $\mathbf{0.48 \pm 0.04}$ | $\mathbf{0.96 \pm 0.03}$ | $\mathbf{0.41 \pm 0.07}$ | $\mathbf{0.09 \pm 0.04}$ | +29.65% ↑ |
| **Respiratory** | Base | $0.49 \pm 0.04$ | $0.91 \pm 0.04$ | $0.43 \pm 0.07$ | $0.09 \pm 0.04$ | |
| *(N = 54)* | Tables2Traces | $\mathbf{0.50 \pm 0.04}$ | $\mathbf{0.94 \pm 0.03}$ | $\mathbf{0.46 \pm 0.07}$ | $\mathbf{0.11 \pm 0.04}$ | +2.28% ↑ |
| **Toxicology** | Base | $0.43 \pm 0.04$ | $0.79 \pm 0.05$ | $0.41 \pm 0.06$ | $0.06 \pm 0.03$ | |
| *(N = 68)* | Tables2Traces | $\mathbf{0.52 \pm 0.04}$ | $\mathbf{0.91 \pm 0.04}$ | $\mathbf{0.47 \pm 0.06}$ | $\mathbf{0.09 \pm 0.04}$ | +20.68% ↑ |
| **Overall** | Base | $0.47 \pm 0.01$ | $0.88 \pm 0.01$ | $0.41 \pm 0.01$ | $0.11 \pm 0.01$ | |
| *(N = 1273)* | Tables2Traces | $\mathbf{0.51 \pm 0.01}$ | $\mathbf{0.93 \pm 0.01}$ | $\mathbf{0.46 \pm 0.01}$ | $0.10 \pm 0.01$ | **+9.19% ↑** |

**Setup.** We assess generalization along two axes: (1) **Domain**—partitioning questions into *cardiovascular* vs. *non-cardiovascular* using model-inferred labels; and (2) **Format**—classifying questions as *patient-specific* or *abstract*, based on whether they describe concrete cases (e.g., `45-year-old man`) or general concepts. `MedQA` is predominantly patient-specific (92.3%), while `MedMCQA` is mostly abstract (83.7%) (see Appendix F).

**Results.** In `MedQA`, Tables2Traces shows strong gains on in-domain (cardiovascular, +17.2%) and patient-specific questions (+10.0%), with smaller gains on abstract questions (+1.8%) (Figure 1). However, Tables2Traces (simple) performs worse on abstract questions (–15.3%), possibly suggesting overfitting to training format. In `MedMCQA`, we observe broader generalization: Tables2Traces improves both patient-specific (+6.8%) and abstract (+7.6%) subsets.

**Takeaway.** Tabular supervision supports generalization beyond its source domain and format. Without counterfactual reasoning, tabular supervision can overfit to its patient-specific training data, resulting in performance degradation in out-of-domain tasks. With counterfactual reasoning, Tables2Traces improves performance even on abstract, unfamiliar questions. These gains on abstract questions, where no patient structure is present, indicate that the model is not relying on patient-specific memorization but has learns higher-level, domain-specific reasoning heuristics introduced by the structured supervision.

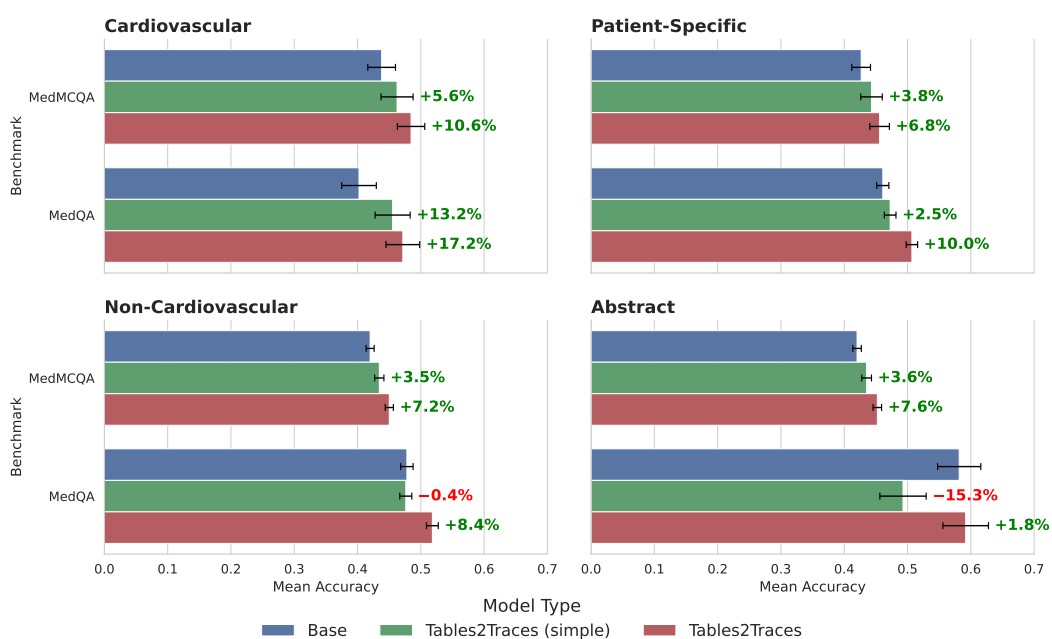

Figure 1: Accuracy comparison across model variants. Subplots show mean accuracy on `MedQA` and `MedMCQA`, with error bars for standard error. Percentages indicate improvement over the Base model. Top row: in-domain (cardiovascular, patient-specific); bottom row: out-of-domain (non-cardiovascular, abstract).

### 4.3 WHERE DO MODELS SUCCEED OR FAIL IN EMBEDDING SPACE?

**Goal.** Visualize how supervision strategies impact the semantic generalization of medical questions.

**Setup.** We embed all `MedQA` and `MedMCQA` questions using `text-embedding-3-large` (OpenAI, 2023) and reduce dimensionality via UMAP. Each point corresponds to a question, colored by clinical category. Background shading shows smoothed relative accuracy of the fine-tuned model compared to the **Base** model. Figure 2 shows two panels: (a) **Tables2Traces**, and (b) **Tables2Traces (simple)**. A complementary plot for `MedMCQA` is provided in Appendix E.

**Results.** Both models show localized gains within the cardiovascular region. However, only Tables2Traces generalizes effectively across distant clusters whereas the patient-style model (b) overfits to regions that closely resemble its training format. Peripheral zones, often containing abstract or non-patient-centered questions (e.g., *Biochemistry*, *Social Medicine*), show degradation under the simple model but improved performance under Tables2Traces. These patterns mirror our quantitative results and extend to the `MedMCQA` visualization.

**Takeaway.** Contrastive reasoning traces lead to broader semantic generalization, increasing performance across diverse question types and topics. In contrast, models trained only on patient description data tend to overfit and struggle in abstract or semantically distant regions of the question space.

### 4.4 HOW DOES PERFORMANCE COMPARE TO A QA-OPTIMIZED MODEL?

**Goal.** To benchmark Tables2Traces against a strong QA-tuned model, we compare it to **Aloe**—a state-of-the-art medical LLM trained on over 750,000 medical QA pairs from datasets including `MedQA` and `MedMCQA`, and further optimized through preference and adversarial feedback tuning (Gururajan et al., 2024). Rather than a direct competitor, Aloe represents a *task-optimized upper bound* built from large-scale QA supervision. Our comparison asks how far Tables2Traces can go using primarily tabular reasoning traces and only a small amount of QA data.

**Setup.** We evaluate **Base**, **Tables2Traces**, and **Aloe** across four subsets: *cardiovascular*, *non-cardiovascular*, *patient-specific*, and *abstract*, using both `MedQA` and `MedMCQA` (Figure 3). Aloe

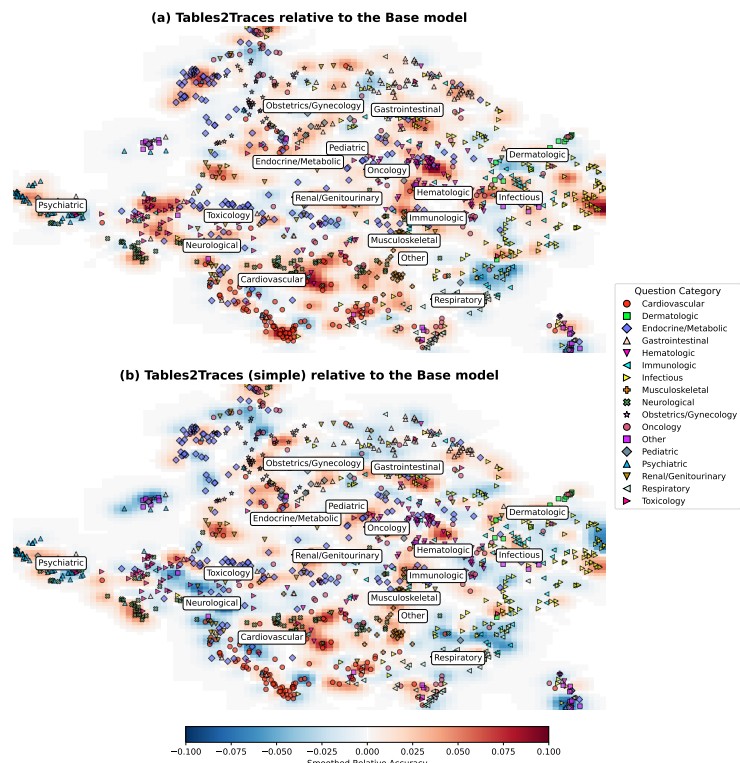

Figure 2: UMAP visualization of MedQA test questions comparing model performance to the **Base** model. Each point is a question, embedded using `text-embedding-3-large` (OpenAI, 2023), and annotated by medical category using distinct marker shapes and colors. Accuracy is smoothed using a Gaussian filter ($\sigma = 1.5$). The background heatmap shows relative performance: red indicates improvement, blue indicates degradation. Cluster labels mark category centroids. **(a)** Tables2Traces achieves broad gains across much of the question space. **(b)** Tables2Traces (simple) yields localized improvements, but also shows notable drops in performance in several regions.

is trained on nearly one million QA-format medical examples (human and synthetic), whereas Tables2Traces uses only about 10k QA-format examples for alignment and relies primarily on 105k reasoning traces generated from tabular data.

**Results.** As expected, Aloe achieves the highest accuracy in all categories. However, Tables2Traces closes a substantial portion of the gap despite using only 1.3% of the QA supervision data used by Aloe. The remaining supervision comes from synthetic reasoning traces, which incur no annotation cost. On `MedQA`, Tables2Traces improves +17.2% on cardiovascular questions (vs. Aloe's +26.0%) and +9.6% on patient-specific questions (vs. +25.3%). In `MedMCQA`, Tables2Traces achieves +10.6% and +6.8% improvements on cardiovascular and patient-specific questions respectively. Notably, even Aloe shows limited gains on abstract, non-patient-specific questions across both benchmarks, suggesting these are structurally more challenging and underrepresented during training.

**Takeaway.** **Tables2Traces achieves strong generalization with minimal QA supervision.** Despite being trained using *75× fewer medical QA samples* compared to Aloe, our approach closes a substantial portion of the performance gap. This highlights the value of clinical reasoning traces from tabular data as a scalable, interpretable, and cost-effective alternative to large-scale QA corpora.

## 5 DISCUSSION

Tables2Traces provides a scalable approach for converting structured tabular data into contrastive reasoning traces and improves LLM performance on medical QA tasks even when trained on a single clinical domain. Trace generation is a one-time preprocessing step that can be reused across

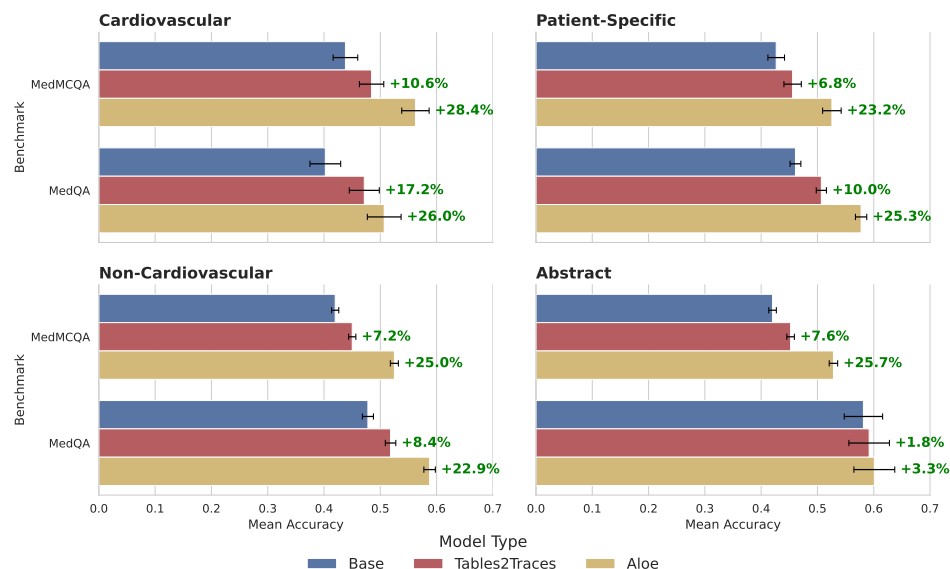

Figure 3: Accuracy comparison between **Base**, **Tables2Traces**, and **Aloe** across question categories, grouped by benchmark (MedMCQA, MedQA). Each subplot reports mean accuracy with standard error bars. Percentage improvements are relative to the Base model. Top row: in-domain categories. Bottom row: out-of-domain generalization.

models, and in practice fine-tuning dominates the total compute. By leveraging structured clinical data, the method introduces a new and broadly applicable source of supervision that encourages domain-specific higher-level reasoning behaviors grounded in patient differences. Importantly, our objective is to study domain-grounded reasoning supervision rather than general-purpose reasoning.

Although all training tabular data comes from a cardiovascular cohort, the model improves performance across many unrelated medical specialties. This suggests that the reasoning signal transfers beyond the training domain. The contrast between the simple and full variants supports this interpretation. Both variants use the same underlying tabular records and the same QA alignment subset, yet the simple version degrades on multiple out-of-domain subsets while the full version improves. This indicates that the gains arise from the structured reasoning components rather than from specialty-specific correlations in the tabular data or the QA alignment subset.

Tables2Traces provides an explicit and inspectable supervision signal: each fine-tuning example includes a structured reasoning trace that researchers can review and analyze. This transparency may support interpretability research during model development, since the training-time reasoning distribution is observable rather than implicit. However, these traces are not clinical explanations and should not be used for medical decision-making. Their interpretability value is limited to understanding the supervision signal itself rather than conferring interpretability at inference time.

As our clinician review indicates, the traces can omit relevant context or exhibit overconfident interpretations because they are based on limited structured features rather than full patient histories. Any biases present in the underlying observational dataset may also propagate into the synthetic traces and the resulting model. These traces are therefore intended solely as a research supervision signal rather than as clinically validated explanations. Combining synthetic reasoning with expert-curated or hybrid supervision represents a promising direction for improving fidelity in future work. More broadly, while tabular supervision can make domain adaptation more accessible, it also inherits any biases present in the underlying dataset. Ensuring the fidelity and safety of synthetic traces in real-world settings remains an important follow-up direction beyond the scope of this study. Additionally, systematically studying how performance varies with the amount of tabular supervision as well as identifying other sources of synthetic supervision are important avenues for future research.

ETHICS STATEMENT

This work uses de-identified UK Biobank data accessed under approved use; all participants provided informed consent and data collection was overseen by the UK Biobank ethics framework. Our method, Tables2Traces, generates synthetic reasoning traces from structured records to fine-tune language models for research purposes only. The models and traces are *not* clinical devices and must not be used for diagnosis or treatment. To gauge plausibility and safety, two cardiologists qualitatively reviewed 10 randomly sampled traces independently using a structured rubric. Clinician review confirmed no safety concerns but did note overconfidence by the model, reflecting the inherent limitations of synthetic data. The cardiology experts noted that outcomes may depend on factors not present in the tabular snapshot; our traces are therefore positioned as research-only supervision signals, not calibrated risk assessments or clinical guidance. We provide an overview of their comments in Appendix O. We release prompts and code to support auditability. Finally, our evaluation is restricted to public medical QA benchmarks and does not involve individual-level deployment or decision support.

REPRODUCIBILITY STATEMENT

All implementation details, prompts, hyperparameters, and evaluation procedures are documented in the Appendix. Upon acceptance we will release the full codebase and configs to reproduce preprocessing, trace generation, fine-tuning, and evaluation, together with exact seeds and scripts that render all tables and figures. Results on public benchmarks (MedQA, MedMCQA) are reproducible with our released scripts and seeds. UK Biobank data cannot be shared; researchers with approved access can regenerate the training traces using our scripts and instructions.

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

APPENDIX

## A  EVALUATION SETUP

We use a standardized evaluation pipeline across all models and benchmarks. Each multiple-choice question is formatted using the appropriate chat template (e.g., using standard templates like `AutoTokenizer.apply_chat_template`) and fed into the model for completion. Evaluation is performed using the vLLM framework with sampling-based generation (`temperature=0.6, top_p=0.85, n=3` completions per prompt, `frequency_penalty=1.5, presence_penalty=0.9, max_tokens=32768`). We extract the final answer (A–D) from the generated output using robust regex-based parsing, and fall back to the reasoning text if a clean answer is not present after multiple attempts.

We stop generation using model-specific stop tokens (e.g., `` for LLaMA, `<|EOT|>` for Qwen), as well as answer-format strings (e.g., `"Answer:  A"`). All completions are post-processed using a training-aware cleaner to remove template artifacts (e.g., `"Assistant:  "` headers). For models fine-tuned on structured reasoning traces, we additionally parse the `<think>...</think>` block and extract the final prediction from the trailing answer segment.

The pipeline includes automatic retries for failed generations, and safely extracts answers even under high sampling variability. This setup ensures consistent evaluation across all models and supports multi-sample decoding strategies such as best-of-$n$, majority vote, and worst-of-$n$.

### A.1  EVALUATION METRICS

All performance metrics are aggregated from 10 independent inference runs per model. For each test question, we collect a binary correctness label (extracted using a robust regex-based parsing) for each of the 10 completions and compute the following evaluation metrics:

- **Average Accuracy**: The average correctness across the 10 runs for each question.
- **Best-of-$n$**: The question is marked correct if at least one of the 10 completions is correct.
- **Majority Vote**: The question is marked correct if a majority of the 10 completions are correct. In the case of a tie, the outcome defaults to incorrect.
- **Worst-of-$n$**: The question is marked correct only if all 10 completions are correct.

Category-level and overall scores are computed by averaging across all test questions per category. Error bars represent the standard error of the mean (SEM) across test examples. Additionally, we report the relative percent change in average accuracy compared with the Base model. In all results tables, the best-performing model is shown in bold for each metric within each category. If multiple models have the same value after rounding, all are shown in bold.

## B   TRAINING CONFIGURATION DETAILS

All models are fine-tuned using Open-R1's supervised fine-tuning pipeline (HuggingFace, 2025), with a single epoch of training on 4×A100 80GB GPUs. We use `FlashAttention-2` and `bfloat16` precision for all experiments. Below, we describe shared configurations and model-specific differences.

### B.1   SHARED CONFIGURATION

- **Precision:** bfloat16 with FlashAttention-2

- **Epochs:** 1 full pass over the training set

- **Batch Size:** 2 per device, 8 gradient accumulation steps

- **Optimizer:** AdamW with learning rate = `5e-6`, cosine decay (min LR ratio = 0.1), weight decay = 0.0001

- **Max Sequence Length:** 32,768 tokens

- **Evaluation:** Every 500 steps on the test split

- **Checkpointing:** Saved every 500 steps, keep only latest

- **Logging:** Via `wandb`, every 5 steps

- **Seed:** 42

- **Gradient Checkpointing:** Enabled (non-reentrant)

- **System Prompt:**

```
You are a helpful AI Assistant that provides
well-reasoned and detailed responses.
You first think about the reasoning process as an internal monologue
and then provide the user with the answer.
Respond in the following format:
<think> ... </think>
<answer> ... </answer>
```

- **Chat Template:** Modified to include reasoning tags (`<think>...</think>`) in the completion and exclude them from the prefill.

## B.2 MODEL VARIANTS

We fine-tune two architectures on two dataset variants, resulting in four total models:

| Model Architecture | Training Data |
|---|---|
| DeepSeek-R1-Distill-Qwen-7B | Patient Descriptions (Tables2Traces (simple)) |
| DeepSeek-R1-Distill-Qwen-7B | Counterfactual Traces (Tables2Traces) |
| DeepSeek-R1-Distill-Llama-8B | Patient Descriptions (Tables2Traces (simple)) |
| DeepSeek-R1-Distill-Llama-8B | Counterfactual Traces (Tables2Traces) |

The **patient descriptions** dataset consists of direct narrative renderings of individual tabular rows, while the **counterfactual traces** dataset includes contrastive triplets with structured reasoning (as described in Section 3). All datasets are processed using 48 parallel workers.

Table 2: Training runtimes for each model variant.

| Model Variant | Architecture | Runtime |
|---|---|---|
| Tables2Traces | 8B (LLaMA) | 20h 37m |
| Tables2Traces (simple) | 8B (LLaMA) | 9h 24m |
| Tables2Traces | 7B (Qwen) | 19h 52m |
| Tables2Traces (simple) | 7B (Qwen) | 9h 18m |

## C PROMPT TEMPLATES

This section documents all prompt templates used during dataset construction, training, and evaluation. Strings enclosed in curly brackets (e.g., {column_names}) represent placeholders that are dynamically replaced with instance-specific values at runtime, similar to Python f-strings.

### C.1 COLUMN NAME MAPPING (TABLE REPRESENTATION)

**Purpose:** Transform raw or abbreviated column headers into clinically accurate feature names.

**Placeholders:** column_names is replaced with a list of all columns of the dataset.

> **Column Name Mapping**
>
> ```
> You are a powerful AI with expertise in medicine.
> You are given a dataset with columns that relate to patients where each patient is a row
> and each column contains different information pertaining to the patient.
>
> As your first task, you are tasked with converting a list of column names that are
> possibly abbreviated or not easy to understand into a fully understandable name for
> medical professionals.
> Please provide the output as a Python dictionary.
>
> The list of column names is: {column_names}
> ```

### C.2 PATIENT DESCRIPTION GENERATION

**Purpose:** Convert structured patient rows into fluent narrative case descriptions.

**Placeholders:** json_file is replaced with a json-file containing the column names as keys and the values of columns as values.



**Patient Description Generation**

```
You are a powerful AI with expertise in medicine.
Your task is to generate a detailed and exhaustive text description for a patient.
You are given all the patient information in a json-format, which contains the clinical attributes
and the results from laboratory tests from real world patients.
The patients in question are patients with cardiovascular disease.
The reader of the description is an expert within this particular medical domain.
The language used in the description should reflect your domain expertise
and your medical reasoning capabilities.
Please provide as many details as possible.
You should ONLY include the patient description!

The json-file containing the information from the patient: {json_file}
```


## C.3 CONTRASTIVE REASONING AND COUNTERFACTUAL TRACES

**Purpose:** Generate reasoning traces comparing a target patient to contrasting neighbors.

**Placeholders:** `target_outcome` is the outcome (Alive / Dead) for the target patient. `survivor_description` is the text description of the nearest neighbor to the target patient who had the outcome "Alive". `survivor_description` is the text description of the nearest neighbor to the target patient who had the outcome "Dead". `target_description` is the text description target patient. All text descriptions are derived using the Patient Description Generation prompt in Appendix C.2.



**Counterfactual Task Generation**

```
### Role ###
Clinical AI analyzing patient outcomes using contrastive case pairs.

### Input Data ###
1) Target patient (labeled {target_outcome})
2) Nearest neighbor who survived
3) Nearest neighbor who died

=== CLOSEST SURVIVOR ===
{survivor_description}
=== CLOSEST DEATH ===
{death_description}
=== TARGET PATIENT ===
{target_description}

### Required Analysis ###
1. Comparison:
    a) Identify 1-3 decisive differences between target and NNs
    b) Focus on features present in ALL THREE cases
    c) Flag any conflicting evidence

2. Label Evaluation:
    a) Assess if {target_outcome} is correct
    b) Confidence score (1-5)

3. Counterfactual:
    a) Modify one feature present in NNs
    b) Predict outcome change
    c) Justify using specific NN evidence

### Response Format ###
1. Comparison:
    1) Outcome alignment: <...>
    2) Decisive factors: ...

2. Label assessment:
    1) Correctness: <...>
    2) Confidence: <...>

3. Counterfactual:
    1) Modification: <...>
    2) Outcome: <...>
    3) Evidence: <...>
```


## C.4 CATEGORIZATION: PATIENT-SPECIFIC VS. ABSTRACT

**Purpose:** Categorize questions as either patient-specific or abstract.

**Placeholders:** `question` is the specific question to be categorized.

```
Patient-specific Categorization

You are a clinical reasoning expert.
Your task is to determine whether a multiple-choice medical question is *patient-specific*.

Definitions:
- A question is **patient-specific** if it describes
a particular patient case -- including their symptoms, medical history, age, lab results, etc.
These questions simulate real-life clinical decision-making.
- A question is **not patient-specific** if it
asks about general medical knowledge or includes references to people (e.g., doctors, nurses)
but *not to a patient's condition*.

Return: {{"patient_specific": true}} or {{"patient_specific": false}}

Examples:

Example 1:
Question: A 67-year-old man presents with sudden chest pain and shortness of breath.
Which of the following is the most likely diagnosis?
Answer: {{"patient_specific": true}}

Example 2:
Question: What is the most common cause of mitral stenosis worldwide?
Answer: {{"patient_specific": false}}

Example 3:
Question: A physician enters the operating room without washing his hands.
What is the correct protocol in this situation?
Answer: {{"patient_specific": false}}

---

Now classify the following question:

Question: {question}
Answer:
```

## C.5 CATEGORIZATION: CARDIOVASCULAR VS. NON-CARDIOVASCULAR (MEDMCQA)

**Purpose:** Categorize MedMCQA questions as either cardiovascular related or not.

**Placeholders:** `question` is the specific question to be categorized.

```
Cardiovascular Categorization

You are a medical assistant helping categorize medical questions.
Given a question, determine whether it pertains to cardiovascular diseases or not.

Only answer `true` or `false` depending on whether the core topic of the question
involves cardiovascular systems, diseases, symptoms, diagnostics, or treatment.
Cardiovascular topics include (but are not limited to) conditions such as: hypertension,
myocardial infarction, arrhythmias, heart failure, atherosclerosis, angina, or cardiac arrest.

Avoid false positives: questions mentioning blood pressure, heart rate, or medications like
beta-blockers must still be relevant to cardiovascular context to count.

Output your answer in the following JSON format:

{{"cardiovascular_related": true}}

Question:
{question}
```

## C.6 CATEGORIZATION: MEDICAL DOMAIN (MEDQA)

**Purpose:** Categorize the medical domain of MedQA questions.

**Placeholders:** `question` is the specific question to be categorized.

---

### Medical Domain Categorization

```
Analyze the medical question and respond EXACTLY as follows:

---STRICT RULES---
1. SINGLE HIGH-CONFIDENCE CATEGORY (>=0.7):
   • If ONE category scores >=0.7:
   "Category = Score"
   • If MULTIPLE categories score >=0.7:
   Choose ONLY THE HIGHEST SCORE (if tie, pick first alphabetically)

2. MULTIPLE LOW-CONFIDENCE CATEGORIES (all <0.7):
   • "Primary: Category1 = Score1, Secondary: Category2 = Score2, Tertiary: Category3 = Score3"

3. IRRELEVANT:
   • "None of the above = 1.0"

---VALID EXAMPLES---
• "Cardiovascular = 0.85"
• "Primary: Infectious = 0.6, Secondary: Hematologic = 0.3, Tertiary: Renal = 0.1"
• "None of the above = 1.0"

---CATEGORIES (ALPHABETICAL ORDER)---
Cardiovascular, Dermatologic, Endocrine/Metabolic, Gastrointestinal, Hematologic,
Immunologic, Infectious, Musculoskeletal, Neurological, Obstetrics/Gynecology,
Oncology, Pediatric, Psychiatric, Renal/Genitourinary, Respiratory, Toxicology

---QUESTION---
{question}

---YOUR RESPONSE (MUST MATCH EXACTLY ONE FORMAT ABOVE)---
```

## C.7 EVALUATION PROMPTS: MEDQA

**Purpose:** Evaluate model on MedQA using reasoning-aware prompting.

---

**MedQA Evaluation Prompt**

```
**Role**: You are a medical knowledge expert.
**Task**: Analyze the following multiple-choice medical question by following these steps:
1. First, use critical clinical reasoning to think about the question
step-by-step before giving a final answer.
2. After completing your reasoning, directly provide your final answer.
3. **IMPORTANT**: Do not provide any explanation beyond your answer in the final output.

**Response Format**:
[Your step-by-step reasoning goes here]
Answer: [Your final choice: A, B, C, or D]
```

---

## C.8 EVALUATION PROMPTS: MEDMCQA

**Purpose:** Evaluate model on MedMCQA using structured reasoning steps.

---

**MedMCQA Evaluation Prompt**

```
**Role**: You are a medical knowledge expert.
**Task**: Analyze the following multiple-choice medical question by reasoning
step-by-step before selecting the best answer.

Follow these steps:
1. Identify the topic and relevant concepts.
2. Recall or infer medical knowledge needed to solve the question.
3. Eliminate incorrect options and justify your final choice.
4. Provide your answer clearly and concisely.

**Response Format**:
[Step-by-step reasoning]
Answer: [A, B, C, or D]

**Constraints**:
- Do not explain anything after giving your final answer.
- Only choose a single option from A-D.
```

# D    QUALITATIVE EXAMPLES

This section provides representative examples of reasoning traces generated during training and evaluation. These examples illustrate the structure, fluency, and clinical plausibility of model outputs for both Tables2Traces and Tables2Traces (simple). All examples are drawn directly from the training data without curation or filtering. Specifically, the example shown here (both the patient description task and the counterfactual task) corresponds to the first available row in the dataset and is included with minimal editing beyond redaction for privacy.

## D.1    QUALITATIVE EXAMPLE: PATIENT DESCRIPTION REASONING (REDACTED)

We include a representative example of a reasoning trace generated from a single patient description prompt. The input data has been redacted for privacy, but the structure of the trace is preserved. The resulting model response corresponds to the training data used for the Tables2Traces (simple) variant and as the generated patient descriptions used for generating the counterfactual tasks.

---

**Patient JSON Input (Redacted)**

You are a powerful AI with expertise in medicine. Your task is to generate a detailed and exhaustive text description for a patient. You are given all the patient information in a json-format, which contains the clinical attributes and the results from laboratory tests from real world patients. The patients in question are patients with cardiovascular disease. The reader of the description is an expert within this particular medical domain. The language used in the description should reflect your domain expertise and your medical reasoning capabilities. Please provide as many details as possible. You should ONLY include the patient description!

———

The json-file containing the information from the patient:

```
Sex                                                   ...
Age (Years)                                           ...
Weight (Kilograms)                                    ...
Height (Centimeters)                                  ...
Smoking Amount (Cigarettes per Day)                   ...
Atrial Fibrillation Diagnosis                         ...
Chronic Kidney Disease Diagnosis                      ...
Rheumatoid Arthritis Diagnosis                        ...
Anti-diabetic Medication Use                          ...
Anti-hypertensive Medication Use                      ...
Diabetes History                                      ...
Lipid-lowering Medication Use                         ...
Oral Contraceptive Use                                ...
Blood Glucose Level (mmol/L)                          ...
Hemoglobin A1c (HbA1c) Percentage                     ...
White Blood Cell Count (x10^9/L)                      ...
Serum Creatinine (µmol/L)                             ...
Triglyceride Level (mmol/L)                           ...
Uric Acid Level (µmol/L)                              ...
Cystatin C Level (mg/L)                               ...
Systolic Blood Pressure (mmHg)                        ...
Urine Microalbumin Concentration (mg/L)               ...
C-Reactive Protein (CRP) Level (mg/L)                 ...
Family History of Cardiovascular Disease (CVD)        ...
Atypical Antipsychotic Medication Use                 ...
Corticosteroid Medication Use                         ...
Migraine History                                      ...
Severe Mental Illness Diagnosis                       ...
Systemic Lupus Erythematosus (SLE) Diagnosis          ...
Total Cholesterol Level (mmol/L)                      ...
High-Density Lipoprotein (HDL) Cholesterol Level (mmol/L)  ...
Ethnicity                                             ...
```

---

---

**Model Reasoning (Redacted)**

<think> Okay, let's start by going through all the patient data provided. The patient is a [...]-year-old [...] [...] with a height of [...] cm and weight of [...] kg. Calculating his BMI: weight divided by height squared in meters. So [...] kg / ([...] m)$^2$ = [...] kg/m$^2$. That puts him in the overweight category, which is a risk factor for cardiovascular issues.

He's a smoker, [...] cigarettes per day. That's significant. Smoking is a major risk factor for cardiovascular diseases, including atherosclerosis and hypertension. His systolic blood pressure is [...] mmHg, which is way above the normal range. Stage 2 hypertension. But he's not on anti-hypertensive meds, which is concerning. Uncontrolled hypertension can lead to heart failure, stroke, etc.

Looking at his lab results. Blood glucose is [...] mmol/L, which is within normal fasting levels. HbA1c is [...]%, also normal, so no indication of diabetes. No history of diabetes either. So glucose metabolism seems okay. But his lipid profile: total cholesterol [...] mmol/L, which is a bit elevated. Triglycerides at [...] mmol/L, which is borderline high. HDL is [...] mmol/L, which is on the lower side for males. Low HDL and elevated triglycerides could contribute to atherosclerosis. But he's not on lipid-lowering meds, so that's another risk factor not being managed.

Serum creatinine is [...] μmol/L. Using the CKD-EPI equation for eGFR. For a [...]-year-old [...], serum creatinine of [...] converts to an eGFR around [...] mL/min/[...]m², which is normal. Cystatin C is [...] mg/L, which also correlates with normal kidney function. Urine microalbumin is [...] mg/L, which is within normal range. So no signs of chronic kidney disease, which aligns with the CKD diagnosis being [...].

CRP is [...] mg/L, which is normal. Low CRP suggests no significant inflammation. No rheumatoid arthritis, SLE, or other inflammatory conditions. Uric acid is [...] μmol/L. Normal range for males is around 416, so his is within normal. Not a hyperuricemia case, so less concern for gout.

He has atrial fibrillation (diagnosis 1), which is a major risk for stroke and heart failure. Given his age and hypertension, that's a critical factor. No family history of CVD, so maybe environmental factors like smoking and hypertension are the main contributors here. No other comorbidities like severe mental illness or migraines.

Other meds: Not on anti-diabetics, lipid-lowering, or antihypertensives. So his AFib and hypertension are untreated, which is alarming. His high systolic BP could be exacerbating the atrial fibrillation, increasing stroke risk. The CHA$_2$DS$_2$-VASc score would be [...], indicating anticoagulation should be considered.

In summary, this patient has multiple modifiable risk factors: smoking, obesity, uncontrolled hypertension, dyslipidemia. The presence of atrial fibrillation without anticoagulation or rate control meds significantly elevates his thromboembolic risk. The absence of diabetes or kidney disease is a positive note, but the untreated hypertension and smoking are critical areas needing intervention. His elevated systolic BP is a key driver for cardiovascular events. Management should focus on smoking cessation, initiating antihypertensive therapy, lipid management, and evaluating for anticoagulation given his AFib. </think>

---

**Model Response (Redacted)**

The patient is a [...]-year-old [...] [...] with a body mass index (BMI) of [...] kg/m² (height: [...] cm; weight: [...] kg), categorizing him as overweight. He reports a significant smoking history of [...] cigarettes/day, a critical modifiable cardiovascular risk factor. Key clinical findings include **stage [...] hypertension** (systolic blood pressure: [...] mmHg) without current anti-hypertensive therapy, **atrial fibrillation** (diagnosis 1), and a lipid profile suggestive of mixed dyslipidemia (total cholesterol: [...] mmol/L, triglycerides: [...] mmol/L, HDL-C: [...] mmol/L). Notably, his LDL-C (calculated via Friedewald equation: [...] mmol/L) exceeds optimal targets, yet he is not on lipid-lowering therapy.

Metabolically, he exhibits normoglycemia (fasting glucose: [...] mmol/L; HbA1c: [...]%) without evidence of diabetes or anti-diabetic medication use. Renal function is preserved (serum creatinine: [...] μmol/L; eGFR >[...] mL/min/[...]m² by CKD-EPI; cystatin C: [...] mg/L; urine microalbumin: [...] mg/L), ruling out chronic kidney disease. Inflammatory markers are unremarkable (CRP: [...] mg/L), with no history of autoimmune or systemic inflammatory conditions. Uric acid levels ([...] μmol/L) are within normal limits, mitigating concern for gout-related contributions.

The absence of anti-hypertensive or anticoagulant therapy in the setting of atrial fibrillation and severe hypertension raises immediate concern for thromboembolic risk (CHA$_2$DS$_2$-VASc score $\leq$ [...]) and hypertensive end-organ damage. His lipid abnormalities—elevated triglycerides, low HDL-C, and borderline-high LDL-C—suggest residual atherogenic risk despite normal BMI-adjusted adiposity metrics. The combination of smoking, untreated hypertension, and dyslipidemia synergistically amplifies his cardiovascular risk profile, warranting aggressive risk factor modification.

No family history of CVD or secondary causes of hypertension (e.g., renal, endocrine) is documented. The absence of diabetes, chronic kidney disease, or inflammatory comorbidities narrows the primary drivers of his cardiovascular pathology to lifestyle factors (smoking, diet) and age-related vascular stiffening. Urgent interventions should prioritize smoking cessation, BP control (target <130/80 mmHg per ACC/AHA guidelines), statin initiation, and atrial fibrillation management with anticoagulation and rhythm control strategies.

## D.2 Qualitative Example: Counterfactual Reasoning (Redacted)

We include a representative example of a reasoning trace generated from a counterfactual reasoning trace for the first row of the dataset. The input data has been redacted for privacy, but the structure of the trace is preserved. This format corresponds to the Tables2Traces variant. For readability, the input to the counterfactual task is shown in three parts. In practice, the full prompt is passed to the LLM as a single string. The split here is purely for presentation purposes.

## Counterfactual Input (Redacted) - Part 1

```
### Role ###
Clinical AI analyzing patient outcomes using contrastive case pairs.

### Input Data ###
1) Target patient (labeled Dead)
2) Nearest neighbor who survived
3) Nearest neighbor who died

=== CLOSEST SURVIVOR ===
**Patient Description**
The patient is a **[...]-year-old [...] male**
with a body mass index (BMI) of **[...] kg/m^2** (weight: [...] kg,
height: [...] cm)), categorizing him as **overweight**,
a significant modifiable risk factor for cardiovascular disease (CVD).
He reports a **[...]-cigarette/day smoking history**, a major independent risk factor
for atherosclerotic CVD and thromboembolic events.

**Cardiovascular and Comorbidity Profile**:
- **Atrial fibrillation (AF)** is confirmed (diagnosis code present),
elevating his risk of thromboembolic complications, including stroke.
- **No diabetes mellitus** (HbA1c: [...]%, fasting glucose: [...] mmol/L))
or chronic kidney disease (CKD) (serum creatinine: [...] mumol/L,
cystatin C: [...] mg/L, urine microalbumin: [...] mg/L).
- **Uncontrolled hypertension** (systolic BP: [...] mmHg) is evident, with no current use
of anti-hypertensive medications, suggesting suboptimal risk factor management.
- **Hyperlipidemia** is present (total cholesterol: [...] mmol/L, HDL: [...] mmol/L,
triglycerides: [...] mmol/L), with an estimated LDL-C of **~[...]
mmol/L** (Friedewald equation), indicative of significant dyslipidemia.
Despite this, no lipid-lowering therapy is documented.

**Inflammatory and Metabolic Markers**:
- **C-reactive protein (CRP)** is within normal limits ([...] mg/L),
suggesting no acute systemic inflammation.
- **Uric acid** levels are borderline elevated ([...] mumol/L),
though below the threshold for clinical hyperuricemia.

**Additional Risk Stratification**:
- **No family history of CVD**, autoimmune disease (e.g., rheumatoid arthritis, SLE),
or severe mental illness.
- **Absence of diabetic, antihypertensive, or lipid-lowering pharmacotherapy**
highlights potential undertreatment of modifiable CVD risk factors.

**Clinical Synthesis**:
This patient presents with **high-risk cardiovascular profile** driven by **age,
smoking, untreated hypertension, and significant hypercholesterolemia**,
compounded by **AF-related thromboembolic risk**.
The absence of diabetes or CKD does not mitigate his overall risk,
as his ASCVD (atherosclerotic cardiovascular disease)
risk score would likely place him in a high-risk category.
Urgent interventions should prioritize **smoking cessation, BP control
(target <130/80 mmHg per guidelines), and statin therapy**
(high-intensity statin indicated for LDL-C reduction >50%).
**Anticoagulation for AF** (CHA_2DS_2-VASc score >=2 given age >=[...] and hypertension)
should be evaluated to mitigate stroke risk.
Close monitoring of renal function (cystatin C-based eGFR) and
lipid profiles is warranted to guide therapeutic efficacy and adherence.
```

## Counterfactual Input (Redacted) - Part 2

```
=== CLOSEST DEATH ===
The patient is a [...]-year-old [...] male with a body mass index (BMI) of
[...] kg/m^2 (weight: [...] kg, height: [...] cm), categorizing him as overweight.
He is an active smoker with a significant tobacco exposure of [...] cigarettes/day,
a major independent risk factor for atherosclerotic cardiovascular disease (ASCVD).
His medical history is notable for atrial fibrillation (AFib),
a critical arrhythmia conferring a 5-fold increased risk of thromboembolic events,
but no evidence of chronic kidney disease (CKD),
diabetes mellitus (DM), or autoimmune disorders.

**Cardiometabolic Profile:**
- **Hypertension:** Uncontrolled stage 2 hypertension (systolic BP: [...] mmHg)
without current antihypertensive therapy. This elevates his 10-year ASCVD risk
substantially, particularly when combined with smoking.
- **Lipid Abnormalities:** Borderline-high total cholesterol ([...] mmol/L) with
elevated calculated LDL-C (~[...] mmol/L via Friedewald equation) and suboptimal
HDL-C ([...] mmol/L), consistent with atherogenic dyslipidemia.
Triglycerides are within normal limits ([...] mmol/L).
- **Glycemic Status:** Normoglycemic (fasting glucose: [...] mmol/L;
HbA1c: [...]%), excluding DM.

**Renal & Inflammatory Markers:**
- Preserved renal function: Serum creatinine [...] mumol/L (eGFR ~[...] mL/min/[...]m^2
by CKD-EPI), cystatin C [...] mg/L, and normoalbuminuria (urine microalbumin: [...] mg/L).
- Mild systemic inflammation: CRP [...] mg/L, potentially reflecting endothelial
dysfunction from smoking or subclinical atherosclerosis.

**Additional Risk Stratification:**
- Absence of protective factors: No lipid-lowering, antihypertensive,
or anticoagulant use despite AFib and ASCVD risk factors.
- No familial predisposition to CVD or contributory medication use
(e.g., corticosteroids, antipsychotics).

**Clinical Implications:**
This patient exhibits a high-risk phenotype for ASCVD
and thromboembolism due to the confluence of modifiable factors: smoking,
untreated hypertension, dyslipidemia, and AFib. The absence of diabetes
or CKD does not mitigate this risk, as his SCORE2-Diabetes equation-adjusted
10-year CVD risk likely exceeds 10%. Urgent interventions should include
smoking cessation, BP control (target <130/80 mmHg per ACC/AHA guidelines),
high-intensity statin therapy (e.g., atorvastatin 40-80 mg),
and CHA_2DS_2-VASc-guided anticoagulation for AFib. Weight optimization and
serial monitoring of inflammatory markers (e.g., CRP)
and cardiac troponins could further refine prognostication.

=== TARGET PATIENT ===
The patient is a [...]-year-old [...] male with a body mass index (BMI) of
[...] kg/m^2 (height: [...] cm; weight: [...] kg), categorizing him as overweight.
He reports a significant smoking history of [...]
cigarettes/day, a critical modifiable cardiovascular risk factor.
Key clinical findings include **stage 2 hypertension**
(systolic blood pressure: [...] mmHg) without current anti-hypertensive
therapy, **atrial fibrillation** (diagnosis confirmed), and a lipid profile
suggestive of mixed dyslipidemia (total cholesterol:
[...] mmol/L, triglycerides: [...] mmol/L, HDL-C: [...] mmol/L).
Notably, his LDL-C (calculated via Friedewald equation: [...] mmol/L) exceeds
optimal targets, yet he is not on lipid-lowering therapy.

Metabolically, he exhibits normoglycemia (fasting glucose: [...] mmol/L; HbA1c: [...]%)
without evidence of diabetes or anti-diabetic medication use. Renal function is
preserved (serum creatinine: [...] mumol/L; eGFR >[...] mL/min/[...] by CKD-EPI;
cystatin C: [...] mg/L; urine microalbumin: [...] mg/L), ruling out chronic kidney
disease. Inflammatory markers are unremarkable (CRP: [...] mg/L), with no history of
autoimmune or systemic inflammatory conditions. Uric acid levels ([...] mumol/L)
are within normal limits, mitigating concern for gout-related contributions.

The absence of anti-hypertensive or anticoagulant therapy in the setting of atrial
fibrillation and severe hypertension raises immediate concern for
thromboembolic risk (CHA_2DS_2-VASc score >=2) and hypertensive end-organ damage.
His lipid abnormalities--elevated triglycerides,
low HDL-C, and borderline-high LDL-C--suggest
residual atherogenic risk despite normal BMI-adjusted adiposity metrics.
The combination of smoking, untreated hypertension, and dyslipidemia
synergistically amplifies his cardiovascular risk profile,
warranting aggressive risk factor modification.
```

## Counterfactual Input (Redacted) - Part 3

```
No family history of CVD or secondary
causes of hypertension (e.g., renal, endocrine) is documented.
The absence of diabetes, chronic kidney disease, or inflammatory comorbidities
narrows the primary drivers of his cardiovascular pathology to lifestyle factors
(smoking, diet) and age-related vascular stiffening. Urgent interventions
should prioritize smoking cessation, BP control (target <130/80 mmHg per
ACC/AHA guidelines), statin initiation, and atrial fibrillation management with
anticoagulation and rhythm control strategies.

### Required Analysis ###
1. Comparison:
   a) Identify 1-3 decisive differences between target and NNs
   b) Focus on features present in ALL THREE cases
   c) Flag any conflicting evidence (e.g., "Target aligns with NN1 in [X] but NN2 in [Y]")

2. Label Evaluation:
   a) Assess if Dead is correct
   b) Confidence score (1-5):
      5 = All evidence strongly agrees
      4 = Most evidence agrees
      3 = Mixed evidence
      2 = Minimal supporting evidence
      1 = No discernible pattern

3. Counterfactual:
   a) Modify one feature present in NNs
   b) Predict outcome change
   c) Justify using specific NN evidence

### Response Format ###
1. Comparison:
   1) Outcome alignment: <Matches Survivor/Matches Death/Uncertain>
   2) Decisive factors:
      1) <Feature>: Target vs NN1 vs NN2
      2) <Feature>: Target vs NN1 vs NN2

2. Label assessment:
   1) Correctness: <Correct/Incorrect/Edge Case>
   2) Confidence: <1-5 with brief rationale>

3. Counterfactual:
   1) Modification: <Feature + change>
   2) Outcome: <Survived/Died>
   3) Evidence: <"Matches [NN] where [feature]=[value]">

### Quality Rules ###
- If confidence =<2, state: "Uncertain because [reason]"
- Counterfactuals must reference features actually present in NNs
- Use original feature names from input data
```

## Model Reasoning (Redacted)

<think> Okay, let's start by looking at the three patients: the target who died, the closest survivor (NN1), and the closest death (NN2). All three are [...] males in their [...], overweight, with a [...]-cigarette/day smoking history, atrial fibrillation, uncontrolled hypertension, and untreated hyperlipidemia. They also don't have diabetes or CKD.

First, comparing the target to the survivor and the other deceased. The target's systolic BP is [...] mmHg, which is higher than both NN1 ([...]) and NN2 ([...]). That's a big difference. High BP is a major risk factor, so maybe the target's much higher BP contributed to his death.

Next, looking at LDL-C levels. The target's LDL is [...] mmol/L, which is lower than NN1's [...] but higher than NN2's [...]. Wait, NN2's LDL was [...], so the target is actually lower than NN2. Hmm, but the survivor had the highest LDL. That's conflicting. Maybe LDL isn't the decisive factor here.

Triglycerides: Target has [...], which is higher than both NN1 ([...]) and NN2 ([...]). Elevated triglycerides can contribute to cardiovascular risk, so this might be a key difference. The target's higher triglycerides could have increased his risk compared to the other two.

So the main differences are systolic BP and triglycerides. The target's BP is the highest, and his triglycerides are higher than both NNs. Even though his LDL is in between, the other factors might be more critical.

For the label assessment, the target died. Both NNs had similar profiles but the target's BP and triglycerides are worse. The survivor had lower BP and lower triglycerides, while the deceased had lower BP than the target. So the higher BP in the target might explain the death. The confidence is high because BP and triglycerides are significant risk factors, and the target's values are worse.

For the counterfactual, if we lower the target's systolic BP to [...] mmHg (like NN1), maybe he would have survived. NN1 had lower BP and survived, so reducing BP to that level might change the outcome. Alternatively, lowering triglycerides to match NN1 or NN2 could also help, but BP seems more directly impactful. The evidence from NN1 shows that with lower BP, survival occurred despite other risks. </think>

## Model Response (Redacted)

```
### 1. Comparison:
1) **Outcome alignment**: Matches Death
2) **Decisive factors**:
   1) **Systolic BP**: Target ([...] mmHg) > NN1 ([...] mmHg) > NN2 ([...] mmHg)
      - Target's severe hypertension (stage 2) exceeds both NNs,
      increasing acute cardiovascular event risk (e.g., stroke, myocardial infarction).
   2) **Triglycerides**: Target ([...] mmol/L) > NN2 ([...] mmol/L) > NN1 ([...] mmol/L)
      - Elevated triglycerides in the target suggest greater
      residual atherogenic risk despite comparable LDL-C levels.

### 2. Label assessment:
1) **Correctness**: Correct
2) **Confidence**: 4/5
   - Rationale: The target's **higher systolic BP** and **triglycerides** align
   with worse outcomes. Both NNs had lower BP and triglycerides, yet the target died.
   However, conflicting evidence exists in LDL-C
   (target: [...] mmol/L vs. NN1: [...] mmol/L),
   where the survivor paradoxically had higher LDL-C. This reduces confidence to 4.

### 3. Counterfactual:
1) **Modification**: Reduce systolic BP to [...] mmHg (NN1's level)
2) **Outcome**: Survived
3) **Evidence**: Matches **NN1** where systolic BP=[...] mmHg (untreated but lower
than target's [...] mmHg). NN1 survived despite similar AFib, smoking, and untreated
hyperlipidemia, suggesting BP control is a pivotal survivability factor.
```

Table 3: Per-category evaluation metrics on MedMCQA for Base and Tables2Trace (8B).

| Category | Model Type | Avg Accuracy | Best-of-n | Majority Vote | Worst-of-n | % Change |
|---|---|---|---|---|---|---|
| **Anaesthesia** | Base | **0.36 ± 0.07** | **0.88 ± 0.07** | **0.29 ± 0.09** | **0.08 ± 0.06** | |
| *(N = 24)* | Tables2Traces | **0.36 ± 0.07** | 0.83 ± 0.08 | **0.29 ± 0.09** | **0.08 ± 0.06** | -1.15% ↓ |
| **Anatomy** | Base | 0.36 ± 0.02 | **0.86 ± 0.03** | 0.26 ± 0.04 | 0.02 ± 0.01 | |
| *(N = 147)* | Tables2Traces | **0.40 ± 0.02** | **0.88 ± 0.03** | **0.31 ± 0.04** | **0.07 ± 0.02** | +11.91% ↑ |
| **Biochemistry** | Base | 0.57 ± 0.03 | 0.90 ± 0.03 | **0.58 ± 0.04** | 0.11 ± 0.03 | |
| *(N = 122)* | Tables2Traces | **0.59 ± 0.03** | **0.95 ± 0.02** | 0.56 ± 0.05 | **0.17 ± 0.03** | +3.62% ↑ |
| **Dental** | Base | 0.35 ± 0.01 | 0.82 ± 0.01 | 0.26 ± 0.02 | 0.05 ± 0.01 | |
| *(N = 845)* | Tables2Traces | **0.39 ± 0.01** | **0.88 ± 0.01** | **0.28 ± 0.02** | **0.06 ± 0.01** | +9.24% ↑ |
| **ENT** | Base | 0.39 ± 0.05 | **0.92 ± 0.04** | 0.26 ± 0.07 | **0.08 ± 0.04** | |
| *(N = 39)* | Tables2Traces | **0.45 ± 0.05** | **0.92 ± 0.04** | **0.36 ± 0.08** | **0.08 ± 0.04** | +16.56% ↑ |
| **Forensic Medicine** | Base | **0.41 ± 0.05** | **0.89 ± 0.05** | **0.32 ± 0.07** | 0.09 ± 0.04 | |
| *(N = 44)* | Tables2Traces | **0.41 ± 0.05** | **0.89 ± 0.05** | 0.30 ± 0.07 | **0.14 ± 0.05** | -1.10% ↓ |
| **Gynaecology & Obstetrics** | Base | 0.40 ± 0.03 | 0.81 ± 0.03 | 0.32 ± 0.04 | **0.09 ± 0.02** | |
| *(N = 154)* | Tables2Traces | **0.42 ± 0.03** | **0.82 ± 0.03** | **0.38 ± 0.04** | 0.08 ± 0.02 | +4.03% ↑ |
| **Medicine** | Base | 0.44 ± 0.03 | 0.84 ± 0.03 | 0.39 ± 0.04 | 0.12 ± 0.02 | |
| *(N = 185)* | Tables2Traces | **0.50 ± 0.03** | **0.88 ± 0.02** | **0.45 ± 0.04** | **0.15 ± 0.03** | +12.17% ↑ |
| **Microbiology** | Base | 0.45 ± 0.04 | 0.84 ± 0.04 | **0.35 ± 0.06** | 0.11 ± 0.04 | |
| *(N = 74)* | Tables2Traces | **0.48 ± 0.04** | **0.91 ± 0.03** | **0.35 ± 0.06** | **0.15 ± 0.04** | +7.55% ↑ |
| **Ophthalmology** | Base | 0.40 ± 0.05 | **0.91 ± 0.04** | **0.30 ± 0.07** | 0.14 ± 0.05 | |
| *(N = 43)* | Tables2Traces | **0.41 ± 0.05** | 0.88 ± 0.05 | **0.30 ± 0.07** | **0.16 ± 0.06** | +1.72% ↑ |
| **Orthopaedics** | Base | **0.40 ± 0.08** | **0.87 ± 0.09** | **0.53 ± 0.13** | **0.00 ± 0.00** | |
| *(N = 15)* | Tables2Traces | 0.34 ± 0.08 | 0.80 ± 0.11 | 0.27 ± 0.12 | **0.00 ± 0.00** | -15.00% ↓ |
| **Pathology** | Base | 0.51 ± 0.02 | 0.89 ± 0.02 | 0.44 ± 0.03 | 0.11 ± 0.02 | |
| *(N = 259)* | Tables2Traces | **0.54 ± 0.02** | **0.92 ± 0.02** | **0.53 ± 0.03** | **0.16 ± 0.04** | +5.82% ↑ |
| **Pediatrics** | Base | 0.44 ± 0.03 | 0.82 ± 0.03 | **0.39 ± 0.04** | 0.09 ± 0.02 | |
| *(N = 133)* | Tables2Traces | **0.47 ± 0.03** | **0.86 ± 0.03** | 0.38 ± 0.04 | **0.12 ± 0.03** | +6.52% ↑ |
| **Pharmacology** | Base | 0.52 ± 0.03 | 0.90 ± 0.02 | 0.46 ± 0.04 | **0.17 ± 0.03** | |
| *(N = 179)* | Tables2Traces | **0.56 ± 0.02** | **0.93 ± 0.02** | **0.55 ± 0.04** | 0.14 ± 0.03 | +8.30% ↑ |
| **Physiology** | Base | 0.46 ± 0.03 | **0.86 ± 0.03** | **0.38 ± 0.04** | **0.16 ± 0.03** | |
| *(N = 133)* | Tables2Traces | **0.47 ± 0.03** | **0.86 ± 0.03** | **0.38 ± 0.04** | 0.14 ± 0.03 | +2.30% ↑ |
| **Psychiatry** | Base | 0.41 ± 0.10 | 0.80 ± 0.13 | 0.30 ± 0.15 | **0.00 ± 0.00** | |
| *(N = 10)* | Tables2Traces | **0.54 ± 0.09** | **0.90 ± 0.10** | **0.50 ± 0.17** | **0.00 ± 0.00** | +31.71% ↑ |
| **Radiology** | Base | **0.49 ± 0.04** | **0.93 ± 0.03** | **0.40 ± 0.07** | **0.05 ± 0.03** | |
| *(N = 57)* | Tables2Traces | 0.45 ± 0.04 | 0.89 ± 0.04 | 0.39 ± 0.07 | 0.04 ± 0.02 | -8.54% ↓ |
| **Skin** | Base | **0.37 ± 0.08** | **0.91 ± 0.09** | **0.27 ± 0.14** | 0.00 ± 0.00 | |
| *(N = 11)* | Tables2Traces | 0.28 ± 0.10 | 0.73 ± 0.14 | 0.18 ± 0.12 | **0.09 ± 0.09** | -24.39% ↓ |
| **Social & Preventive Medicine** | Base | 0.44 ± 0.04 | 0.81 ± 0.04 | 0.34 ± 0.05 | **0.10 ± 0.03** | |
| *(N = 91)* | Tables2Traces | **0.47 ± 0.04** | **0.87 ± 0.04** | **0.43 ± 0.05** | 0.09 ± 0.03 | +7.30% ↑ |
| **Surgery** | Base | 0.41 ± 0.02 | **0.86 ± 0.02** | 0.35 ± 0.03 | **0.08 ± 0.02** | |
| *(N = 249)* | Tables2Traces | **0.46 ± 0.02** | **0.86 ± 0.02** | **0.40 ± 0.03** | **0.08 ± 0.02** | +12.12% ↑ |
| **Unknown** | Base | 0.30 ± 0.30 | 0.50 ± 0.50 | 0.50 ± 0.50 | 0.00 ± 0.00 | |
| *(N = 2)* | Tables2Traces | **0.35 ± 0.35** | 0.50 ± 0.50 | 0.50 ± 0.50 | 0.00 ± 0.00 | +16.67% ↑ |
| **Overall** | Base | 0.42 ± 0.01 | 0.85 ± 0.01 | 0.35 ± 0.01 | 0.09 ± 0.01 | |
| *(N = 2816)* | Tables2Traces | **0.45 ± 0.01** | **0.88 ± 0.01** | **0.38 ± 0.01** | **0.10 ± 0.01** | **+7.49% ↑** |

## E  MEDMCQA CATEGORY-LEVEL RESULTS

To further evaluate generalization, we analyze performance across medical specialties on the MedM-CQA benchmark. As shown in Table 3, Tables2Traces improves performance across 17 of 21 categories, despite being fine-tuned exclusively on tabular data from a single clinical domain (cardiovascular). Notable gains appear in ENT (+16.56%), Social & Preventive Medicine (+16.67%), and Medicine (+12.71%), among others. While a few categories see drops (e.g., Skin, Orthopaedics), the overall gain is +7.49%. These results demonstrate that contrastive supervision derived from structured data can support generalization even to out-of-domain medical topics.

Table 4: Distribution of question types in `MedQA` and `MedMCQA` using LLM-based classification. Values are shown as raw counts and percentages of each dataset.

| Benchmark | Patient-Specific | Abstract | Cardiovascular | Non-Cardiovascular |
|---|---|---|---|---|
| MedQA | 1175 (92.3%) | 98 (7.7%) | 130 (10.2%) | 1143 (89.8%) |
| MedMCQA | 460 (16.3%) | 2356 (83.7%) | 226 (8.0%) | 2590 (92.0%) |

## F  QUESTION TYPE DISTRIBUTIONS

To better understand the nature of the questions in each benchmark, we classify them along two axes using an LLM-based approach: whether a question is *patient-specific* (referring to a concrete clinical case) or *abstract* (testing general medical knowledge), and whether it falls within the *cardiovascular* domain. As shown in Table 4, MedQA is overwhelmingly patient-specific (92.3%) and contains a small cardiovascular subset (10.2%). In contrast, MedMCQA is largely abstract (83.7%) and similarly skewed toward non-cardiovascular questions. This highlights the generalization challenge: our fine-tuned models, trained only on cardiovascular tabular data, are evaluated on questions that are mostly out-of-domain and structurally distinct.

Table 5: Aloe fine-tuning with Tables2Traces supervision. Means and standard error estimates over 10 inference runs.

| Model | Avg Accuracy |
|---|---|
| Aloe | $0.58 \pm 0.01$ |
| Aloe + Tables2Traces | $0.56 \pm 0.01$ |

## G   ALOE FINE-TUNING RESULTS

Aloe is a strong medical QA system trained on many curated datasets with synthetic chain-of-thought, guideline-based answers, and adversarial supervision. It is optimized for direct question answering rather than multi-step or counterfactual reasoning. We include Aloe as a point of contrast and test alignment: does reasoning supervision from Tables2Traces improve a QA-oriented model? We fine-tuned Aloe on the same Tables2Traces prompt–trace pairs and evaluated under identical test-time prompts and decoding settings as in the main experiments. Average accuracy decreases from 0.58 to 0.56 with the same standard error, indicating no benefit from reasoning-based supervision. This supports the claim that Tables2Traces is orthogonal to expensive QA curation and that QA-specific training is misaligned with reasoning traces.

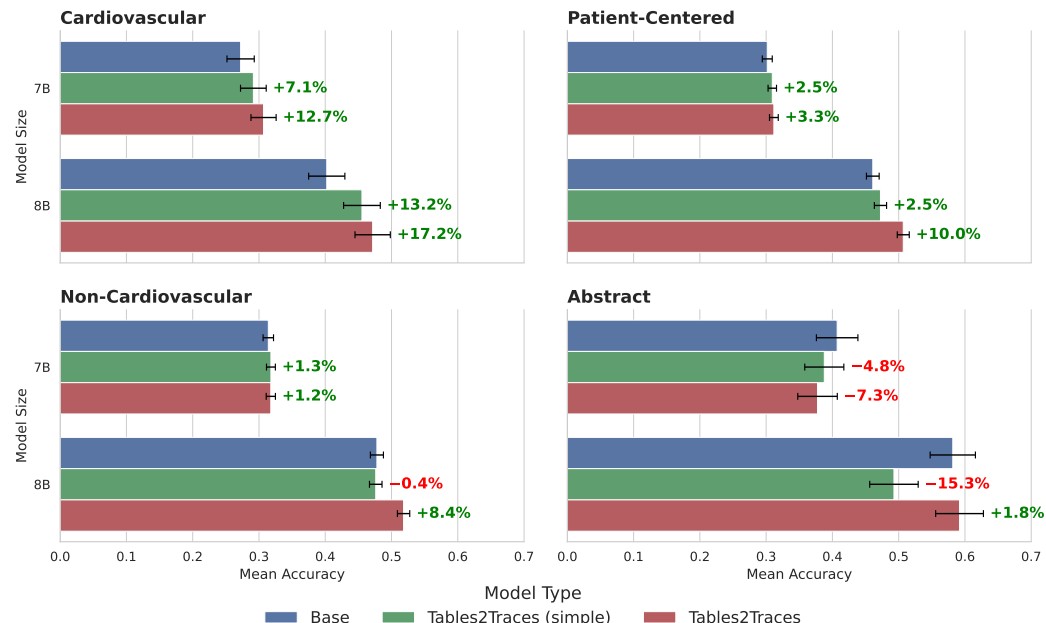

Figure 4: Accuracy on different question types in the MedQA benchmark across model sizes (7B and 8B) and fine-tuning methods. Tables2Traces yields large gains on cardiovascular and more modest gains patient-specific questions. On both cardiovascular and patient-specific questions both 7B and 8B models show consistent improvement. Minor gains are observed for non-cardiovascular questions except for the 8B Tables2Traces model. On abstract questions, all models underperform compared to the base model, except for the 8B Tables2Traces model. Values reflect relative improvement over the base model, with error bars denoting standard error across inference runs.

## H RESULTS FROM QWEN-7B MODELS

To assess whether the benefits of Tables2Traces generalize across model scales, we replicate our main experiments using Qwen models with 7 billion parameters. These models are evaluated on the same MedQA and MedMCQA benchmarks, using identical training procedures as the 8B counterparts. Unlike the 8B results, however, we observe that Tables2Traces provides less consistent improvements at this smaller scale—particularly on out-of-domain or abstract questions. In some cases, performance even degrades relative to the base model.

It is important to note that this comparison involves both a change in model size (8B → 7B) and architecture (LLaMA → Qwen), so the effects cannot be attributed to scaling alone. These results suggest that both model capacity and architecture may influence the effectiveness of structured, trace-based supervision.

### H.1 MEDQA

Figure 4 shows performance on the MedQA benchmark, stratified by question type and model size (7B vs. 8B). Tables2Traces yields substantial improvements on cardiovascular questions (up to +17.2%) and consistent gains on patient-specific questions, especially at the 8B scale. This suggests that structured reasoning supervision is particularly effective for case-based clinical reasoning tasks.

Performance on non-cardiovascular questions improves only modestly, and the Tables2Traces (simple) variant offers little benefit over the base model. For abstract questions, all 7B models underperform, and only Tables2Traces 8B retains accuracy. These results highlight the importance of contrastive, trace-based supervision for enabling models to generalize beyond narrowly defined training inputs.

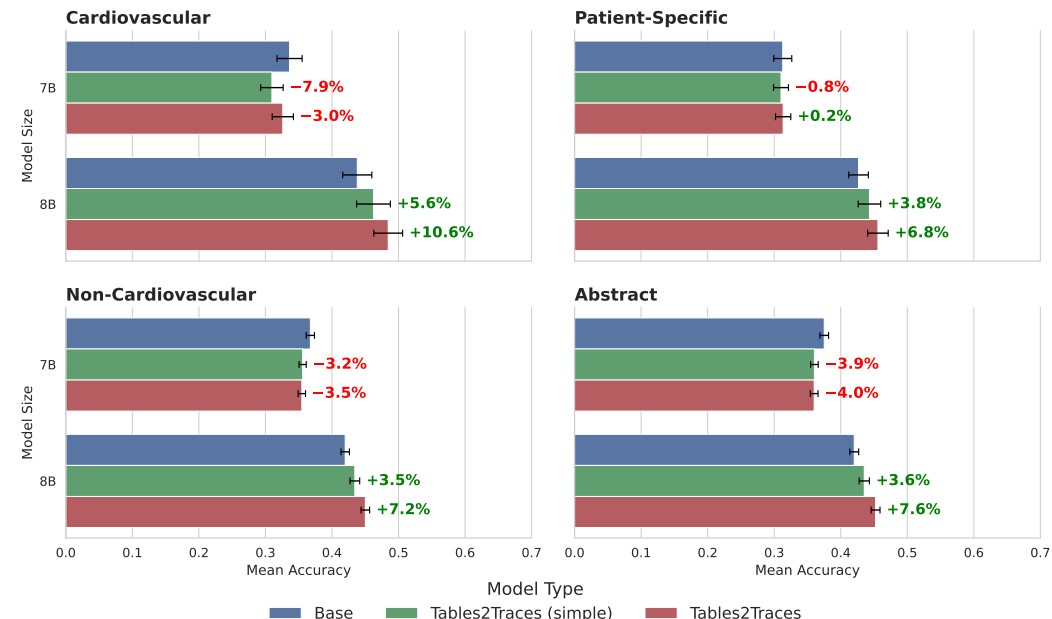

Figure 5: Accuracy on different question types in the MedMCQA benchmark across model sizes (7B and 8B) and fine-tuning methods. At 8B, Tables2Traces improves performance on all question types, including abstract and non-cardiovascular questions. In contrast, 7B models show inconsistent or negative gains, particularly for out-of-domain categories. These results suggest that contrastive supervision derived from tabular data is more effective at scale, and can generalize beyond the source domain when model capacity is sufficient. Values show relative accuracy improvements over the base model, with error bars denoting standard error across inference runs.

## H.2 MEDMCQA

Figure 5 shows model performance on the MedMCQA benchmark, stratified by question type and model size. Tables2Traces yields gains at the 8B scale, improving accuracy on cardiovascular, abstract, and non-cardiovascular questions. Relative gains reach +10.6% on cardiovascular questions and +7.6% on abstract ones.

At the 7B scale, results are more mixed. Both Tables2Traces and Tables2Traces (simple) underperform the base model on most question types, suggesting that smaller models struggle to benefit from structured supervision alone. These findings reinforce the idea that contrastive, trace-based supervision is especially valuable when paired with sufficient model capacity.

Table 6: Per-category evaluation metrics on MedQA for Tables2Traces (simple) and Tables2Traces (8B). % Change refers to change in performance relative to the Base model.

| Category | Model Type | Avg Accuracy | Best-of-n | Majority Vote | Worst-of-n | % Change |
|----------|-----------|--------------|-----------|---------------|------------|----------|
| **Cardiovascular** | Tables2Traces (simple) | $0.46 \pm 0.03$ | $0.86 \pm 0.03$ | $\mathbf{0.42 \pm 0.04}$ | $0.06 \pm 0.02$ | +13.19% ↑ |
| *(N = 130)* | Tables2Traces | $\mathbf{0.47 \pm 0.03}$ | $\mathbf{0.91 \pm 0.03}$ | $0.42 \pm 0.04$ | $0.06 \pm 0.02$ | +17.21% ↑ |
| **Dermatologic** | Tables2Traces (simple) | $\mathbf{0.71 \pm 0.09}$ | $\mathbf{0.94 \pm 0.06}$ | $\mathbf{0.76 \pm 0.11}$ | $\mathbf{0.29 \pm 0.11}$ | +18.81% ↑ |
| *(N = 17)* | Tables2Traces | $0.60 \pm 0.08$ | $0.88 \pm 0.08$ | $0.59 \pm 0.12$ | $0.12 \pm 0.08$ | +0.99% ↑ |
| **Endocrine/Metabolic** | Tables2Traces (simple) | $\mathbf{0.52 \pm 0.02}$ | $\mathbf{0.95 \pm 0.02}$ | $\mathbf{0.47 \pm 0.04}$ | $0.10 \pm 0.02$ | 6.09% ↑ |
| *(N = 179)* | Tables2Traces | $0.51 \pm 0.02$ | $0.91 \pm 0.02$ | $0.46 \pm 0.04$ | $\mathbf{0.10 \pm 0.02}$ | 4.71% ↑ |
| **Gastrointestinal** | Tables2Traces (simple) | $0.44 \pm 0.04$ | $0.88 \pm 0.04$ | $0.38 \pm 0.05$ | $\mathbf{0.09 \pm 0.03}$ | -5.72% ↓ |
| *(N = 86)* | Tables2Traces | $\mathbf{0.50 \pm 0.04}$ | $\mathbf{0.91 \pm 0.03}$ | $\mathbf{0.47 \pm 0.05}$ | $0.08 \pm 0.03$ | +6.72% ↑ |
| **Hematologic** | Tables2Traces (simple) | $0.38 \pm 0.03$ | $0.90 \pm 0.04$ | $0.25 \pm 0.05$ | $0.03 \pm 0.02$ | -6.93% ↓ |
| *(N = 68)* | Tables2Traces | $\mathbf{0.48 \pm 0.04}$ | $\mathbf{0.91 \pm 0.04}$ | $\mathbf{0.43 \pm 0.06}$ | $\mathbf{0.07 \pm 0.03}$ | +18.98% ↑ |
| **Immunologic** | Tables2Traces (simple) | $0.50 \pm 0.04$ | $0.93 \pm 0.03$ | $0.44 \pm 0.06$ | $0.09 \pm 0.03$ | -2.67% ↓ |
| *(N = 81)* | Tables2Traces | $\mathbf{0.54 \pm 0.04}$ | $\mathbf{0.94 \pm 0.03}$ | $\mathbf{0.46 \pm 0.06}$ | $\mathbf{0.17 \pm 0.04}$ | +6.80% ↑ |
| **Infectious** | Tables2Traces (simple) | $0.46 \pm 0.02$ | $0.92 \pm 0.02$ | $0.40 \pm 0.04$ | $0.06 \pm 0.02$ | -3.44% ↓ |
| *(N = 176)* | Tables2Traces | $\mathbf{0.53 \pm 0.02}$ | $\mathbf{0.94 \pm 0.02}$ | $\mathbf{0.45 \pm 0.04}$ | $\mathbf{0.11 \pm 0.02}$ | +9.73% ↑ |
| **Musculoskeletal** | Tables2Traces (simple) | $0.48 \pm 0.05$ | $0.89 \pm 0.05$ | $\mathbf{0.42 \pm 0.07}$ | $0.04 \pm 0.03$ | -2.71% ↓ |
| *(N = 45)* | Tables2Traces | $\mathbf{0.51 \pm 0.04}$ | $\mathbf{0.96 \pm 0.03}$ | $0.40 \pm 0.07$ | $\mathbf{0.07 \pm 0.04}$ | +4.07% ↑ |
| **Neurological** | Tables2Traces (simple) | $0.47 \pm 0.04$ | $0.86 \pm 0.04$ | $0.42 \pm 0.06$ | $\mathbf{0.09 \pm 0.03}$ | +6.89% ↑ |
| *(N = 77)* | Tables2Traces | $\mathbf{0.50 \pm 0.04}$ | $\mathbf{0.90 \pm 0.04}$ | $\mathbf{0.43 \pm 0.06}$ | $0.05 \pm 0.02$ | +15.15% ↑ |
| **Obstetrics/Gynecology** | Tables2Traces (simple) | $0.46 \pm 0.04$ | $0.93 \pm 0.03$ | $\mathbf{0.43 \pm 0.06}$ | $\mathbf{0.07 \pm 0.03}$ | +0.93% ↑ |
| *(N = 70)* | Tables2Traces | $\mathbf{0.47 \pm 0.03}$ | $\mathbf{0.94 \pm 0.03}$ | $0.40 \pm 0.06$ | $0.03 \pm 0.02$ | +2.80% ↑ |
| **Oncology** | Tables2Traces (simple) | $0.50 \pm 0.04$ | $0.86 \pm 0.04$ | $0.47 \pm 0.06$ | $\mathbf{0.15 \pm 0.04}$ | -4.76% ↓ |
| *(N = 72)* | Tables2Traces | $\mathbf{0.56 \pm 0.04}$ | $\mathbf{0.93 \pm 0.03}$ | $\mathbf{0.53 \pm 0.06}$ | $0.14 \pm 0.04$ | +5.82% ↑ |
| **Other** | Tables2Traces (simple) | $0.45 \pm 0.06$ | $0.84 \pm 0.07$ | $\mathbf{0.45 \pm 0.09}$ | $0.10 \pm 0.05$ | -15.24% ↓ |
| *(N = 31)* | Tables2Traces | $\mathbf{0.50 \pm 0.07}$ | $\mathbf{0.87 \pm 0.06}$ | $0.42 \pm 0.09$ | $\mathbf{0.19 \pm 0.07}$ | -4.88% ↓ |
| **Pediatric** | Tables2Traces (simple) | $0.32 \pm 0.07$ | $0.92 \pm 0.08$ | $0.23 \pm 0.12$ | $\mathbf{0.00 \pm 0.00}$ | -19.61% ↓ |
| *(N = 13)* | Tables2Traces | $\mathbf{0.39 \pm 0.05}$ | $\mathbf{1.00 \pm 0.00}$ | $\mathbf{0.31 \pm 0.13}$ | $\mathbf{0.00 \pm 0.00}$ | -1.96% ↓ |
| **Psychiatric** | Tables2Traces (simple) | $0.60 \pm 0.05$ | $\mathbf{0.94 \pm 0.03}$ | $0.58 \pm 0.07$ | $\mathbf{0.21 \pm 0.06}$ | +2.30% ↑ |
| *(N = 52)* | Tables2Traces | $\mathbf{0.62 \pm 0.05}$ | $0.90 \pm 0.04$ | $\mathbf{0.61 \pm 0.07}$ | $\mathbf{0.21 \pm 0.06}$ | +5.57% ↑ |
| **Renal/Genitourinary** | Tables2Traces (simple) | $0.42 \pm 0.04$ | $0.93 \pm 0.04$ | $0.35 \pm 0.07$ | $0.06 \pm 0.03$ | +15.08% ↑ |
| *(N = 54)* | Tables2Traces | $\mathbf{0.48 \pm 0.04}$ | $\mathbf{0.96 \pm 0.03}$ | $\mathbf{0.41 \pm 0.07}$ | $\mathbf{0.09 \pm 0.04}$ | +29.65% ↑ |
| **Respiratory** | Tables2Traces (simple) | $0.49 \pm 0.04$ | $0.93 \pm 0.04$ | $0.44 \pm 0.07$ | $\mathbf{0.11 \pm 0.04}$ | +0.76% ↑ |
| *(N = 54)* | Tables2Traces | $\mathbf{0.50 \pm 0.04}$ | $\mathbf{0.94 \pm 0.03}$ | $\mathbf{0.46 \pm 0.07}$ | $\mathbf{0.11 \pm 0.04}$ | +2.28% ↑ |
| **Toxicology** | Tables2Traces (simple) | $0.41 \pm 0.04$ | $\mathbf{0.93 \pm 0.03}$ | $0.35 \pm 0.06$ | $0.03 \pm 0.02$ | -6.10% ↓ |
| *(N = 68)* | Tables2Traces | $\mathbf{0.52 \pm 0.04}$ | $0.91 \pm 0.04$ | $\mathbf{0.47 \pm 0.06}$ | $\mathbf{0.09 \pm 0.04}$ | +20.68% ↑ |
| **Overall** | Tables2Traces (simple) | $0.47 \pm 0.01$ | $0.91 \pm 0.01$ | $0.42 \pm 0.01$ | $0.08 \pm 0.01$ | +0.82% ↑ |
| *(N = 1273)* | Tables2Traces | $\mathbf{0.51 \pm 0.01}$ | $\mathbf{0.93 \pm 0.01}$ | $\mathbf{0.46 \pm 0.01}$ | $\mathbf{0.10 \pm 0.01}$ | +9.19% ↑ |

## I PER-CATEGORY RESULTS FROM TABLES2TRACES (SIMPLE)

Table 6 reports category-level results for both Tables2Traces and its ablated variant, Tables2Traces (simple), on the MedQA benchmark. Across most categories, the full method consistently outperforms the simple variant, highlighting the added value of contrastive and counterfactual reasoning supervision. However, the simple variant still delivers strong gains over the base model in several categories, including Cardiovascular (+13.19%), Renal/Genitourinary (+15.08%) and Neurological (+6.89%). This table complements the main figures by providing a more granular view of how each model variant performs across medical specialties.

Table 7: Per-category evaluation metrics on MedMCQA for Base and Aloe (8B).

| Category | Model Type | Avg Accuracy | Best-of-n | Majority Vote | Worst-of-n | % Change |
|---|---|---|---|---|---|---|
| **Anaesthesia** | Base | $0.36 \pm 0.07$ | $\mathbf{0.88 \pm 0.07}$ | $0.29 \pm 0.09$ | $\mathbf{0.08 \pm 0.06}$ | |
| *(N = 24)* | Aloe | $\mathbf{0.46 \pm 0.07}$ | $0.79 \pm 0.08$ | $\mathbf{0.46 \pm 0.10}$ | $\mathbf{0.08 \pm 0.06}$ | +26.44%↑ |
| **Anatomy** | Base | $0.36 \pm 0.02$ | $\mathbf{0.86 \pm 0.03}$ | $0.26 \pm 0.04$ | $0.02 \pm 0.01$ | |
| *(N = 147)* | Aloe | $\mathbf{0.49 \pm 0.03}$ | $\mathbf{0.86 \pm 0.03}$ | $\mathbf{0.42 \pm 0.04}$ | $\mathbf{0.14 \pm 0.03}$ | +37.05%↑ |
| **Biochemistry** | Base | $0.57 \pm 0.03$ | $0.90 \pm 0.03$ | $0.58 \pm 0.04$ | $0.11 \pm 0.03$ | |
| *(N = 122)* | Aloe | $\mathbf{0.69 \pm 0.03}$ | $\mathbf{0.93 \pm 0.02}$ | $\mathbf{0.66 \pm 0.04}$ | $\mathbf{0.38 \pm 0.04}$ | +21.59%↑ |
| **Dental** | Base | $0.35 \pm 0.01$ | $0.82 \pm 0.01$ | $0.26 \pm 0.02$ | $0.05 \pm 0.01$ | |
| *(N = 845)* | Aloe | $\mathbf{0.41 \pm 0.01}$ | $\mathbf{0.84 \pm 0.01}$ | $\mathbf{0.34 \pm 0.02}$ | $\mathbf{0.11 \pm 0.01}$ | +15.61%↑ |
| **ENT** | Base | $0.39 \pm 0.05$ | $\mathbf{0.92 \pm 0.04}$ | $0.26 \pm 0.07$ | $0.08 \pm 0.04$ | |
| *(N = 39)* | Aloe | $\mathbf{0.55 \pm 0.06}$ | $0.90 \pm 0.05$ | $\mathbf{0.51 \pm 0.08}$ | $\mathbf{0.23 \pm 0.07}$ | +41.06%↑ |
| **Forensic Medicine** | Base | $0.41 \pm 0.05$ | $\mathbf{0.89 \pm 0.05}$ | $0.32 \pm 0.07$ | $0.09 \pm 0.04$ | |
| *(N = 44)* | Aloe | $\mathbf{0.50 \pm 0.05}$ | $\mathbf{0.89 \pm 0.05}$ | $\mathbf{0.43 \pm 0.08}$ | $\mathbf{0.20 \pm 0.06}$ | +20.88%↑ |
| **Gynaecology & Obstetrics** | Base | $0.40 \pm 0.03$ | $0.81 \pm 0.03$ | $0.32 \pm 0.04$ | $0.09 \pm 0.02$ | |
| *(N = 154)* | Aloe | $\mathbf{0.53 \pm 0.03}$ | $\mathbf{0.89 \pm 0.03}$ | $\mathbf{0.46 \pm 0.04}$ | $\mathbf{0.21 \pm 0.03}$ | +30.43%↑ |
| **Medicine** | Base | $0.44 \pm 0.03$ | $0.84 \pm 0.03$ | $0.39 \pm 0.04$ | $0.12 \pm 0.02$ | |
| *(N = 185)* | Aloe | $\mathbf{0.58 \pm 0.03}$ | $\mathbf{0.90 \pm 0.02}$ | $\mathbf{0.54 \pm 0.04}$ | $\mathbf{0.26 \pm 0.03}$ | +29.93%↑ |
| **Microbiology** | Base | $0.45 \pm 0.04$ | $0.84 \pm 0.04$ | $0.35 \pm 0.06$ | $0.11 \pm 0.04$ | |
| *(N = 74)* | Aloe | $\mathbf{0.58 \pm 0.04}$ | $\mathbf{0.89 \pm 0.04}$ | $\mathbf{0.55 \pm 0.06}$ | $\mathbf{0.24 \pm 0.05}$ | +29.31%↑ |
| **Ophthalmology** | Base | $0.40 \pm 0.05$ | $0.91 \pm 0.04$ | $0.30 \pm 0.07$ | $0.14 \pm 0.05$ | |
| *(N = 43)* | Aloe | $\mathbf{0.54 \pm 0.06}$ | $\mathbf{0.93 \pm 0.04}$ | $\mathbf{0.49 \pm 0.08}$ | $\mathbf{0.23 \pm 0.07}$ | +34.48%↑ |
| **Orthopaedics** | Base | $0.40 \pm 0.08$ | $0.87 \pm 0.09$ | $0.53 \pm 0.13$ | $0.00 \pm 0.00$ | |
| *(N = 15)* | Aloe | $\mathbf{0.59 \pm 0.08}$ | $\mathbf{0.93 \pm 0.07}$ | $\mathbf{0.60 \pm 0.13}$ | $\mathbf{0.13 \pm 0.09}$ | +46.67%↑ |
| **Pathology** | Base | $0.51 \pm 0.02$ | $0.89 \pm 0.02$ | $0.44 \pm 0.03$ | $0.11 \pm 0.02$ | |
| *(N = 259)* | Aloe | $\mathbf{0.65 \pm 0.02}$ | $\mathbf{0.91 \pm 0.02}$ | $\mathbf{0.64 \pm 0.03}$ | $\mathbf{0.32 \pm 0.03}$ | +27.73%↑ |
| **Pediatrics** | Base | $0.44 \pm 0.03$ | $0.82 \pm 0.03$ | $0.39 \pm 0.04$ | $0.09 \pm 0.02$ | |
| *(N = 133)* | Aloe | $\mathbf{0.57 \pm 0.03}$ | $\mathbf{0.90 \pm 0.03}$ | $\mathbf{0.51 \pm 0.04}$ | $\mathbf{0.19 \pm 0.03}$ | +31.05%↑ |
| **Pharmacology** | Base | $0.52 \pm 0.03$ | $0.90 \pm 0.02$ | $0.46 \pm 0.04$ | $0.17 \pm 0.03$ | |
| *(N = 179)* | Aloe | $\mathbf{0.72 \pm 0.03}$ | $\mathbf{0.93 \pm 0.02}$ | $\mathbf{0.69 \pm 0.03}$ | $\mathbf{0.43 \pm 0.04}$ | +38.04%↑ |
| **Physiology** | Base | $0.46 \pm 0.03$ | $0.86 \pm 0.03$ | $0.38 \pm 0.04$ | $0.16 \pm 0.03$ | |
| *(N = 133)* | Aloe | $\mathbf{0.60 \pm 0.03}$ | $\mathbf{0.89 \pm 0.03}$ | $\mathbf{0.58 \pm 0.04}$ | $\mathbf{0.29 \pm 0.04}$ | +31.2%↑ |
| **Psychiatry** | Base | $0.41 \pm 0.10$ | $0.80 \pm 0.13$ | $0.30 \pm 0.15$ | $0.00 \pm 0.00$ | |
| *(N = 10)* | Aloe | $\mathbf{0.60 \pm 0.13}$ | $\mathbf{0.90 \pm 0.10}$ | $\mathbf{0.60 \pm 0.16}$ | $\mathbf{0.30 \pm 0.15}$ | +46.34%↑ |
| **Radiology** | Base | $0.49 \pm 0.04$ | $\mathbf{0.93 \pm 0.03}$ | $0.40 \pm 0.07$ | $0.05 \pm 0.03$ | |
| *(N = 57)* | Aloe | $\mathbf{0.51 \pm 0.05}$ | $0.89 \pm 0.04$ | $\mathbf{0.44 \pm 0.07}$ | $\mathbf{0.14 \pm 0.05}$ | +2.85%↑ |
| **Skin** | Base | $0.37 \pm 0.08$ | $\mathbf{0.91 \pm 0.09}$ | $0.27 \pm 0.14$ | $0.00 \pm 0.00$ | |
| *(N = 11)* | Aloe | $\mathbf{0.47 \pm 0.12}$ | $0.73 \pm 0.14$ | $\mathbf{0.36 \pm 0.15}$ | $\mathbf{0.18 \pm 0.12}$ | +26.83%↑ |
| **Social & Preventive Medicine** | Base | $0.44 \pm 0.04$ | $0.81 \pm 0.04$ | $0.34 \pm 0.05$ | $0.10 \pm 0.03$ | |
| *(N = 91)* | Aloe | $\mathbf{0.52 \pm 0.04}$ | $\mathbf{0.88 \pm 0.03}$ | $\mathbf{0.47 \pm 0.05}$ | $\mathbf{0.20 \pm 0.04}$ | +19.14%↑ |
| **Surgery** | Base | $0.41 \pm 0.02$ | $\mathbf{0.86 \pm 0.02}$ | $0.35 \pm 0.03$ | $0.08 \pm 0.02$ | |
| *(N = 249)* | Aloe | $\mathbf{0.51 \pm 0.02}$ | $\mathbf{0.86 \pm 0.02}$ | $\mathbf{0.47 \pm 0.03}$ | $\mathbf{0.17 \pm 0.02}$ | +24.24%↑ |
| **Unknown** | Base | $\mathbf{0.30 \pm 0.30}$ | $\mathbf{0.50 \pm 0.50}$ | $\mathbf{0.50 \pm 0.50}$ | $\mathbf{0.00 \pm 0.00}$ | |
| *(N = 2)* | Aloe | $0.20 \pm 0.20$ | $\mathbf{0.50 \pm 0.50}$ | $0.00 \pm 0.00$ | $\mathbf{0.00 \pm 0.00}$ | -33.33%↓ |
| **Overall** | Base | $0.42 \pm 0.01$ | $0.85 \pm 0.01$ | $0.35 \pm 0.01$ | $0.09 \pm 0.01$ | |
| *(N = 2816)* | Aloe | $\mathbf{0.53 \pm 0.01}$ | $\mathbf{0.88 \pm 0.01}$ | $\mathbf{0.48 \pm 0.01}$ | $\mathbf{0.21 \pm 0.01}$ | +25.31%↑ |

## J  PER-CATEGORY RESULTS FROM ALOE

For completeness, we report a category-level breakdown of Aloe's performance on the MedMCQA benchmark in Table 7. Aloe achieves consistent improvements over the base model across nearly all medical specialties, with an overall relative gain of +25.31%. Gains are especially large in domains such as Psychiatry (+46.34%), Orthopaedics (+46.67%), and Pharmacology (+38.04%). Only one category (Unknown) shows a performance regression, but it notably only contains two questions. These results align with Aloe's strong overall performance and provide additional insight into which specialties benefit most from its QA-style supervision. We note that Aloe is an upper-bound baseline and that our work is best viewed as a complementary approach rather than a competing one.

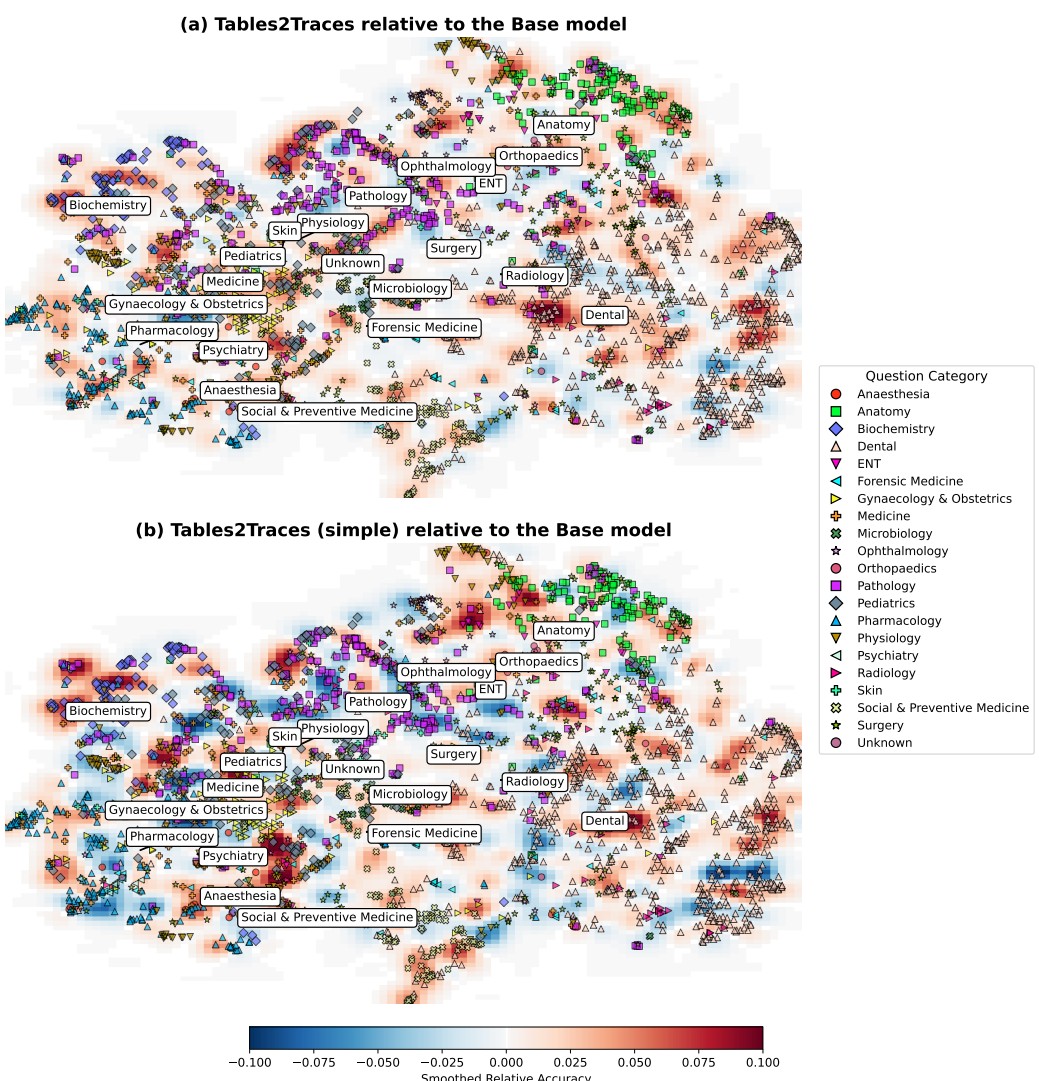

Figure 6: UMAP visualization of MedMCQA test questions, comparing model performance to the **Base** model. Each point represents a question, embedded using `text-embedding-3-large` (OpenAI, 2023), and annotated by medical category using distinct marker shapes and colors. The background heatmap reflects smoothed relative accuracy: red indicates improved performance, blue indicates degradation. **(a)** Tables2Traces shows consistent gains across diverse medical categories. **(b)** Tables2Traces (simple) displays more variable patterns, with several regions showing decreased performance. Cluster labels indicate category centroids.

## K  UMAP VISUALIZATION OF MEDMCQA

Figure 6 shows a UMAP projection of MedMCQA test questions, colored by medical category and overlaid with performance changes relative to the base model. As in MedQA, Tables2Traces (Figure 6a) shows widespread gains across the space. Notable improvements appear in regions corresponding to Anatomy, Dental, and Pathology.

In contrast, Tables2Traces (simple) (Figure 6b) demonstrates a more fragmented pattern. While some clusters benefit (e.g., Dental, Anatomy), others experience performance drops, particularly in Biochemistry and Pharmacology. These results further support the conclusion that structured contrastive supervision is critical for consistent generalization beyond the source domain.

Table 8: Per-category evaluation metrics on the MedQA benchmark for Base and QA-Finetuning (8B).

| Category | Model Type | Avg Accuracy | Best-of-n | Majority Vote | Worst-of-n | % Change |
|---|---|---|---|---|---|---|
| **Cardiovascular** | Base | $0.40 \pm 0.03$ | $0.86 \pm 0.03$ | $0.31 \pm 0.04$ | $0.06 \pm 0.02$ | |
| *(N = 130)* | QA-Finetuning | $0.39 \pm 0.02$ | $\mathbf{0.89 \pm 0.03}$ | $0.30 \pm 0.04$ | $0.03 \pm 0.02$ | -3.63% ↓ |
| **Dermatologic** | Base | $0.59 \pm 0.08$ | $0.94 \pm 0.06$ | $0.53 \pm 0.12$ | $0.06 \pm 0.06$ | |
| *(N = 17)* | QA-Finetuning | $0.54 \pm 0.07$ | $\mathbf{0.94 \pm 0.06}$ | $\mathbf{0.53 \pm 0.12}$ | $\mathbf{0.06 \pm 0.06}$ | -9.9% ↓ |
| **Endocrine/Metabolic** | Base | $0.49 \pm 0.03$ | $0.89 \pm 0.02$ | $0.45 \pm 0.04$ | $0.13 \pm 0.03$ | |
| *(N = 179)* | QA-Finetuning | $0.45 \pm 0.02$ | $\mathbf{0.91 \pm 0.02}$ | $0.40 \pm 0.04$ | $0.06 \pm 0.02$ | -6.55% ↓ |
| **Gastrointestinal** | Base | $0.47 \pm 0.04$ | $0.87 \pm 0.04$ | $0.40 \pm 0.05$ | $0.12 \pm 0.04$ | |
| *(N = 86)* | QA-Finetuning | $0.45 \pm 0.03$ | $\mathbf{0.88 \pm 0.03}$ | $0.37 \pm 0.05$ | $0.07 \pm 0.03$ | -3.73% ↓ |
| **Hematologic** | Base | $0.40 \pm 0.04$ | $0.84 \pm 0.04$ | $0.34 \pm 0.06$ | $0.04 \pm 0.03$ | |
| *(N = 68)* | QA-Finetuning | $\mathbf{0.42 \pm 0.04}$ | $\mathbf{0.87 \pm 0.04}$ | $\mathbf{0.37 \pm 0.06}$ | $\mathbf{0.04 \pm 0.03}$ | +5.11% ↑ |
| **Immunologic** | Base | $0.51 \pm 0.04$ | $0.85 \pm 0.04$ | $0.47 \pm 0.06$ | $0.22 \pm 0.05$ | |
| *(N = 81)* | QA-Finetuning | $0.50 \pm 0.03$ | $\mathbf{0.94 \pm 0.03}$ | $0.43 \pm 0.06$ | $0.10 \pm 0.03$ | -2.43% ↓ |
| **Infectious** | Base | $0.48 \pm 0.03$ | $0.92 \pm 0.02$ | $0.41 \pm 0.04$ | $0.11 \pm 0.02$ | |
| *(N = 176)* | QA-Finetuning | $0.46 \pm 0.02$ | $0.90 \pm 0.02$ | $0.37 \pm 0.04$ | $0.07 \pm 0.02$ | -4.98% ↓ |
| **Musculoskeletal** | Base | $0.49 \pm 0.05$ | $0.89 \pm 0.05$ | $0.49 \pm 0.07$ | $0.04 \pm 0.03$ | |
| *(N = 45)* | QA-Finetuning | $0.45 \pm 0.04$ | $\mathbf{0.98 \pm 0.02}$ | $0.33 \pm 0.07$ | $0.02 \pm 0.02$ | -8.14% ↓ |
| **Neurological** | Base | $0.47 \pm 0.04$ | $0.86 \pm 0.04$ | $0.42 \pm 0.06$ | $0.09 \pm 0.03$ | |
| *(N = 77)* | QA-Finetuning | $\mathbf{0.51 \pm 0.03}$ | $\mathbf{0.96 \pm 0.02}$ | $\mathbf{0.44 \pm 0.06}$ | $0.06 \pm 0.03$ | +8.26% ↑ |
| **Obstetrics/Gynecology** | Base | $0.46 \pm 0.04$ | $0.90 \pm 0.04$ | $0.39 \pm 0.06$ | $0.09 \pm 0.03$ | |
| *(N = 70)* | QA-Finetuning | $0.45 \pm 0.03$ | $\mathbf{0.91 \pm 0.03}$ | $\mathbf{0.39 \pm 0.06}$ | $0.03 \pm 0.02$ | -1.86% ↓ |
| **Oncology** | Base | $0.53 \pm 0.04$ | $0.92 \pm 0.03$ | $0.47 \pm 0.06$ | $0.11 \pm 0.04$ | |
| *(N = 72)* | QA-Finetuning | $0.46 \pm 0.04$ | $0.90 \pm 0.04$ | $0.44 \pm 0.06$ | $0.06 \pm 0.03$ | -13.23% ↓ |
| **Other** | Base | $0.53 \pm 0.07$ | $0.77 \pm 0.08$ | $0.45 \pm 0.09$ | $0.23 \pm 0.08$ | |
| *(N = 31)* | QA-Finetuning | $0.48 \pm 0.06$ | $\mathbf{0.90 \pm 0.05}$ | $\mathbf{0.45 \pm 0.09}$ | $0.16 \pm 0.07$ | -8.54% ↓ |
| **Pediatric** | Base | $0.39 \pm 0.09$ | $0.77 \pm 0.12$ | $\mathbf{0.39 \pm 0.14}$ | $0.00 \pm 0.00$ | |
| *(N = 13)* | QA-Finetuning | $\mathbf{0.43 \pm 0.09}$ | $\mathbf{0.85 \pm 0.10}$ | $0.38 \pm 0.14$ | $\mathbf{0.08 \pm 0.08}$ | +9.8% ↑ |
| **Psychiatric** | Base | $0.59 \pm 0.05$ | $0.94 \pm 0.03$ | $0.54 \pm 0.07$ | $0.23 \pm 0.06$ | |
| *(N = 52)* | QA-Finetuning | $0.53 \pm 0.04$ | $0.88 \pm 0.04$ | $0.50 \pm 0.07$ | $0.13 \pm 0.05$ | -8.85% ↓ |
| **Renal/Genitourinary** | Base | $0.37 \pm 0.04$ | $0.85 \pm 0.05$ | $0.26 \pm 0.06$ | $0.04 \pm 0.03$ | |
| *(N = 54)* | QA-Finetuning | $\mathbf{0.42 \pm 0.04}$ | $\mathbf{0.93 \pm 0.04}$ | $\mathbf{0.30 \pm 0.06}$ | $\mathbf{0.06 \pm 0.03}$ | +13.57% ↑ |
| **Respiratory** | Base | $0.49 \pm 0.04$ | $\mathbf{0.91 \pm 0.04}$ | $0.43 \pm 0.07$ | $0.09 \pm 0.04$ | |
| *(N = 54)* | QA-Finetuning | $\mathbf{0.50 \pm 0.04}$ | $\mathbf{0.91 \pm 0.04}$ | $\mathbf{0.48 \pm 0.07}$ | $\mathbf{0.09 \pm 0.04}$ | +2.66% ↑ |
| **Toxicology** | Base | $\mathbf{0.43 \pm 0.04}$ | $0.79 \pm 0.05$ | $\mathbf{0.41 \pm 0.06}$ | $0.06 \pm 0.03$ | |
| *(N = 68)* | QA-Finetuning | $\mathbf{0.43 \pm 0.03}$ | $\mathbf{0.93 \pm 0.03}$ | $0.29 \pm 0.06$ | $0.03 \pm 0.02$ | -1.69% ↓ |
| **Overall** | Base | $\mathbf{0.47 \pm 0.01}$ | $0.88 \pm 0.01$ | $\mathbf{0.41 \pm 0.01}$ | $\mathbf{0.11 \pm 0.01}$ | |
| *(N = 1273)* | QA-Finetuning | $0.46 \pm 0.01$ | $\mathbf{0.91 \pm 0.01}$ | $0.39 \pm 0.01$ | $0.06 \pm 0.01$ | -3.17% ↓ |

## L  QA-ONLY ABLATION RESULTS

To assess whether standard QA-format supervision could account for the performance improvements observed in our full method, we conduct an ablation where the model is fine-tuned exclusively on the 10k QA-format examples used in the mixed setup. Importantly, these 10K QA-format examples do not overlap with the questions used for evaluation. Results are shown in Table 8 (MedQA) and Table 9 (MedMCQA).

On MedQA, the QA-only model performs comparably or slightly worse than the base model (0.46 vs. 0.47 average accuracy), with inconsistent effects across clinical categories. On MedMCQA, the QA-only model performs worse than the base model overall (0.40 vs. 0.42 average accuracy) and shows negative or negligible gains across most categories. These results indicate that the QA examples alone do not explain the improvements observed in our main models.

Table 9: Per-category evaluation metrics on MedMCQA for Base and QA-finetuning (8B).

| Category | Model Type | Avg Accuracy | Best-of-n | Majority Vote | Worst-of-n | % Change |
|---|---|---|---|---|---|---|
| **Anaesthesia** | Base | **0.36 ± 0.07** | **0.88 ± 0.07** | **0.29 ± 0.09** | **0.08 ± 0.06** | |
| *(N = 24)* | QA-Finetuning | 0.25 ± 0.04 | 0.83 ± 0.08 | 0.04 ± 0.04 | 0.00 ± 0.00 | -29.89%↓ |
| **Anatomy** | Base | **0.36 ± 0.02** | 0.86 ± 0.03 | **0.26 ± 0.04** | **0.02 ± 0.01** | |
| *(N = 147)* | QA-Finetuning | 0.33 ± 0.02 | **0.89 ± 0.03** | 0.23 ± 0.03 | 0.01 ± 0.01 | -8.70%↓ |
| **Biochemistry** | Base | **0.57 ± 0.03** | **0.90 ± 0.03** | **0.58 ± 0.04** | 0.11 ± 0.03 | |
| *(N = 122)* | QA-Finetuning | 0.51 ± 0.03 | **0.90 ± 0.03** | 0.49 ± 0.05 | **0.12 ± 0.03** | -9.57%↓ |
| **Dental** | Base | **0.35 ± 0.01** | 0.82 ± 0.01 | **0.26 ± 0.02** | **0.05 ± 0.01** | |
| *(N = 845)* | QA-Finetuning | 0.34 ± 0.01 | **0.86 ± 0.01** | 0.23 ± 0.01 | 0.03 ± 0.01 | -4.90%↓ |
| **ENT** | Base | 0.39 ± 0.05 | **0.92 ± 0.04** | 0.26 ± 0.07 | **0.08 ± 0.04** | |
| *(N = 39)* | QA-Finetuning | **0.42 ± 0.05** | 0.90 ± 0.05 | **0.28 ± 0.07** | **0.08 ± 0.04** | +7.28%↑ |
| **Forensic Medicine** | Base | **0.41 ± 0.05** | **0.89 ± 0.05** | **0.32 ± 0.07** | **0.09 ± 0.04** | |
| *(N = 44)* | QA-Finetuning | 0.36 ± 0.05 | 0.86 ± 0.05 | 0.25 ± 0.07 | 0.07 ± 0.04 | -12.64%↓ |
| **Gynaecology & Obstetrics** | Base | **0.40 ± 0.03** | 0.81 ± 0.03 | **0.32 ± 0.04** | **0.09 ± 0.02** | |
| *(N = 154)* | QA-Finetuning | 0.37 ± 0.02 | **0.86 ± 0.03** | 0.31 ± 0.04 | 0.03 ± 0.01 | -7.41%↓ |
| **Medicine** | Base | **0.44 ± 0.03** | 0.84 ± 0.03 | **0.39 ± 0.04** | **0.12 ± 0.02** | |
| *(N = 185)* | QA-Finetuning | **0.44 ± 0.02** | **0.87 ± 0.02** | 0.36 ± 0.04 | 0.09 ± 0.02 | -0.73%↓ |
| **Microbiology** | Base | **0.45 ± 0.04** | 0.84 ± 0.04 | 0.35 ± 0.06 | **0.11 ± 0.04** | |
| *(N = 74)* | QA-Finetuning | 0.42 ± 0.04 | **0.86 ± 0.04** | **0.36 ± 0.06** | 0.04 ± 0.02 | -5.74%↓ |
| **Ophthalmology** | Base | 0.40 ± 0.05 | 0.91 ± 0.04 | 0.30 ± 0.07 | **0.14 ± 0.05** | |
| *(N = 43)* | QA-Finetuning | **0.41 ± 0.05** | **0.93 ± 0.04** | **0.33 ± 0.07** | 0.05 ± 0.03 | +0.57%↑ |
| **Orthopaedics** | Base | **0.40 ± 0.08** | **0.87 ± 0.09** | **0.53 ± 0.13** | 0.00 ± 0.00 | |
| *(N = 15)* | QA-Finetuning | 0.38 ± 0.07 | **0.87 ± 0.09** | 0.27 ± 0.12 | **0.07 ± 0.07** | -5.00%↓ |
| **Pathology** | Base | **0.51 ± 0.02** | 0.89 ± 0.02 | **0.44 ± 0.03** | **0.11 ± 0.02** | |
| *(N = 259)* | QA-Finetuning | 0.45 ± 0.02 | **0.90 ± 0.02** | 0.37 ± 0.03 | 0.06 ± 0.01 | -10.92%↓ |
| **Pediatrics** | Base | **0.44 ± 0.03** | 0.82 ± 0.03 | **0.39 ± 0.04** | **0.09 ± 0.02** | |
| *(N = 133)* | QA-Finetuning | **0.44 ± 0.03** | **0.87 ± 0.03** | 0.37 ± 0.04 | 0.05 ± 0.02 | +0.86%↑ |
| **Pharmacology** | Base | **0.52 ± 0.03** | **0.90 ± 0.02** | **0.46 ± 0.04** | **0.17 ± 0.03** | |
| *(N = 179)* | QA-Finetuning | 0.50 ± 0.02 | **0.90 ± 0.02** | 0.44 ± 0.04 | 0.12 ± 0.02 | -3.13%↓ |
| **Physiology** | Base | **0.46 ± 0.03** | 0.86 ± 0.03 | **0.38 ± 0.04** | **0.16 ± 0.03** | |
| *(N = 133)* | QA-Finetuning | 0.45 ± 0.03 | **0.87 ± 0.03** | **0.38 ± 0.04** | 0.12 ± 0.03 | -2.79%↓ |
| **Psychiatry** | Base | 0.41 ± 0.10 | **0.80 ± 0.13** | 0.30 ± 0.15 | 0.00 ± 0.00 | |
| *(N = 10)* | QA-Finetuning | 00.49 ± 0.10 | **0.80 ± 0.13** | **0.50 ± 0.17** | 0.00 ± 0.00 | +19.51%↑ |
| **Radiology** | Base | **0.49 ± 0.04** | **0.93 ± 0.03** | **0.40 ± 0.07** | **0.05 ± 0.03** | |
| *(N = 57)* | QA-Finetuning | 0.45 ± 0.04 | **0.93 ± 0.03** | **0.40 ± 0.07** | 0.04 ± 0.02 | -8.54%↓ |
| **Skin** | Base | 0.37 ± 0.08 | 0.91 ± 0.09 | **0.27 ± 0.14** | 0.00 ± 0.00 | |
| *(N = 11)* | QA-Finetuning | **0.39 ± 0.10** | **1.00 ± 0.00** | **0.27 ± 0.14** | **0.09 ± 0.09** | +4.88%↑ |
| **Social & Preventive Medicine** | Base | **0.44 ± 0.04** | 0.81 ± 0.04 | **0.34 ± 0.05** | **0.10 ± 0.03** | |
| *(N = 91)* | QA-Finetuning | 0.43 ± 0.03 | **0.89 ± 0.03** | 0.32 ± 0.05 | 0.08 ± 0.03 | -2.52%↓ |
| **Surgery** | Base | **0.41 ± 0.02** | 0.86 ± 0.02 | **0.35 ± 0.03** | **0.08 ± 0.02** | |
| *(N = 249)* | QA-Finetuning | 0.40 ± 0.02 | **0.88 ± 0.02** | 0.31 ± 0.03 | 0.05 ± 0.01 | -2.93%↓ |
| **Unknown** | Base | 0.30 ± 0.30 | 0.50 ± 0.50 | **0.50 ± 0.50** | 0.00 ± 0.00 | |
| *(N = 2)* | QA-Finetuning | **0.45 ± 0.25** | **1.00 ± 0.00** | **0.50 ± 0.50** | 0.00 ± 0.00 | +50.00%↑ |
| **Overall** | Base | **0.42 ± 0.01** | 0.85 ± 0.01 | **0.35 ± 0.01** | **0.09 ± 0.01** | |
| *(N = 2816)* | QA-Finetuning | 0.40 ± 0.01 | **0.88 ± 0.01** | 0.31 ± 0.01 | 0.06 ± 0.00 | -5.10%↓ |

## M  DISTANCE METRIC CHOICE FOR CONTRASTIVE NEIGHBOR SELECTION

**Rationale.** We use the Gower distance because it is data-type agnostic and compares heterogeneous features (numeric, binary, categorical) without domain-specific encodings. It provides a simple, interpretable default for mixed clinical tables.

**Alternatives.** The pipeline is metric-agnostic. In principle, other choices can be substituted in the neighbor retrieval step, for example: (i) scaled Euclidean on normalized numeric features with one-hot categories, (ii) Hamming distance for categorical-only subsets, (iii) cosine distance on serialized or embedded representations, or (iv) learned metrics (e.g., Mahalanobis) if one wishes to tune feature weights.

**Scope.** A full comparison of distance functions is outside the scope of this paper. We adopt Gower as a simple and effective default for mixed-type data, and future work could explore learned or task-specific metrics.

## N   CLINICIAN EVALUATION PROTOCOL AND RUBRIC

**Protocol.**   We randomly sampled 10 supervision traces from the training corpus. Two independent clinicians with cardiology expertise reviewed the same set, each completing a structured rubric for every trace without seeing the other's responses. Cases contained only de-identified, synthesized patient descriptions derived from tabular rows (anchor and neighbors). The clinicians were asked to rate each trace along five dimensions and optionally add a one-line comment. We report the normalized tallies in Tables 10–11.

**Rubric (per trace).**   Each trace was rated on the following dimensions with the indicated discrete scale.

1. **Overall clinical plausibility:** *Yes / Partially / No.*
2. **Unsafe or inappropriate recommendations:** *None / Minor / Concerning.*
3. **Appropriate weighting of key factors:** *Yes / Partially / No.*
4. **Comparative reasoning quality (why target vs. neighbor):** *Clear / Partial / Superficial.*
5. **Uncertainty expression:** *Understated / Appropriate / Overstated.*
6. **One-line comment (optional):** free-text note (e.g., phrasing, missing considerations).

**Guidance provided to raters.**

- *Plausibility* asks whether the narrative could reasonably reflect clinical reasoning given only the provided variables.
- *Unsafe/inappropriate* flags any recommendation that would be clinically unsafe or clearly inappropriate in context; "Minor" covers low-risk or borderline phrasing.
- *Weighting* assesses whether major risk factors are emphasized appropriately relative to minor ones.
- *Comparative reasoning* evaluates whether differences between target and neighbors are identified and used to justify outcomes.
- *Uncertainty* evaluates acknowledgment of limits of the available variables (avoid overconfidence or implying hidden labels).

**Limitations.**   This review is qualitative and small-scale ($n{=}10$ traces), with no rater training or adjudication; results should be interpreted as a plausibility/safety check for *research-only* supervision rather than clinical validation or calibrated risk assessment. Importantly, we also note the high disagreement between the two clinicians.

Table 10: Clinician (A): tally of ratings across 10 traces.

|  | Positive | Partial / Minor | Negative |
|---|---|---|---|
| Plausibility | **5 (Yes)** | 4 (Partially) | 1 (No) |
| Unsafe / inappropriate | **7 (None)** | 3 (Minor) | 0 (Concerning) |
| Weighting | **5 (Yes)** | 4 (Partially) | 1 (No) |
| Reasoning | **5 (Clear)** | 4 (Partial) | 1 (Superficial) |
| Uncertainty | **5 (Appropriate)** | – | 5 (Overstated) |

Table 11: Clinician (B): tally of ratings across 10 traces.

|  | Positive | Partial / Minor | Negative |
|---|---|---|---|
| Plausibility | 1 (Yes) | **9 (Partially)** | 0 (No) |
| Unsafe / inappropriate | 4 (None) | **6 (Minor)** | 0 (Concerning) |
| Weighting | 0 (Yes) | **8 (Partially)** | 2 (No) |
| Reasoning | 0 (Clear) | 4 (Partial) | **6 (Superficial)** |
| Uncertainty | 0 (Appropriate) | – | **10 (Overstated)** |

## O    CLINICIAN VALIDATION OF SUPERVISION TRACES

Out of the 10 randomly sampled traces, **no trace received a "Concerning" safety rating, and only one traces was judged to be implausible by Clinician (A).** In addition to these discrete measures, we also include a table containing the qualitative feedback from Clinician (B) on 10 randomly sampled traces. Clinician (A) only had one comment, stating that "CRP should be acted on. Recommend finding the cause of CRP 45, like cancer". We therefore only include a table for Clinician (B), who had additional comments for all traces. Tags indicate recurring themes; comments are lightly abridged for brevity.

Table 12: Clinician (B) qualitative review of 10 traces.

| Patient | Issue tags | Clinician comment (abridged) |
|---|---|---|
| 1 | Vague phrasing; overconfidence | "BP 'way above' is not clinical phrasing—use concrete categories (e.g., stage 2 hypertension). Consider guideline scores (e.g., CHADS$_2$-VASc)." |
| 2 | Overstates intervention benefit; weighting | "Hyperlipidemia not that serious for a non-smoking woman without diabetes, even with grade 2 hypertension." |
| 3 | Speculative; conflicting factors; circular counterfactual | "Acknowledge conflict between anthropometric and biochemical factors; counterfactual goes in circles." |
| 4 | Partial weighting; overconfidence | "Reasoning partial; certainty overstated given available variables." |
| 5 | Baseline risk omitted | "Age not addressed correctly—baseline mortality risk." |
| 6 | Counterfactual focus misaligned | "BG change not the primary modifiable variable in this context; reasoning superficial." |
| 7 | Overconfidence; superficial | "Reasoning superficial; certainty overstated." |
| 8 | Misinterpretation of CRP | "Elevated CRP may reflect infection—don't treat as CVD risk alone." |
| 9 | Lab inconsistency note | "Glucose/HbA1c discrepancy is often seen (e.g., after a meal)." |
| 10 | Risk calibration; BP control | "CVD risk overstated; emphasize blood-pressure control (cf. risk charts/guidelines)." |

## P LLM USAGE

In accordance with the ICLR 2026 Author Guide, we disclose that large language models (LLMs) were used solely to assist with text polishing and minor formatting during paper preparation. LLMs were not involved in research ideation, experiment design, or result interpretation. All scientific contributions, methods, and conclusions are the work of the authors.

## Q QUALITATIVE REASONING EXAMPLES

To make the model's internal reasoning structure transparent, we present annotated reasoning traces for a selection of representative clinical cases. These qualitative examples illustrate the structured reasoning competencies that our training objective is designed to cultivate.

In this section we include *verbatim reasoning traces* solely for interpretability and qualitative analysis. These traces are not used for training, are not exposed by the deployed model, and serve only to evaluate the reasoning behaviors emerging from our supervision scheme.

To parallel the three components introduced in Section C.3, each reasoning trace is annotated using a color-coded scheme highlighting three qualitatively distinct reasoning modes.

**Color–coding of reasoning modes.** We define three colors corresponding to the three structured reasoning behaviors encouraged by our supervision signal:

- **(1) Clinical Differentiation (Blue)**
  Inspired by the *Differential Reasoning* step in our prompt design, this mode captures the model's ability to identify clinically decisive features, contrast competing diagnoses, and articulate structured differences between plausible etiologies. It answers the question: *"Which competing explanations matter here, and how do they differ?"*

- **(2) Plausibility & Consistency Checking (Green)**
  Analogous to the *Label Plausibility* step, this mode reflects the model's ability to evaluate whether the clinical picture is internally coherent. It is checking whether symptoms, vitals, risks, and pathophysiology align. It answers: *"Does this presentation make sense, and what evidence supports or contradicts it?"*

- **(3) Priority-Based Clinical Action (Red)**
  Reflecting the *Counterfactual / Action Planning* component, this mode highlights how the model resolves the case into a prioritized, causally-grounded clinical action. This is often the highest-urgency, safety-critical intervention recommended. It answers: *"Given this reasoning, what is the one intervention that must occur next?"*

**Selection of examples.** To ensure that the qualitative analysis is both fair and informative, we select questions for which the base model demonstrates non-trivial but unreliable performance (answering correctly at least 2/10 times), while the fine-tuned model answers correctly in a stable majority of

runs ($> 5/10$). This guarantees that the examples reflect genuine, reproducible improvements in reasoning rather than cherry-picked successes or failures.

**Purpose of qualitative examples.** The annotated traces below provide qualitative evidence that the model: (i) performs structured differential comparisons, (ii) assesses internal clinical plausibility, and (iii) selects prioritized interventions grounded in causal medical reasoning. These examples are included exclusively for interpretability and are not representative of the model's inference-time outputs.

We present first the question, then the Base model's reasoning followed by the full Tables2Traces reasoning.

## Q.1 ACUTE HIV COMPLICATION AND EMERGENT AIRWAY COLLAPSE

**Question**

A 55-year-old woman with a history of HIV presents to the emergency department with progressive dyspnea, fever, and dry cough for the past 4 days. She has not been compliant with antiretroviral or prophylactic medication. On evaluation, the patient is in moderate respiratory distress. The temperature is 38.8°C (102.0°F), the blood pressure is 124/82 mm Hg, the pulse is 96/min, and the respiratory rate is 20/min. Pulse oximetry is 92% on 4 L oxygen by nasal cannula. Her CD4 count is 180 cells/$\mu$L. IV antibiotics and glucocorticoids are administered. After 30 minutes, the patient develops severe respiratory distress. Repeat vital signs show: temperature 38.3°C (101.0°F), blood pressure 80/50 mm Hg, pulse 104/min, respiration rate 32/min and pulse oximetry 85% on nasal cannula. The trachea deviates to the left. Breath sounds are absent on the right side and the neck veins are distended. Telemetry shows sinus tachycardia. Which of the following is the best next step in the management of this patient?
A: 'Discontinue antibiotics'
B: 'Perform an arterial blood gas'
C: 'Needle thoracostomy'
D: 'Stat chest X-ray'

---

**Base Reasoning**

Okay, so Im̓ trying to figure out the best next step for this patient. Letś start by going through all the information given.

The patient is a 55-year-old woman with HIV who presented to the emergency department with symptoms of progressive dyspnea, fever, and dry cough for four days. She hasnt́ been taking her antiretroviral or prophylactic medications as prescribed. That makes me think she might have an opportunistic infection because HIV can weaken her immune system.

Her vital signs upon arrival are: temperature at 38.8°C (which is slightly elevated), blood pressure around 124/82 mmHg, pulse is 96 beats per minute, and respiratory rate is upped at 20 breaths per minute. Her oxygen saturation on nasal cannula at 4 L/min is down to 92%. So sheś breathing harder than normal and not getting enough oxygen from just that.

CD4 count of 180 cells/$\mu$L—this suggests her immune suppression might be moderate since CD4 counts below a certain threshold indicate more severe immunodeficiency in HIV patients; maybe she has some level of immunity left but definitely compromised.

She was treated with IV antibiotics and glucocorticoids initially because those are common treatments for infections like pneumonia or something causing an inflammatory response—maybe *Pneumocystis carinii* or another bacterial infection given her history of non-compliance.

But then after half an hour (30 minutes later), things took a turn for the worse: now she's in severe respiratory distress—temperature went down slightly but still febrile? Wait no—it decreased from $\sim 39°C$ to about $\sim 38°C$ which isn't too bad—but BP dropped significantly into hypotensive range (80/50) which indicates shock possibly? Pulse jumped up to 104 bpm—a tachycardia—and respiration rate doubled—to 32—which shows increasing difficulty breathing even more than before—and oxygenation dropped further despite being on higher flow (still only at nasal cannula). Also noted that trachea deviates left—that sounds like thereś fluid buildup on one side making it hard to breathe; probably right-sided issue since tracheal deviation left means mediastinal shift towards left?

Breath sounds absent on right side—if you cant́ hear lung sounds there, it could mean either complete collapse or effusion/tension pneumothorax perhaps? Neck veins distended suggest jugular venous distension—which often points towards heart failure or fluid overload leading to increased intracranial pressure if undiagnosed quickly.

Telemetry shows sinus tachycardia—that's consistent with stress responses like anxiety/fear but also possible electrolyte imbalances such as low potassium/hypokalemia which causes arrhythmias including sinus tachycardia sometimes when K+ levels drop too low—or could be due directly related organ dysfunction affecting heart rhythm control centers in brainstem?

So putting this together: The initial presentation points toward an acute medical condition needing prompt attention—the most likely culprit here seems like Acute Respiratory Distress Syndrome (ARDS) considering pulmonary issues alongside systemic compromise indicated by dropping BP into hypotensive ranges along with rising heart rate indicating circulatory insufficiency—but wait ARDS usually comes from direct lung injury vs indirect causes... Alternatively sepsis maybe?

Alternatively thinking about *Pneumocystis jiroveci* pneumonia—is common in HIV patients when theyŕe not compliant because their immune systems arent́ controlling it well anymore especially if they missed their meds preventing PCP reactivation.

Wait though—in typical cases where someone presents similarly without any other comorbidity risk factors beyond age—they'd workup would include chest X-ray first before moving onto more invasive tests.

But let me think again—the question says "best" next step after these findings post-treatment changes...

> **Base Reasoning (continued)**
>
> In terms of management steps:
> Option A) Discontinue antibiotics—I don't see why stop them immediately unless we know they're ineffective/mutually harmful.
> Option B) Perform ABG—to check actual $O_2$ saturation levels accurately rather than relying on pulse oximeter readings because sometimes pulse oximetry can give false readings especially if skin perfusion poor etc.—ABG gives precise measure. Also useful info whether pH/acid-base status via blood gas.
> Option C) Needle thoracostomy—is used mainly for tension pneumothorax where needle decompression needed rapidly... But here doesń seem immediate need unless confirmed.
> Option D) Stat chest X-ray—to rule out serious conditions like massive collapse, lung opacities etc., hence imaging needed.
> Given all above—what happened after treatment leads us closer—are we missing something else? After starting IV abx + steroids got worse—not better—as opposed expected improvement seen typically wouldń happen so quickly except maybe progression until correct therapy found.
> Maybe underlying cause needs confirmation—with worsening clinical picture pointing towards bacterial etiology requiring different approach.
> Another angle: If CD4 count is >200 usually considered advanced disease state—an indication that full-blown AIDS-related complications should already be managed accordingly... Or does lower CD4 imply less likely ability respond effectively without supportive care?
> Considering time frame—from ED evaluation onward: They gave IV abx + glucocortics hoping treating community-acquired pneumonia/pneumonia caused by *Streptococcus*/*H. influenzae*/*Pseudomonas* etc.; however rapid decline suggests either wrong choice meds OR presence of MDR-TB/HIV-associated tuberculosis/*Legionella*/*Mycoplasma*... Especially TB commonly occurs latently becoming active due lack adherence HAART leading backsliding...
> If suspected TB would require appropriate susceptibility testing/bigger guns—in ED setting assuming availability limited though.
> So what do I do now:
> 1st thought – get imaging done stat via CXR prior proceeding further tests—as ABCDs offer various options;
> Now looking at choices:
> A–D provided. B says perform ABG—important data point but not definitive treatment; D stat CXR—essential diagnostic tool.
> Which between B & D?
> Additionally, could consider doing both—but among options given only choose one option. Assuming question asks best next step regardless previous tx outcome.
> Possibly ruling out life-threatening issues requires seeing CXR first—even if started tx early—youd́ want confirm diagnosis correctly.
> Thus answer D)—stat chest xray—
> However alternative reasoning arises—tracheal deviation implies possible mass lesion pushing diaphragm upwards compressing opposite lung fields thus limiting sound transmission meaning underdiagnosed collapsed lungs → necessitating immediate intervention else risk aspiration/collapse.
> Wait—if breath sounds absent right side implying total collapse/right main stem intubation required? No—heavily depends how critical situation—attempted intubation may complicate matters further...
> Alternative line of thought: Trauma/Bleeding/Tamponade—but unlikely context-wise w/o trauma hx. Pulmonary embolism—risk factors high VTE = age, female, hormones, HIV-associated hypercoagulability potentially...

> **Base Reasoning (continued)**
>
> Hmm.. Maybe go back basics:
>
> The key labs mentioned were Trachea deviates Left → probably Right Main Bronchus obstruction/stenosis/trouble breathing → could also indicate central airway obstruction requiring bronchoscopy/endoscopic procedures BUT again timing concerning urgency during acute episode.
>
> Alternately, mass lesion impinging on airway causing shift ⇒ dyspnea/stridor/wheezing heard over affected area—not sure description said Breath Sounds Absent though—so no wheezes/sounds → suggestive of no air movement → complete collapse/lung deflation → indicative of massive PE?, TPA blockage, malignancy compression, tumour obstructing major bronchi, etc.—all possibilities.
>
> But how urgent is it? Can you manage without knowing precisely while waiting for tests?
>
> Well, Chest X-ray will show opacity or not, pneumoconiosis, bronchiectasis, pneumonia, wedge-shaped opacity... In case of abscess, large bullae etc. Alternatively CT scan offers better detail, but "stat" refers to fastest available method—CXR provides quick insight.
>
> Therefore, the Best Next Step Would Be performing Stat Chest X-Ray to determine underlying pathology definitively guiding subsequent management decisions regarding specific treatment approaches, such as antivirals, antibiotics tailored based on cultures, further specialist referral, intubation if necessary...
>
> Yes, I think Option D) Stat Chest X-ray Is The Best Next Step After Initial Treatment Didn't Yield Improvement And With Worsening Symptoms.
>
> The best next step in managing this patient's condition would be conducting a **stat chest X-ray**. This diagnostic tool will provide crucial information about potential pathological processes affecting the lungs and surrounding areas following initial antibiotic and glucocorticoid administration failing to improve symptoms promptly. Given the clinical presentation— including tracheal deviation indicating possible airway obstruction—the STAT CXR helps differentiate between conditions such as pneumonia, consolidation, masses/tumors compressing airflow pathways versus other critical scenarios requiring immediate intervention like tension pneumothorax or mechanical ventilation considerations once stabilised otherwise.
>
> **Answer: D**

### Tables2Traces Reasoning

The question presents a 55-year-old woman with HIV who has been non-compliant with antiretroviral and prophylactic medications. She presents to the emergency department with progressive dyspnea, fever, and dry cough for the past 4 days. Her vital signs indicate moderate respiratory distress, and she is in severe respiratory distress after receiving IV antibiotics and glucocorticoids. The patient's CD4 count is low at 180 cells/$\mu$L, which suggests that her immune system may be compromised due to poor management of her HIV infection.

Given this information, the differential strongly includes an opportunistic infection such as *Pneumocystis jirovecii* pneumonia (PJP), which is common in patients with advanced HIV or inadequate prophylaxis.

Now let's break down your options:

Option A: Discontinue antibiotics — This option would be inappropriate because there is no evidence of antibiotic resistance or a contraindication to continuing treatment.

Option B: Perform an arterial blood gas — An ABG provides useful information about oxygenation and acid–base status, but would not address the primary, rapidly deteriorating complication suspected here.

Option C: Needle thoracostomy — The unilateral absence of breath sounds together with tracheal deviation is clinically consistent with a tension pneumothorax, a life-threatening complication that can occur during assisted ventilation.

Given these findings, the most urgent and appropriate next step is needle thoracostomy (Option C), as immediate decompression is required to relieve trapped intrapleural air and prevent cardiovascular collapse.

Although a chest X-ray could confirm the diagnosis, waiting for imaging would delay a time-critical, life-saving intervention.

Answer: C

## Q.2 MULTISYSTEM INFECTION WITH ATYPICAL FEATURES

**Question**

A 57-year-old man presents to the emergency department with confusion. His symptoms started a few days ago and have been gradually worsening. Initially, the patient had a fever and a cough which has progressed to abdominal pain, diarrhea, and confusion. His temperature is 102°F (38.9°C), blood pressure is 127/68 mmHg, pulse is 120/min, respirations are 17/min, and oxygen saturation is 94% on room air. Physical exam is notable for localized crackles and decreased breath sounds in the left lower lung field. The patient is confused and is answering questions inappropriately. Laboratory studies are ordered as seen below.
Hemoglobin: 12 g/dL
Hematocrit: 36%
Leukocyte count: 16,500/mm$^3$ with normal differential
Platelet count: 190,000/mm$^3$
Serum:
Na+: 130 mEq/L
Cl-: 100 mEq/L
K+: 4.3 mEq/L
HCO3-: 25 mEq/L
BUN: 20 mg/dL
Glucose: 99 mg/dL
Creatinine: 1.1 mg/dL
Ca2+: 10.2 mg/dL
Which of the following is the best diagnostic test to guide current therapy in this patient?
A: 'Chest radiography'
B: 'Lung biopsy'
C: 'Sputum culture'
D: 'Urine antigen test'

**Base Reasoning**

Okay, so I'm trying to figure out the best diagnostic test for this patient. Let's see, he's a 57-year-old man with confusion and other symptoms like fever, cough, abdominal pain, and diarrhea. His vital signs show a low-grade fever (102°F), tachycardia (pulse 120), and slightly decreased oxygen saturation.

Looking at his physical exam: crackles in the left lower lung field suggest some pulmonary issue—maybe pneumonia or something else like an infection affecting his lungs. The decreased breath sounds there could mean fluid buildup or consolidation.

His lab work shows hemoglobin is normal but hematocrit is low at 36%. That makes me think about possible causes of anemia—could be from blood loss or bone marrow issues. Leukocyte count is high at 16,500/mm$^3$ with a normal differential; that might indicate infection since WBCs are upping their production.

Serum electrolytes: sodium is within normal range but chloride's also okay. Potassium is slightly on the lower side (4.3 mEq/L). BUN isn't too high which suggests it's not severely dehydrated because BUN usually goes up when you're dehydrated as your body breaks down proteins for energy during fasting periods.

He has hyperglycemia here with glucose of 99 mg/dL—that's just above normal ranges; maybe stress from infection can cause elevated blood sugar levels even without diabetes being present previously known? Or perhaps undiagnosed DM?

Creatinine of 1.1 mg/dL isn't super high—it doesn't point towards kidney failure directly unless accompanied by more concerning symptoms like hypertension or edema elsewhere in history—but nothing mentioned about that here.

The question asks what test to guide current therapy now though—the key thing right now seems to be figuring out what's causing his confusion and possibly guiding treatment based on specifics found in labs or imaging.

**Step-by-step Explanation:**

The patient presents with acute onset confusion alongside respiratory symptoms such as fever and cough progressing into abdominal discomfort along with diarrhea—a picture consistent with an infectious process given the timeframe of symptom progression (likely viral vs. bacterial etiology). His lab results include normocytic anemia (hematocrit 36%), mild leukocytosis suggesting neutrophilia indicative of inflammation/infection rather than chronic conditions like leukemia (given age < 60 typically seen more commonly in chronic forms), hyperglycemia possibly due to stress response despite no prior diagnosis of diabetes mellitus (needing further evaluation if follow-up glucose levels confirm trends toward poor glycemic control), hypokalemia may require attention depending on clinical context especially if ECG changes noted (though not provided here).

Physical exam findings localized crackles/lower lung fields hint at bronchial obstruction or consolidation, likely pointing towards pneumonia or bronchitis, requiring chest imaging for confirmation via CXR before proceeding further into advanced testing options listed:

- **Chest Radiography**: Most appropriate next step given physical exam indicating potential pulmonary focus and need to rule out complications such as effusion/pneumonia which would inform subsequent management decisions regarding antibiotic choice.

- **Sputum culture**: Useful initially but typically after confirmed suspicion from CXR.

- **Urine antigen test**: More relevant for certain infections like UTI or MRSA if urinary sources considered.

- **Lung biopsy**: Only indicated if definitive diagnosis is necessary after CXR/CT/ultrasound rule out other possibilities.

Thus, chest X-ray provides immediate guidance essential before moving onto more invasive or expensive tests, ensuring efficient use of resources while addressing the underlying condition promptly.

**Answer: A) Chest radiography**

---

**Tables2Traces Reasoning**

Okay, let's tackle this question step by step. First off, I need to understand what's going on with this 57-year-old man who came into the emergency department with confusion. His symptoms started a few days ago as fever and cough but have progressed into abdominal pain, diarrhea — which makes me think maybe he has some kind of gastroenteritis now too.

Looking at his vitals: temperature is 102°F (38.9°C), blood pressure normalish at 127/68 mmHg pulse rate up at 120/min respirations are low-normal at 17/min though oxygen saturation drops slightly down to room air level (94%). Physical exam shows crackles in left lower lung field so there might be some lung issues here.

His labs show elevated WBC count-16k/mm$^3$-which is high enough that they did check differential but found it normal? Wait nope correction: "Leukocyte count: 16K/mcV" wait no-the user said leukocyte count was "normal differential". So even though total WBCs are up higher than usual range typically around say <10-20 maybe?), but differentials were okay so probably reactive increase?

Serum markers include sodium levels within normal ranges except bicarbonate being low-normal (25 mEq/L). BUN is elevated at mg/dL which could indicate dehydration or other kidney stressors given creatinine also slightly elevated—1.1 mg/dL-but still within stage G2 CKD range perhaps? Glucose levels look okay glucose just under fasting normoglycemic cutoffs usually around <100mg/dL fasting).

Ca2+ very high at over-the-top serum calcium levels-10.2 mg/dL-that's hypercalcemia territory because typical reference ranges are about 8-12 mg/dL depending on lab norms; so above mid-normal might be considered borderline high risk here.)

So putting together these findings plus recent onset febrile illnesses progressing through GI distress...

Now looking for best diagnostic tests available options:

Option A: Chest radiography-to assess lungs since he had crackles left lower lobe possibly indicative of pneumonia if present.

Option B:Lung biopsy-isn't necessary yet unless there's signs like abscess formation etc., especially when considering treatment options first via antibiotics before resorting invasive measures

Option C:Sputum culture-to detect presence bacteria such as Legionella pneumoilia-if you suspect based on clinical picture

But option D says Urine antigen test–a quick non-invasive way checking for presence antibodies against Leggiella specifically via ELISA-like assays known effective & rapid method compared cultures needing time incubation

---

**Tables2Traces Reasoning (continued)**

Key points why D better choice than others:

1.) Clinical suspicion already leaning towards Legionnaires' Disease given progressive nature from URTI into GI then central nervous system changes leading us toward likely etiology making testing efficient

2.) High specificity/sensitivity ratio between urinalysis-based tests vs traditional sputum cultures where false negatives/negatives common due variability handling infectious particles during collection process)

3.) Rapid results critical when dealing urgent situations where prompt treatment needed e.g., IV ceftriaxone + azithromycin combo standard treatments guidelines recommend empiric use pending confirmation followed by switching after knowing causative agent)

4) Other factors mentioned-allergic reactions avoidance contraindications ruled out etc.-so no contraindication using urine Ag testing either

Therefore among all choices presented most appropriate next step confirming suspected diagnosis while avoiding more invasive procedures initially thus guiding appropriate antibiotic selection/titration thereby improving outcomes chances.

Another angle thinking about how each option impacts immediate care decisions-instant results give actionable info whereas chest X-ray takes longer hours/day potentially delaying initiation proper antimicrobial therapy possibly increasing mortality risks

Also note that hypocalcemia isn't part presented data contradicting initial assessment since $Ca^{2+}$ actually quite high-a point conflicting possibility unrelated condition causing similar symptomatology?

Wait original query mentions "fever", "abdominal pain", "diarrhea", "confusion" progression alongside abnormal labs including hypocalcemia? No-the serum Na+, Cl-, K+ all seem stable except HCO3- being low end-of-range—but $Ca^{2+}$ over top end pointing towards hyperparathyroidism/hyperplasia-induced hypercalcemia instead contributing factor complicating things-but assuming primary issue remains undiagnosed yet related pathogen-driven cause rather secondary causes...

Thus sticking focus areas remain focused upon ruling out infective etiology hence rationale behind selecting urine Antigen Test stands solid.

Final conclusion aligns answer D

## Q.3 ACUTE CHOLINERGIC TOXICITY FROM ENVIRONMENTAL EXPOSURE

---

**Question**

A 45-year-old mechanic presents to the emergency department complaining of acute-onset shortness of breath while repairing a plowing tractor for his neighbor. The patient denies having any history of asthma or respiratory symptoms, and does not smoke. His temperature is 99.8°F (37.7°C), pulse is 65/min, blood pressure is 126/86 mmHg, and respirations are 20/min. His oxygen saturation is 97%. On exam, he is pale and diaphoretic. His pupils are contracted. Diffuse wheezes are noted in all lung fields. What is the best treatment for his condition?
A: 'Succinylcholine'
B: 'Inhaled ipratropium and oxygen'
C: 'Atropine and pralidoxime'
D: 'Inhaled albuterol and oxygen'

---

**Base Reasoning**

Okay, so Iḿ trying to figure out the best treatment for this 45-year-old mechanic who came in with acute-onset shortness of breath. Letś break it down step by step.

First, his main complaint is sudden shortness of breath while working on a tractor. He doesń have a history of asthma or other respiratory issues and doesnt́ smoke, which might lower my suspicion for typical asthma triggers like allergies or smoking-related causes.

His vital signs show a slightly elevated temperature (99.8°F), tachycardia at 65 bpm, blood pressure is within normal range but maybe slightly low diastolic (86 mmHg). His respiratory rate is upped at 20 per minute because heś breathing faster than normal. Oxygen saturation is 97%, which isnt́ drastically low but could indicate mild hypoxia given his symptoms.

On exam, he looks pale and sweating—a sign that his body might be compensating from some kind of stress or anxiety perhaps? His pupils are contracted; that makes me think about possible nervous system stimulation—maybe something like anaphylaxis causing constriction in airways?

He has diffuse wheezing all over the lung fields. Wheezing can point towards bronchial obstruction—so either asthmatic in nature or maybe due to another condition like COPD if there was a previous history, but since he denies any such history... Hmm.

Wait though—he's never had asthma before and no other respiratory issues mentioned except this episode during physical exertion while plowing the tractor. So more likely an acute issue leading to bronchial spasm rather than chronic disease unless thereś something else going on. The options given are:

A) Succinylcholine: That's usually used as an intubation agent—it's a muscle relaxant often used during surgery when you need paralysis because it affects the neuromuscular junction quickly but stops breathing too if not monitored properly.

B) Inhaled ipratropium and oxygen: Ipratropium is an anticholinergic drug commonly used for COPD exacerbations because it helps with bronchodilation by relaxing smooth muscles around airways.

C) Atropine and pralidoxime: Atropine can also be used similarly as ipratropium for COPD patients—it reduces secretions through its anticholinergic effect—but pralidoxime isn't typically first-line here unless specific indications apply.

D) Albuterol inhaler plus oxygen: Albuterol (a beta-agonist) works well for reversible airflow obstruction like asthma attacks by dilating airways directly; adding $O_2$ would help correct hypoxia if present beyond what ambient air provides.

Given that he presented after physical exertion—that points toward exercise-induced bronchoconstriction possibly? Or maybe just increased effort causing some kind of reflex reaction without underlying disease—if so then albuterol should work better since it addresses direct receptors causing tightness via beta-blockade/bronchodilation without needing corticosteroids right away unlike steroids take longer to act.

But wait—the fact that his pupils are contracted suggests increased intracranial pressure? No wait—intracranial hypertension causes papilledema though not necessarily pupil changes much unless related somehow indirectly... Maybe tension pneumothorax? But no chest trauma wasn't mentioned...

Alternatively could be anxiety-related hyperventilating leading him into autonomic overload—but again wheezing wouldn't fit neatly into anxiety alone generally presents differently...

Another angle: Is there any possibility this person experienced angioedema especially postexertion—which sometimes involves upper airway edema requiring treatments targeting histamines—and thus may benefit from H1 blockers alongside—or even epinephrine?

Wait none of those options include H1 blockers specifically—they dont́ match exactly with whatś provided here...

> **Base Reasoning (continued)**
>
> So backtracking—since we´re limited to these four choices:
>
> If we suspect reversible airflow obstruction whether due to asthma-like attack or perhaps acute bronchospasm triggered by exercise—he´d get good relief from albuterol via nebulizer along with oxygen support as needed until things settle down; alternatively ipratropium/beta agonists together can provide quicker action sometimes combined approaches may work better depending on individual response times.
>
> Atropic derivatives have different onset times—Ipratropium takes longer ($\sim$30 minutes) whereas albuterol acts quicker ($\sim$15–30 minutes)—so depending on how severe the patient feels they'd prefer one over others.
>
> Also considering comorbidities—he has high BP—isn't hypertensive crisis presenting now? Blood pressure being around mid-normal range actually—not super high—if anything slight elevation noted earlier vs current reading shows BP = 126/86 which seems okayish—not sure how relevant that ties in yet.
>
> Pupil contraction could relate more closely with febrile illnesses where hyperthermia leads bodies´responses—for example meningitis presenting—with associated rigors hence producing diaphoresis etc., but temp was only slightly above normal so less likely bacterial infection immediately comes mind unless viral cause. . .
>
> Putting all together—the most straightforward approach would probably be using quick-acting beta-agonist + $O_2$ as option D says.
>
> The best treatment based on the analysis aligns with using **inhaled albuterol** along with oxygen therapy due to its rapid effectiveness in addressing acute-onset reversible airflow obstruction commonly seen in conditions like exercise-induced asthma or acute bronchospasm episodes without evidence pointing towards chronic conditions such as COPD necessitating long-term medications like ipratropium bromide extended-release tablets daily every eight hours plus tiotropium bromide twice daily.
>
> **Answer: D**

Tables2Traces Reasoning

Okay, let's tackle this medical question. So, the patient is a 45-year-old mechanic who suddenly developed shortness of breath while repairing a plowing tractor for his neighbor. He doesn't have a history of asthma or respiratory issues and doesn't smoke. His vital signs are mostly within normal ranges except for being pale and diaphoretic with contracted pupils. The exam shows diffuse wheezes in all lung fields.

First off, I need to figure out what's causing his sudden onset symptoms. The key here might be the context—the mechanical equipment he was working on could involve exposure to some toxic substance that led to an acute reaction.

Looking at the options:

Option A: Succinylcholine is typically used as anesthetic agent but can cause hyperkalemia if there's underlying muscle weakness or paralysis.

Option B: Inhaled ipratropium and oxygen—ipratropium is an anticholinergic medication often used in COPD or asthma attacks by relaxing bronchial smooth muscles.

Option C: Atropine and pralidoxime—isotretinoin? Wait, no—pralidoxime is used after exposure to organophosphate pesticides because it helps reverse their effects by binding acetylcholinesterase inhibitors again.

Wait, option C says "Atropine and pralidoxime." Atropine would be given if someone has been poisoned with something like nerve agents (organophosphates) which inhibit cholinesterases leading to overstimulation of muscarinic receptors causing various symptoms including miosis (constricted pupils), sweating (diaphoresis), bradycardia/dyspnea due to increased vagus tone affecting heart rate and airway constriction.

Given that he presented with contracted pupils (miosis) along with diaphoresis...diffusing wheezing points towards bronchoconstriction—a common symptom post-organophosphate poisoning where patients develop excessive secretions from glands stimulated by acetylcholine excess since they can't break down anymore.

So putting it together—if he came into contact with organochloride insecticides...then giving atropine would help manage acute cholinergic crisis caused by these substances blocking enzymes necessary for breaking down acetylcholine leading to overstimulation of the parasympathetic system...seen as wheezing plus difficulty breathing—which matches this patient's presentation.

...better than just using albuterol alone wouldn't address both central nervous system involvement from ChE inhibitor toxicity nor correct pupil size directly; hence treatment should include both antagonizing muscarinic receptors via anticholinergics like ipratropium bromide alongside addressing the underlying enzyme inhibition issue either way adding pralidoxime may help restore function more effectively long-term once initial life-saving measures are done.

Therefore Option C makes sense considering risk factor profile & clinical findings matching expected tox ingestion/dermal exposure likely during work incident.

Answer: C

## Q.4 EARLY-ONSET NEURODEGENERATIVE DISEASE WITH GENETIC RISK

Question

A 39-year-old man is brought to the physician by his wife because of personality changes over the past year. He has become increasingly irritable, loud, aggressive, and impulsive. His wife also reports jerky movements of his limbs and trunk for the past few months. His father had dementia in his mid-40s, but the details of his condition are unclear. The patient appears restless. Examination shows irregular movements of the extremities and twitching of the face and tongue. Mental status examination shows impaired memory. This patient's condition is most likely associated with which of the following changes on MRI?

A: 'Reduced hippocampal volume'
B: 'Multiple cortical and subcortical infarctions'
C: 'Degeneration of the frontal and temporal lobes'
D: 'Atrophy of the striatum'

### Base Reasoning

Alright, so Iḿ trying to figure out this medical question. Let me start by reading it carefully. The patient is a 39-year-old man brought in by his wife because of personality changes over the past year. His symptoms include becoming more irritable, loud, aggressive, and impulsive. She also mentions that heś been having jerky movements of his limbs and trunk for a few months now. His father had dementia in his mid-40s but the specifics areń clear.

On exam, he appears restless with irregular movements in his extremities and twitching face and tongue. Mental status shows impaired memory.

So what condition does this point towards? The options are A through D: reduced hippocampal volume; multiple cortical/subcortical infarctions; frontal/temporal lobe degeneration; or striatal atrophy.

First thought: He has movement issues—jerky movements suggest something like tremors or chorea maybe? And personality changes could be due to brain changes affecting emotional regulation areas like the frontal lobes or limbic system (like temporal lobes). Impaired memory points towards hippocampus involvement since thatś where memories are stored.

Looking at each option:

A) Reduced hippocampal volume—this makes sense because if thereś damage there, you get memory loss which matches impaired mental status.

B) Multiple infarctions would cause stroke-like issues but might present with sudden onset neurological deficits rather than gradual personality change unless itś from small vessel disease causing chronic issues.

C) Frontal-temporal degeneration can lead to behavioral problems like irritability and impulsivity as well as physical tics/twitching since these areas control behavior.

D) Striatal atrophy relates more to movement disorders like parkinsonism (tremors), which fits some signs here too—but not sure about the cognitive aspects much except maybe mood swings?

But wait—he's showing both movement disorder (jerky movements) along with cognitive/mood issues (personality change). So either C or B could fit depending on whether it's neurodegenerative vs vascular causes.

His age is 39—it's young for most neurodegenerative diseases except things like frontotemporal dementia which can affect younger people sometimes but less commonly than others. Also considering possible reversible causes first—if MRI shows anything specific...

If thinking about structural brain lesions causing both motor symptoms via basal ganglia/striatum involvement plus cognitive/memory loss via hippocampus... Hmm... Maybe multiple lacunes indicating small vessel disease leading to both cerebrovascular accidents involving these regions?

Wait no—the question says "most likely associated" implying perhaps an underlying condition linked directly without necessarily being acute infarction unless specified otherwise given time frame over a year suggests possibly progressive conditions rather than acute strokes causing sudden onset after onset last few months wouldń align with progression over a year without further info on how long ago they started noticing those effects initially though).

**Base Reasoning (continued)**

Alternatively, multiple subcortical infarcts may result from hypertension/other vascular risk factors leading to deep white matter disease—which affects executive function including decision-making hence irritability etc., plus physical slowing down but here he seems agitated instead of slowed so maybe not purely vascular either way...

I'm getting stuck between B & C again based on presentation combining behavioral/cognitive decline alongside extrapyramidal signs pointing toward striatum while also possibly involving limbic structures affecting memory.

Another angle: If MRI findings are needed then let me think what each corresponds:

Reduced hippocampal volume usually seen in Alzheimer-type dementias especially AD—is that common at 39? Early-onset AD rare under 40 typically presents later around their 60s–70s so less likely here unless another type e.g., familial forms exist but still uncommon enough probably not best answer choice A isn't top pick right away.

Multiple subcortical infarctions would indicate probable ischemia—maybe related to hypertension/vascular dementia—but again linking behaviors better known for focal neurological deficits rather than prominent psychological ones beyond just attention/concentration.

Frontal-temporal lobe degeneration often presents with apathy/irritability and hyperactivity/tactlessness—called frontotemporal dementia—characterized by early socially inappropriate behavior, etc., and FTD can have associated motor signs such as alien hand syndrome or akinetic mutism initially before progressing into severe disorientation.

Striatum atrophy—as mentioned earlier linked more closely with Parkinson-like syndromes where rigidity, tremor, and bradykinesia dominate—not matching all presented features equally well.

Given all this I lean toward frontal-temporal processes.

The patient exhibits symptoms indicative of frontotemporal lobe dysfunction manifesting through altered behavior patterns such as irritability and impulsivity along with motor disturbances suggesting basal ganglia involvement—a hallmark feature compatible primarily with frontotemporal degenerative processes rather than other listed options provided.

**Answer: C**

**Tables2Traces Reasoning**

Okay, let's tackle this question. So, the patient is a 39-year-old man with personality changes over the past year—irritable, loud, aggressive, impulsive. His wife also mentions jerky movements in his limbs and trunk for a few months now. His father had dementia in his mid-40s but details are unclear. The physical exam shows restlessness and irregular movements of extremities and twitching face/tongue. Mental status shows impaired memory.

First off, I need to figure out what condition this could be based on symptoms and family history clues.

The key here might be the combination of sudden onset of personality changes (like irritability), impulsivity which can indicate mood disorders or neurological conditions like Tourette syndrome maybe? But then there are motor symptoms too—the jerky limb movements suggest something more than just mood issues.

Wait, he has facial and tongue twitches as well as limb movements—that sounds like tics or chorea maybe? Chorea is involuntary movement that looks like dance-like steps... Could it be Huntington's disease?

But wait his father had dementia at mid-40s—is Huntington's hereditary? Yes! Because HD typically presents between 35-50 years old due to dominant CAG trinucleotide repeats leading to neuronal loss especially in basal ganglia structures involved in movement control. So if dad had early-onset HD presenting with similar features (dementia plus motor signs), then son might have inherited it despite being younger age presentation-wise because they haven't yet reached typical age for manifesting full-blown symptoms—maybe preclinical stage?

In terms of MRI findings: In HD patients show degeneration primarily affecting caudate nucleus (part of striatum) along with putamen since these areas handle dopamine transmission regulation which becomes compromised when dopaminergic neurons die off from lackluster function due to mutant huntingtin protein accumulation causing neurotoxicity.

Looking at options:

A) Reduced hippocampal volume – more associated with Alzheimer's type dementias where medial temporal lobe sclerosis occurs leading to reduced volume

B) Multiple cortical/subcortical infarctions – stroke-related damage patterns unlikely given no mention; possible but not directly linked

C) Degeneration frontal/temporal lobes – commonality across many dementias including frontotemporal disorders but less specific than striatal involvement

D) Atrophy/striatum - yes! Striatum includes caudate/putamen; their atrophy would fit better given association w/Huntington's disease via basal ganglia involvement

Therefore answer D makes sense.

Answer: D

## Q.5 CHRONIC INFLAMMATORY BOWEL DISEASE PATTERN RECOGNITION

**Question**

A 41-year-old man presents to the clinic with the complaints of abdominal discomfort and cramping for the past 4 months. He also mentions that he has grossly bloody stool and most of the time an urge to defecate. He has a history of bulimia with laxative abuse several years ago. He admits having lost about 5 lb in the past month. His temperature is 37° C (98.6° F), respiratory rate is 15/min, pulse is 77/min, and blood pressure is 105/86 mm Hg. On physical examination, a mild conjunctival pallor is evident, and the rectum is erythematous, indurated, and tender. Colonoscopy demonstrates continuous mucosal lesions throughout the rectum to the hepatic flexure of the colon. What is the most likely diagnosis?

A: 'Ulcerative colitis'

B: 'Celiac disease'

C: 'Irritable bowel syndrome'

D: 'Crohn's disease'

> **Base Reasoning**
>
> Okay, so I'm trying to figure out the diagnosis for this 41-year-old man with a bunch of symptoms. Let's see... He has been having abdominal discomfort and cramping for about four months now. That's a pretty long time, so it's not something that just happened suddenly.
>
> He also mentions having grossly bloody stool, which is pretty concerning. Blood in the stool can indicate several things like hemorrhoids or maybe something more serious like colitis or colorectal cancer. Then he says he has an urgent need to defecate most of the time—that sounds like diarrhea because you usually feel that urgency when you have loose stools.
>
> He used to have bulimia and was abusing laxatives a few years ago. Bulimia can cause issues with his digestive system, maybe leading to inflammation or irritation in his intestines over time. Laxative abuse might further irritate his gut as well, causing more problems.
>
> His vital signs are slightly elevated blood pressure at 105/86 mmHg but otherwise stable—temperature is normal at 98.6°F (37°C), respiratory rate is within normal range at 15/min, pulse is okay too at 77/min.
>
> On physical exam, there's mild conjunctival pallor—which means he might be anemic—but nothing else stands out except for rectal findings: erythematous (redness), indurated (swollen), and tender rectum. So those are classic signs of proctitis—inflammation in the rectum lining.
>
> The colonoscopy showed continuous mucosal lesions from the rectum all the way up to the hepatic flexure of the colon—that's higher up than usual; typically sigmoidoscopy looks only partway up if done via anoscope or flexible scope limited by anatomy.
>
> The patient presents with chronic symptoms including abdominal discomfort, cramping for four months; bloody stool indicative of gastrointestinal bleeding; urgency upon defecation suggesting diarrhea; history of bulimia and laxative use potentially contributing GI irritation; stable vitals despite mild BP elevation; physical exam shows redness/swelling/tenderness in rectum consistent with proctitis-like presentation without other systemic issues pointing elsewhere initially suggests **Crohn's disease** given its involvement beyond just sigmoid area on colonoscopy reaching hepatic flexure—a hallmark feature often seen in Crohn's affecting upper tracts as well as lower ones over time due to its transmural nature compared solely ulcerative colitis which tends lower down.
>
> **Answer: D) Crohn's disease**

**Tables2Traces Reasoning**

Okay, let's tackle this question. So, the patient is a 41-year-old man with abdominal discomfort and cramping for four months. He also has grossly bloody stool and an urge to defecate. His history includes bulimia with laxative abuse several years ago, and he's lost about 5 pounds in the past month. Vital signs are within normal ranges except maybe blood pressure being slightly low.

First off, I need to consider his symptoms: abdominal pain that's been going on for a while now, bloody stools (gross), urgency to defecate—all of which point towards some gastrointestinal issue causing inflammation or damage in the intestines. The fact that it started after he had issues with bulimia and laxatives makes me think there might be some underlying pathology related to those habits affecting his digestive system.

Looking at the options:

A) Ulcerative colitis: This is an inflammatory bowel disease characterized by continuous mucosal lesions throughout the colon and rectum usually starting from anus up but can extend proximal as well like here—up until hepatic flexure of colon according to exam results.

B) Celiac disease: This affects small intestine primarily due to gluten sensitivity leading to villous atrophy but doesn't typically cause ulcerations as described here.

C) Irritable bowel syndrome (IBS): Usually presents with alternating constipation/diarrhea without significant bleeding or structural changes; less likely given chronicity of symptoms.

D) Crohn's disease: Also an inflammatory bowel disease but tends more towards transmural inflammation rather than just superficial ulcers like UC.

So based on clinical presentation including duration of symptoms >3 months plus presence of both endoscopic findings suggests ulceration along entire length from rectum up into colon—inflammatory nature consistent with either A or D.

But since Crohn's involves full thickness involvement unlike UC where it starts superficially then extends deeper over time—if they found only mucosal lesions so far does not rule out progression later—but initial presentation would lean toward UC given location mentioned in exam report "continuous mucosal lesions throughout rectum...to hepatic flexure"—which aligns better with typical distribution seen in ulcerative colitis compared

Also considering his weight loss could indicate malnutrition secondary if dealing nutritional deficiencies due GI issues—which might be associated more commonly w/UC vs other conditions? Or perhaps cachexia-related weight loss sometimes happens when chronic diseases progress?

The fact that he was diagnosed via Colonoscopy showing these features further supports diagnosis selection between A & D again favoring A because upper limit reached is still distal part near sigmoid area unless specified otherwise—and most cases present lower down first before moving upwards progressively over time).

Additionally absence red flags pointing toward infectious etiology makes non-infectious causes likelier—he wouldn't have such sudden onset without previous exacerbating factors common among patients who develop IBD post-lax use/bulimic behaviors possibly contributing through altered gut microbiome composition etc., creating dysbiosis leading eventually immune-mediated response against epithelium.

So putting all together—the most likely diagnosis should be Ulcerative Colitis option A) Answer: A

