# OpenReview forum: "Tables2Traces: Distilling Tabular Data to Improve LLM Reasoning in Healthcare"
_ICLR.cc/2026/Conference — Submitted to ICLR 2026_

### Official Review · Reviewer_m5kw · 2025-10-29

**Soundness:** 3
**Presentation:** 3
**Contribution:** 2
**Rating:** 4
**Confidence:** 4

**Summary:**

This paper presents Tables2Traces, a framework that turns tabular clinical data into reasoning traces to fine-tune large language models. Using cardiovascular data from UK Biobank, the method shows clear improvements on MedQA and MedMCQA, demonstrating that structured data can provide useful reasoning supervision.

**Strengths:**

- **Novel data-centric perspective.**

The paper introduces an interesting data-to-trace distillation framework that converts structured tabular data into natural language reasoning traces, providing a fresh and potentially impactful perspective on how non-textual data can be used to enhance LLM reasoning. This idea is interesting.

- **Clear motivation and methodology.**

The paper is well-organized and easy to follow. The motivation for leveraging structured clinical data is clearly articulated, and the experimental setup is described with sufficient detail for reproducibility.

**Weaknesses:**

**Major Weakness:**

- **Unverified impact on general reasoning.**

While the method improves performance on medical QA benchmarks, the authors did not assess whether fine-tuning on domain-specific traces affects the model’s general reasoning or language understanding. No results are reported on standard reasoning benchmarks (e.g., GSM8K, MMLU, ARC), leaving the potential risk of catastrophic forgetting unexamined. It also remains unclear whether the observed gains stem from genuine reasoning improvement or from overfitting to the question–answering format.

- **Narrow evaluation scope.**

Experiments are restricted to two medical QA datasets of similar structure. It remains uncertain whether the proposed approach benefits other reasoning tasks such as clinical summarization, diagnostic inference, or structured-to-text generation. The generalizability across task types has not been demonstrated. Although the paper repeatedly claims that Tables2Traces is domain-agnostic, all training data originate exclusively from a cardiovascular-disease cohort. No experiments are conducted on other medical or non-medical domains. Consequently, it is unclear whether **this method** generalizes beyond this specific disease area, or whether the observed improvements arise from domain-specific correlations/biases within the cardiovascular dataset.

- **Insufficient ablation and analysis.**

Only a coarse comparison between the “simple” and “full” variants is provided. The paper lacks finer-grained ablations isolating the contributions of the contrastive, counterfactual, and anchor components, as well as sensitivity analyses regarding the number or quality of generated traces.

**Minor Weakness:**

- **Trace quality unverified.**

All reasoning traces are automatically generated by an existing LLM without human or quantitative validation. Although the overall empirical gains suggest that these traces provide useful supervision in aggregate, the absence of quality verification weakens the claim that the model genuinely learns better reasoning rather than benefiting from correlated linguistic patterns, especially for the medical domain.

- **Limited algorithmic novelty.**

The proposed framework introduces no new optimization objective or model architecture. It essentially performs standard instruction tuning on synthetic data obtained through prompt engineering. The contribution lies primarily in the data-to-trace distillation paradigm rather than in algorithmic innovation.

- **Conceptual inconsistency with the stated motivation.**

The introduction emphasizes grounding LLM reasoning in domain-specific structured data, whereas the discussion later claims that the model learns “general reasoning patterns.” This interpretation conflicts with the paper’s initial motivation and lacks empirical support. The authors should clarify whether their goal is to enhance domain-grounded reasoning or to induce domain-agnostic reasoning transfer.

**Questions:**

See Above.

---

> ### Author Response · Authors · 2025-11-22
> **Rebuttal for Reviewer m5kw**
>
> ## Rebuttal for Reviewer m5kw
> We want to thank the reviewer for taking their time to provide good and thoughtful feedback on our work. We provide a point by point response below.
>
> ## 1. General reasoning benchmarks (GSM8K, ARC, MMLU) & catastrophic forgetting
> We thank the reviewer for raising the question about general reasoning benchmarks. Our goal in this work is not to improve general-purpose reasoning, but to examine whether domain-grounded tabular supervision can improve medical reasoning tasks. This focus stems from our core motivation: in many real-world settings, especially in medicine, clinicians or institutions may have access to rich tabular patient data but little or no curated QA corpora. A key question, therefore, is whether such domain-owned structured data can be used to improve an LLM’s reasoning within that domain. The cardiovascular cohort provides structured data with medically meaningful variation, allowing us to evaluate the effect of reasoning traces on clinical QA tasks where domain knowledge is essential.
> We agree that evaluating effects on non-medical reasoning benchmarks such as GSM8K or ARC is an interesting direction, but it is outside the scope of this study. More importantly, our experiments already provide evidence against simple format overfitting:
> The Tables2Traces (simple) variant, which shares the same QA-format alignment data, shows degradations on several out-of-distribution subsets.
>
> The full Tables2Traces model improves not only on patient-style questions but also on abstract questions that contain no patient structure.
>
> This strongly suggests that performance gains are not due to QA pattern memorization but arise from the structured reasoning signal introduced by the full method.
> ### Updated manuscript:
> (1) In the Discussion, we have clarified that our goal is to study domain-grounded reasoning supervision rather than general-purpose reasoning.
>
> ## 2. Domain generality and concerns about cardiovascular-specific correlations
> We appreciate the reviewer’s concern about the limited training domain. While all tabular data come from cardiovascular patients, the downstream evaluations span dozens of medical specialties, and the model improves across 15 of 17 MedQA domains (Table 1) and 16 of 21 MedMCQA domains (Table 3). This suggests that the reasoning signal generalizes beyond the training domain.
> Importantly, if improvements were driven by cardiovascular-specific correlations or biases, we would expect the Tables2Traces (simple) variant, which is trained on the same tabular data and uses the same QA subset, to exhibit similar improvements. Instead, the simple variant degrades on multiple domains, both semantically out-of-distribution (non-cardiology topics) and stylistically out-of-distribution (abstract questions). This divergence indicates that the full method’s improvements stem from the structured reasoning components rather than domain-specific tabular correlations. We have included a specific section dedicated to this point in the Discussion.
>
> We agree that extending the approach to additional medical or non-medical tabular datasets would be valuable future work. Our present aim is to introduce and validate the feasibility of a tabular-to-reasoning supervision paradigm in a well-defined clinical setting.
>
> ### Updated manuscript:
> (1) We have added a paragraph to the Discussion explaining why the simple-vs-full comparison provides evidence against cardiovascular-specific overfitting and clarified that cross-domain generalization is empirically observed even with single-domain tabular training data.

---

> ### Author Response · Authors · 2025-11-22
> **Rebuttal to Reviewer m5kw (continued)**
>
> ## 3. Ablations and component isolation
> We appreciate the reviewer’s suggestion to investigate the contribution of each component in isolation. Understanding which elements of a structured reasoning trace contribute most strongly to downstream performance is indeed an interesting direction for future work.
> In this paper, however, our primary goal is narrower and more foundational: to test whether structured tabular data can serve as an effective source of reasoning supervision for domain-specific LLM alignment. For this purpose, the most informative comparisons are the three regimes we study:
> - Base model, which receives no tabular supervision
>
>
> - Tables2Traces (simple), which introduces tabular information in the most direct form through patient descriptions
>
>
> - Tables2Traces (full), which introduces a structured reasoning scaffold combining contrastive comparisons, plausibility evaluation, and counterfactual edits
>
>
> These comparisons probe the core scientific question. The simple variant shows clear overfitting effects, both semantically and stylistically. The full variant mitigates these effects and improves across medical specialties and on abstract questions without patient structure. This contrast demonstrates that not only can tabular data be used as supervision, but also that the form of supervision strongly matters.
>
> The internal components of the full method are intentionally designed to operate together as a single structured reasoning mechanism, drawing on ideas from contrastive learning and causal reasoning. While isolating each component would be scientifically valuable and could reveal further insights, such fine-grained analysis is orthogonal to the central goal of determining whether tabular-derived reasoning traces can meaningfully improve domain-specific reasoning.
>
> ## 4. Trace quality and verification
> We agree that extensive trace-quality evaluation is valuable. Our current clinician review (Appendix O) found no safety-critical issues, but noted expected limitations such as missing context and occasional overconfidence due to the coarse nature of tabular features. These limitations are inherent in working with snapshot tabular records rather than full patient histories.
> To address concerns about whether the model genuinely learns better domain-specific reasoning rather than correlated linguistic templates, we are adding five qualitative examples where the base model fails and Tables2Traces succeeds. These examples illustrate the types of reasoning patterns the model acquires, such as differentiating cases, assessing plausibility, or considering hypothetical changes.
> ### Updated manuscript:
> (1) In the Appendix, we have added qualitative examples illustrating improved reasoning behaviors.
>
> (2) In the Discussion, we have added a paragraph to acknowledge the limitations of synthetic trace quality and potential bias inheritance.
>
> ## 5. Algorithmic novelty and scope of contribution
> We agree with the reviewer that Tables2Traces does not introduce new model architectures or optimization objectives. Our contribution is data-centric: we show that structured tabular data can be transformed into reasoning supervision that enhances downstream performance. This perspective complements algorithmic developments and is aligned with recent directions emphasizing the importance of data quality and supervision format in LLM fine-tuning.
> ### Updated manuscript:
> (1) In the Introduction (specifically the Contributions paragraph), we have emphasized the data-centric nature of the work and remove phrasing that could be interpreted as algorithmic novelty.
> ## 6. Clarifying the conceptual goal (domain-grounded vs. domain-general reasoning)
> We thank the reviewer for pointing out a potential inconsistency. Our primary goal is to ground LLM reasoning in structured tabular data. The observation that the resulting model generalizes across medical specialties does not imply that we aim to induce universal domain-agnostic reasoning. Rather, it suggests that domain-grounded reasoning signals can transfer beyond the original tabular domain. We will clarify this distinction in the revised manuscript.
> ### Updated manuscript:
> (1) We have added wording in the Abstract to clarify that the goal is domain-grounded reasoning, not general improvements to reasoning.
>
> (2) We have further clarified the intent in Experiments (Section 4)
>
> (3) In the Discussion (Section 5), we have explicitly delineated claims about improved reasoning to emphasize that improvements are domain-grounded clinical reasoning rather than general improvements.

---

### Official Review · Reviewer_q4yp · 2025-10-30

**Soundness:** 2
**Presentation:** 3
**Contribution:** 2
**Rating:** 4
**Confidence:** 4

**Summary:**

This paper proposed TABLES2TRACES, a textual supervision corpus generation pipeline using structured tabular data to improve LLM reasoning competency. Backgrounded in medical QA tasks, TABLES2TRACES process each tabular entry into a corpus sample in four steps: (1) feed detailed feature content to a base LLM to generate a textual summary; (2) find nearest two entries from survived tabular entries and deceased ones using Gower distance; (3) organize the summaries of these three entries with the designed prompt template as corpus inputs; (4) pass the prompt to the base LLM to generate reasoning trace as corpus target. Notably, 90% of the final fine-tuning corpus is TABLES2TRACES-converted samples with the rest 10% from QA datasets. Experiments on MedQA and MedMCQA benchmarks show the consistent gains brought from the converted tabular supervision corpus, and comparison with fully QA-optimized models are conducted, indicating potential usability of TABLES2TRACES.

**Strengths:**

## originality
Structured tabular data could be potential form of supervision corpus to enhance LLM reasoning competency: This paper assume a possible scenario where users have limited access to curated unstructured text corpus while rich tabular data exists, and TABLES2TRACES provides a tool to enable LLMs to consume such structured modality, and the paper firstly verified such usability.

## significance
Although cannot surpass optimized models supervised by curated QA corpus in the paper, the experiment results still suggest a future trend that tabular data has potential to be a supplementary or even alternative supervision source for LLM reasoning tasks.

**Weaknesses:**

## Overall Assessment
From my opinion, the empirical contribution of TABLES2TRACES far outweighs its technical contribution. The paper verified supervision usability of tabular data  modality on LLM reasoning tasks, while the core component for tabular-to-corpus conversion is driven by heuristic prompt engineering, all rely on LLM’s own capability, thus it is hard to convey clear technical inspiration to audience, or hard to figure out whether such prompt design is optimal. Besides, the experiment is not rigorous enough to fully support the paper conclusion. Several points are detailed bellow.

## Method
-	In “Contrastive Neighbor Selection” we use Gower distance (line 163) to fetch similar entries, while whether the selection of other distance metric functions will impact the results is not discussed.
-	In “Reasoning extraction via prompt design” we assign all tasks, including “Differential reasoning” to list decisive feature difference, “Label Plausibility” to recognize label noise, “Counterfactual Planning” to suggest one minimal feature edit, to the LLM itself rather than alternative machine learning methods (e.g., calculate feature importance with tree models, noisy label learning to recognize label plausibility, and Shapely Value to find sensitive features), the superiority of such “all done with LLM” methodology is not clear.

## Experiment
-	Apart from tabular-generated texts, the final training corpus also contains 10% QA examples from MedQA (or MedMCQA) datasets, which hinders the direct recognition on performance gain brought by tabular data, or why we include QA examples for training is not explained.
-	Although ablation study on “LLM only tuned with these 10% QA examples” is conducted and performance decline is observed, this result cannot demonstrate that the QA corpus has no positive effect on the main experiment results, since the usability of the data depends on its scale.

## Others
-	In related work of line 111-113, TabNet and FT-Transformer is transformer-based small neural networks rather than language models for tabular prediction, maybe recent works like LLaMA-GTL[1], TP-BERTa[2] are more suitable.
-	Line 164, the meaning of denotation s is not given.
-	Over-claimed contributions, line 81, “without human annotation” is not strict, since some labeled tabular data also requires human annotation; “without QA corpora” while the main experiment still include QA examples.

### Reference

[1] From Supervised to Generative: A Novel Paradigm for Tabular Deep Learning with Large Language Models, KDD 24.

[2] Making Pre-trained Language Models Great on Tabular Prediction, ICLR 24.

**Questions:**

1. How about the results of only using tabular-generated corpus for the main experiment?
2. see weakness

---

> ### Author Response · Authors · 2025-11-22
> **Rebuttal for Reviewer q4yp**
>
> ## Rebuttal for Reviewer q4yp
> We thank the reviewer for their thoughtful review of our work. Below, we provide a point by point response to the points you addressed.
> ## 1. Choice of distance metrics
> We appreciate the reviewer’s question regarding the use of Gower distance. We selected Gower because it is a standard metric for mixed-type medical data, but the Tables2Traces framework is not tied to this specific choice. The distance metric is a modular component: any alternative measure for mixed-type similarity (or even domain-specific learned metrics) can be substituted without altering the overall pipeline. If a different distance metric was found to be superior for downstream performance, incorporating it would simply be changing the distance metric and would not undermine the proposed framework. We will clarify in the manuscript that the neighbor-selection step is metric-agnostic and that distance learning is a promising direction for future instantiations.
> ### Updated manuscript:
> (1) We have added a short sentence in Section 3.2 stating that the neighbor-selection module is independent of the specific distance metric and that alternative or learned metrics can be incorporated without modifying the framework.
>
> ## 2. Why use an LLM-only pipeline for differential reasoning, plausibility, and counterfactual edits?
> We agree that many machine learning methods (e.g., feature importance from tree models, noisy-label detection, SHAP sensitivity analyses) can shed light on tabular structure. We also agree that relying solely on large language models for analysis regardless of the context is ill-advised. However, Tables2Traces is specifically designed to produce textual reasoning traces that serve as natural-language supervision. For this reason, incorporating insights gained from tools such as SHAP or tree feature-importance methods would require the numerical representations to be translated into human-readable reasoning sequences in natural language after being computed. Additionally, computing such metrics would often require the training of a model, thereby increasing the computational cost.
>
> The motivation for using LLMs is therefore not convenience, but necessity: the model is trained on textual reasoning signals, so the supervision must itself be expressed in text. To illustrate that the model is internalizing the high-level behavior encouraged by the prompts (differential comparisons, plausibility checks, counterfactual modifications), we have included illustrative examples in Appendix Q. These are cases where the Base model fails, but where our framework succeeds. We have annotated the specific instantiations of the reasoning behavior. This strengthens the justification for the pipeline design.
>
> ### Updated manuscript:
> (1) In Appendix Q, we have provided qualitative examples of reasoning traces when answering Medical QA questions, showcasing the behaviors of the model.
>
> ## 3. Inclusion of 10% QA examples and their role
> We thank the reviewer for raising this point. The 10% QA subset is included for a single purpose: task-format alignment, which is a standard procedure when the main supervision signal (here, reasoning traces) does not teach the model how to structure its final answers. The QA data acts as a lightweight calibration step, not as a substantive source of task knowledge.
> Crucially, Tables2Traces (simple) also uses this same QA subset. If the QA examples were responsible for performance improvements, the simple variant would benefit as well. Instead, it shows performance degradations in several out-of-distribution settings, indicating that the QA data is not the driver of the gains observed in the full method.
> As the reviewer correctly points out, we also include an ablation showing that training only on this 10% QA subset performs slightly worse than the base model. Combined with the very small size of this subset relative to the 105k reasoning traces, this strongly supports that the QA samples do not meaningfully influence model performance and are not a confounding factor.
> ### Updated manuscript:
> (1) In Section 4 (specifically in the “Setup” paragraph), we clarify that the QA subset is used solely for task-format alignment.
>
> (2) In the Discussion, we clarify that the comparison to the simple variant showcases that the 10% QA inclusion does not contribute meaningfully to the performance increase.

---

> ### Author Response · Authors · 2025-11-22
> **Rebuttal for Reviewer q4yp (continued)**
>
> ## 4. Results using only tabular-derived supervision & commentary on scalability
> We appreciate the reviewer’s interest in evaluating the model trained exclusively on tabular-generated traces. While this is an interesting direction, the core evidence in the paper already isolates the effect of the tabular supervision. First, the QA-only ablation underperforms the base model, demonstrating that the QA subset is not a meaningful source of task knowledge. Second, both the simple and full variants use the same small QA subset, yet only the full method shows robust improvements; the simple variant even degrades in several out-of-distribution settings. This strongly indicates that the gains stem from the reasoning supervision rather than from the QA examples.
>
> Regarding scalability: Tables2Traces employs standard supervised finetuning, and thus inherits its well-understood computational scaling properties; training cost grows linearly with dataset size, and trace generation is a one-time, trivially parallelizable step. If the reviewer’s question concerns performance scaling with different numbers of traces, that is a modeling question and a natural direction for future work. It is orthogonal to the finetuning mechanism itself and does not affect the validity of the current conclusions.
> ### Updated manuscript:
> (1) In the Discussion, we have explained that the existing ablations already demonstrate that the QA subset does not drive the observed improvements.
>
> ## 5. Technical contribution and prompt design
> We appreciate the reviewer’s concern regarding the role of prompt design. Tables2Traces is a framework in which tabular data is systematically transformed into structured natural-language supervision. Although the framework is modular, the specific instantiation used in this paper is not arbitrary or heuristic. Each component of the prompt design is motivated by principles drawn from established machine learning ideas:
> Contrastive reasoning is inspired by contrastive learning. Presenting a patient alongside near-neighbors with different outcomes encourages the model to identify discriminative factors, rather than memorize absolute feature values.
>
> - Label plausibility checks mirror the idea of evaluating whether an observed outcome aligns with available evidence, similar in spirit to noisy-label detection and uncertainty estimation, but expressed in natural-language form so that it can serve as supervision for an LLM.
>
> - Counterfactual edits reflect core ideas from causal inference and feature-sensitivity analysis. Asking “what minimal change could alter the outcome” encourages the model to internalize directional and mechanistic relationships rather than surface correlations.
>
> Our experiments show that this structured prompt design provides substantially stronger supervision than naively converting each row into a free-form patient description. The qualitative examples in Appendix Q illustrate that the model trained with Tables2Traces learns to perform the kinds of high-level reasoning behaviors that the structured design was intended to encourage.
> At the same time, Tables2Traces does not require this exact form of reasoning extraction. The framework allows future work to incorporate alternative or hybrid mechanisms (for example integrating feature-importance methods or learned tabular representations) as long as they yield textual reasoning supervision.
> ### Updated manuscript:
> (1) In Section 3 (specifically the “Reasoning extraction via prompt design” paragraph), we have described the motivation behind the structured prompts, linking each part of the template to established ideas in contrastive learning and causal reasoning, and clarifying that the existing implementation is a principled instantiation rather than an ad-hoc prompt template.
>
> ## 6. Minor issues
> We thank the reviewer for identifying several smaller issues.
>
> ### Updated manuscript:
> (1) In Related Works (Section 2), we have included LLaMA-GTL and TP-BERTa, to better reflect recent tabular-prediction approaches.
>
> (2) In the Methods (Section 3.2), we have corrected the missing definition of sss.
>
> (3) Throughout the paper, we have tried to clarify that the main method does use a small amount of QA data for task alignment.

---

### Official Review · Reviewer_BRY8 · 2025-10-31

**Soundness:** 2
**Presentation:** 4
**Contribution:** 3
**Rating:** 6
**Confidence:** 3

**Summary:**

The paper explores using tabular data to generate contrastive reasoning examples for LLM fine-tuning in the clinical domain. While fine-tuning with contrastive reasoning examples is an established approach, automatically generating these reasoning examples from tabular data is a novel contribution. Past work on LLMs for tabular data has mostly focused on enhancing LLM understanding of tabular data, which is not the focus of the paper.

The contrastive reasoning examples are generated from tabular data through a multi-step process. First, an LLM is prompted to transform the tabular data into textual summaries. Next, triples are formed by finding nearest survivors and nearest deceased patients (by Gower distance). For each triple, an LLM is again prompted to generate a contrastive reasoning example, with explicit direction provided intended to enhance differential reasoning, label plausibility, and counterfactual planning. (Human experts provide a qualitative evaluation of their generated contrastive reasoning examples.)

The authors report results on common medical QA benchmarks and compare their method to the SOTA model Aloe. They also consider fine-tuning based on patient narratives generated from the tabular data and find that contrastive reasoning examples perform better. Finally, they demonstrate the potential of their method to enhance performance on out-of-domain questions.

**Strengths:**

- The paper explores an original idea (supervised fine-tuning for clinical reasoning on contrastive reasoning examples automatically generated from tabular data) with high potential.

- The paper is clearly written and well-situated with respect to the most important related works in LLM fine-tuning and LLMs for tabular data.

- The experimental analysis shows clear gains in out-of-domain questions and demonstrates the effectiveness of using tabular data to generate specifically contrastive reasoning examples rather than patient narratives.

**Weaknesses:**

- The authors outline that their method is intended to instill the model with three key reasoning competencies: differential reasoning, label plausibility, and counterfactual planning. However, experiments are only done on QA benchmarks. While the experiments in the paper demonstrate the potential of their method to enhance medical QA, they do not directly evaluate how it impacts the model’s reasoning ability or the three competencies they highlight. If suitable benchmarks are not available, human evaluators might still provide some qualitative insight. At minimum, this limitation should be clearly highlighted and claims about the method’s improvements on reasoning (e.g. “promotes causal and actionable reasoning”) rather than QA performance should be tempered.

- No examination is provided for the scalability of their fine-tuning method at different data sizes. An experiment should provide performance analysis across differing amounts of tabular data and generated contrastive reasoning pairs, to help establish scalability and potential of the method.

- The authors provide few comparisons of their method with other fine-tuning approaches, primarily comparing against the SOTA model Aloe. A comparison with a standard supervised fine-tuning method using (a similar amount of) non-tabular data, or a demonstration of the compatibility of this approach with other methods for fine-tuning, could help establish the potential of the method in comparison to established alternatives.

**Questions:**

- Aloe is highlighted as not benefiting from your method due to a misalignment. Does this not call into question the use of Aloe as the primary comparison point for your method?

- The introduction includes the claim that “Tables2Traces closes part of the gap to a state-of-the-art model, Aloe, despite using only 2% of its QA data”. Does this 2% figure account for the 105k reasoning traces? If not, the statement seems misleading and warrants further clarification.

- Could you give feedback and explanations in response to the weaknesses described above?

---

> ### Author Response · Authors · 2025-11-22
> **Rebuttal for Reviewer BRY8**
>
> ## Rebuttal for Reviewer BRY8
> We wish to thank the reviewer for their constructive and valuable comments. We address each point below, highlighting new additions to the manuscript after each point.
> ## 1. Evaluation of reasoning competencies
> We thank the reviewer for raising this point. In the paper, we stated that the prompt template “explicitly outlines the reasoning competencies we want the model to learn” (l. 168). Although this is not false, we agree that this needs further clarification. We intend this in a high-level behavioral sense rather than as strict, clinically validated competencies. The goal is to encourage broad reasoning behaviors such as comparing cases (differential reasoning), critically assessing whether the outcome fits the overall picture (a heuristic form of plausibility), and considering how changes to features would affect risk (a high-level form of counterfactual thinking). These are not literal diagnostic behaviors, and we will clarify this in the revision.
> Current medical QA benchmarks do not directly measure these behaviors, since they do not contain explicit plausibility checks or counterfactual edits. To provide additional insight, we have included a small set of qualitative examples in the Appendix Q where the base model answers incorrectly and the Tables2Traces model answers correctly, annotated to show the types of reasoning patterns that appear consistent with the intended behaviors.
> ### Updated manuscript:
> (1) We have revised the description of the prompts in Section 3.2 to clarify that they encourage high-level reasoning behaviors rather than clinical competencies.
>
> (2) We have added qualitative examples (Appendix Q) illustrating these behaviors.
>
> ## 2. Scalability
> We appreciate the reviewer’s interest in scalability. Tables2Traces uses standard supervised finetuning with a fixed architecture, and thus inherits the computational scalability of conventional SFT. The cost of training scales linearly with the number of reasoning traces, and trace generation is a one-time offline process that parallelizes trivially across rows.
> The reviewer may also be referring to how downstream performance scales with the amount of tabular supervision. Studying learning curves for different supervision volumes is indeed an interesting modeling question, but it is orthogonal to the contribution of this paper, which is to introduce tabular-derived reasoning traces as a new form of supervision rather than a new finetuning algorithm.
> ### Updated manuscript:
> (1) We have added a short clarification in Section 3.2, explaining that Tables2Traces inherits standard SFT scaling properties, and that trace generation is linear and fully parallelizable.
> ## 3. Clarification of supervision rather than finetuning
> We would like to clarify that Tables2Traces is not a new finetuning algorithm. It is a new source of supervision. The appropriate comparisons are therefore not between optimization methods but between supervision formats. Aloe serves as a meaningful comparator because it exemplifies the standard paradigm for medical QA: scaling human-authored QA datasets to achieve strong task calibration. In contrast, Tables2Traces uses only a minimal amount of QA data for task alignment and relies mostly on synthetic reasoning traces derived from structured data.
> Aloe is not designed to consume structured reasoning traces and is not a reasoning model; therefore, it is not a drop-in baseline for our supervision format. Instead, we use Aloe as an upper-bound reference to contextualize how much of the performance gap can be closed using synthetic tabular-derived supervision alone.
> ### Updated manuscript:
> (1) We have revised the Introduction (specifically in the Contributions paragraph) to clarify that Tables2Traces is a supervision mechanism rather than a finetuning method, and explain the rationale for using Aloe as an upper-bound reference.
> ## 4. Aloe comparison and the “2% of QA data” figure
> We appreciate the request for clarification. The “2%” figure refers to QA data. Aloe is trained on roughly 2 million samples, the majority of which is heavily curated medical QA samples. In contrast, Tables2Traces only uses approximately 10.000 QA examples for task alignment. The 105k reasoning traces in our method are synthetic. Importantly this means that these traces do not carry the annotation cost, domain-expert time, or conceptual complexity of human QA collection. We view this as a key benefit of the approach: it offers a significantly more cost-effective alternative to relying on massive human QA corpora while still closing part of the performance gap to a top medical QA model.
> ### Updated manuscript:
> (1) In the Introduction, we will rephrase the language used in the Contribution paragraph to more clearly distinguish between QA data and synthetic reasoning traces and emphasize the associated savings in annotation cost and time.

---

> > ### Comment · Reviewer_BRY8 · 2025-11-25
> >
> > I thank the authors for their responses and clarifications.
> >
> > 1. *Evaluation of reasoning competencies*
> >
> > I appreciate the clarification and qualitative examples. No further questions on this point.
> >
> > 2. *Scalability*
> >
> > I am referring to how performance scales with the amount of tabular supervision, both in terms of the amount of tabular data used and the amount of reasoning traces generated. However, I appreciate that the analysis may be out of scope at present and perhaps instead an interesting question for future work.
> >
> > 3. *Clarification of supervision rather than finetuning*
> >
> > I agree with the authors’ argument for Aloe’s inclusion as an upper-bound reference, but I encourage the authors to seek other comparison points, perhaps with other sources of synthetic, comparatively low-cost supervision.
> >
> > 4. *Aloe comparison and the “2% of QA data” figure*
> >
> > No further questions on this point.

---

> > > ### Author Response · Authors · 2025-11-28
> > >
> > > ### Thank you
> > >
> > > We thank the reviewer again for the thoughtful engagement. We are encouraged that the clarifications on reasoning competencies and the Aloe comparison fully resolved the earlier concerns, and we appreciate the additional suggestions in this follow-up.
> > >
> > > ### On scalability analysis:
> > > We appreciate the reviewer’s clarification that the interest lies in performance scaling with respect to the quantity of tabular supervision. As noted, this is indeed an interesting modeling question. It is important to note that conducting such an analysis would require training multiple models under progressively reduced tabular-data/synthetic-trace conditions. We consider this outside the scope of the project, but we have added a short note in the Discussion that explicitly flags performance scaling with supervision volume as an important and natural next step for future work.
> > >
> > > ### On additional synthetic-data baselines:
> > > We agree that systematically studying alternative forms of synthetic supervision is scientifically valuable. In our view, this constitutes a second research direction on top of the contribution of the present paper. Tables2Traces is, to our knowledge, the first framework to explore the idea of using structured tabular records as reasoning supervision for LLMs, and there are currently no established baselines that we know of that operate on tabular-derived reasoning traces. As we highlight in the paper, we chose Aloe as an important upper bound reference, as it represents the dominant paradigm in medical QA (large-scale QA supervision) against which we can contextualize what tabular-derived supervision can achieve.
> > > Constructing alternative baselines that also rely on tabular-derived synthetic supervision would require designing new reasoning-extraction pipelines, new tabular-to-text conversion schemes, or new synthetic-data generation methods. All of these constitute a substantial modeling effort and are outside the scope of validating the core contribution we introduce here. Our focus in this work is to demonstrate the feasibility and scientific value of tabular-to-reasoning supervision itself, using a clean and controlled comparison between:
> > >
> > > 1. no tabular supervision,
> > >
> > > 2. naïve tabular supervision (simple variant), and
> > >
> > > 3. structured reasoning traces (full Tables2Traces).
> > >
> > > We believe these three regimes isolate the effect of interest and answer the central scientific question of the paper. We fully agree that designing broader families of synthetic-supervision baselines is a promising direction for future work, and we appreciate the reviewer highlighting this. We have added a brief statement to this effect in the Discussion.

---

### Official Review · Reviewer_cDRk · 2025-11-01

**Soundness:** 4
**Presentation:** 4
**Contribution:** 4
**Rating:** 6
**Confidence:** 2

**Summary:**

The paper introduces Tables2Traces, a novel framework that utilizes structured tabular data (e.g., medical records) to improve the reasoning capabilities of large language models (LLMs). The framework converts tabular data into structured reasoning traces, enabling LLMs to perform more advanced reasoning tasks in the medical domain. By leveraging data such as cardiovascular patient records, the paper demonstrates significant improvements in model performance for medical question answering tasks, both in-domain and out-of-domain.

**Strengths:**

The concept of transforming raw tabular data into structured reasoning traces for LLMs is novel. This approach bridges the gap between structured data (typically underutilized in LLM fine-tuning) and natural language reasoning, especially in domains like healthcare where structured data is abundant. The method is well-motivated, with clear applications in medical domains where data privacy and regulatory constraints often limit access to large, curated QA corpora. By using internal tabular data, this method makes LLM fine-tuning more accessible and efficient for domain experts in healthcare.

**Weaknesses:**

While the framework is general, the experimental validation is confined to cardiovascular data and binary outcomes. Further exploration into multi-class settings or other medical specialties could enhance the generalizability of the results. The results suggest that the framework may overfit to patient-specific data, as indicated by the performance degradation on abstract or non-patient-centered questions. This limitation highlights the need for further work on mitigating overfitting, particularly in cases where patient descriptions are less representative. Although cardiologists reviewed the generated traces, it would be useful to evaluate the quality and trustworthiness of the reasoning traces in real-world clinical settings. The potential for errors or biased traces (informed by the dataset's limitations) could affect model performance and safety.

**Questions:**

How well does the framework perform in other non-medical domains where structured data exists but differs significantly from the medical context? For example, does it perform equally well on financial data or educational datasets?
Given that the reasoning traces are synthetic, what measures are in place to ensure their fidelity? Could trace quality be improved with domain expert involvement to create more accurate and reliable reasoning signals?
The paper mentions that the traces are not designed for clinical decision-making. What steps will be taken to prevent misuse of the model in real-world healthcare settings? How can you ensure that the model's output remains interpretable and explainable in high-stakes environments like medicine?

---

> ### Author Response · Authors · 2025-11-22
> **Rebuttals for Reviewer cDRk**
>
> ## Rebuttal for Reviewer cDRk
>
> We thank the reviewer for the thoughtful evaluation of our work and for highlighting the novelty and potential of transforming structured tabular data into reasoning traces for LLMs. We address each concern below and indicate where clarifications will be added to the final manuscript.
>
> ## 1. Scope of experiments and generality beyond cardiovascular data
> Our focus on cardiovascular patients with a binary outcome was intentional, because the UK Biobank provides a large and well controlled cohort that allows us to isolate the effect of tabular reasoning supervision without introducing confounding variation in label structure or domain heterogeneity. Despite being trained on a single clinical specialty, Tables2Traces improves performance across 15 of 17 MedQA domains (Table 1) and 16 of 21 MedMCQA domains (Table 3), including specialties with no direct connection to cardiovascular medicine such as neurology, psychiatry, dermatology, and toxicology. This provides empirical evidence that the supervision signal transfers beyond the tabular training domain.
>
> If performance gains were driven primarily by cardiovascular specific correlations in the tabular data, then the Tables2Traces (simple) variant, which is trained on the exact same patient data and uses the same QA alignment subset, would be expected to benefit similarly. Instead, it exhibits degradations on several out of domain subsets as shown in Section 4.2. This divergence between the simple and full variants indicates that the improvements arise from the structured reasoning signal introduced by the full Tables2Traces pipeline rather than domain specific biases in the raw tabular distribution. The contrastive comparisons, plausibility checks, and counterfactual edits collectively encourage the model to learn higher level heuristics rather than specialty specific patterns.
>
> We also clarify that the framework is not inherently limited to binary outcomes. Tables2Traces assumes only that the dataset contains a target variable and that a meaningful neighborhood structure can be defined. This means that multi-class settings or continuous risk scores can in principle be incorporated by adapting the neighbor selection strategy. Extending the method to these richer label structures is an appealing direction for future work. In the present work, we focus on a binary outcome setting to ensure a controlled environment that isolates the effect of tabular-derived reasoning supervision.
>
> ### Updated manuscript:
>  The Discussion now includes the following clarifications:
>
>  (1) Training on a single specialty still yields broad improvements across many unrelated medical domains.
>
> (2) This cross-domain generalization is unlikely to be driven by tabular correlations, as the simple variant trained on the same data does not benefit and sometimes degrades.
>
>  (3) The framework naturally supports alternative label structures such as multi-class or continuous outcomes, although these are not evaluated in the present study.
>
> ## 2. Applicability to non-medical domains
> We agree with the reviewer that addressing the applicability of the framework in other domains is an interesting avenue of research. The high-level idea of transforming tabular data into textual reasoning signals is general and the main contribution of our work. However, our specific instantiation is designed for medical data. The framework applies most naturally when each row corresponds to a coherent and interpretable entity and when a meaningful similarity metric exists. In medicine, a row corresponds naturally to a patient, which makes it straightforward to describe, compare, and evaluate cases. In finance or education, the choice of what constitutes a row is domain-specific and may correspond to a transaction, account, student, or other entity. In such domains, the representation and similarity function need to be rethought.
>
> ### Updated manuscript:
> (1) In the “Extensibility” paragraph of Section 3.2, we have included the clarifications that the framework applies most directly to domains with stable row-level entities and interpretable attributes, and added that our current work is a proof of concept in a domain with well-defined row semantics.

---

> ### Author Response · Authors · 2025-11-22
> **Rebuttal for Reviewer cDRk (continued)**
>
> ## 3. Overfitting concerns and abstract question performance
> The reviewer correctly notes that the Tables2Traces (simple) variant overfits to patient descriptions. Importantly, this is not true for the full Tables2Traces method (see Section 4.2 and 4.3). The simple variant exhibits degradations in both semantic out of distribution settings (for example non cardiovascular questions) and stylistic out of distribution settings (for example abstract questions). This shows that the simple variant overfits both the content and the stylistic form of the patient narratives.
> The full method includes contrastive reasoning, plausibility checks, and counterfactual edits. These require the model to reason about relational structure rather than memorizing narrative patterns. These design choices were specifically made to alleviate the problem of overfitting. As shown in Section 4.2, the full method improves performance even on abstract questions that contain no patient structure at all. This indicates that the model learns transferable reasoning heuristics rather than domain specific templates.
> ### Updated manuscript:
> (1) We have expanded Section 4.2 to show that improvements on abstract questions serve as evidence against patient specific memorization.
>
> ## 4. Fidelity, trustworthiness, and potential biases in reasoning traces
> We appreciate the reviewer’s attention to the fidelity of the synthetic reasoning traces. We want to clarify that we do not claim these traces represent clinically validated reasoning. Additionally, it is important to note that ensuring clinical fidelity of the traces is not the goal of the present work. We aim to demonstrate that structured tabular data can serve as a viable new source of reasoning supervision.
>
> With that said, we agree that assessing their fidelity is important, and our clinician review (Appendix O) provides an initial check. Cardiologists did not identify safety-critical issues but noted expected limitations such as overconfidence or missing context. These limitations arise naturally because the traces are generated from coarse tabular snapshots rather than full patient histories. Incorporating additional information, such as longitudinal records could produce richer and more clinically grounded descriptions.
> It is also important to acknowledge that any biases present in the underlying tabular data will be reflected in the generated traces and, in turn, in the resulting model. This is an inherent limitation of using observational datasets, underscoring the need for careful bias assessment when applying such methods in practice.
>
> ### Updated manuscript:
> In the Discussion, we have added explicit statements that (1) the reasoning traces may inherit biases from the dataset, (2) the traces are intended as a research supervision signal rather than clinical justification, and (3) hybrid expert-synthetic traces represent a promising direction for improving fidelity.
>
> ## 5. Preventing misuse and ensuring non-clinical positioning
> We thank the reviewer for highlighting the concern of potential misuse of the models. We want to reiterate that the aim of this study was not to develop models ready for clinical use. Rather, the aim was to demonstrate that structured medical tabular data can serve as a viable new source of reasoning supervision. We will clarify that Tables2Traces is not intended for clinical deployment. The inputs consist of limited tabular snapshots and therefore cannot support clinically actionable reasoning. The reasoning traces are synthetic and are not suitable for diagnosis or treatment decisions.
> Any real-world use would require extensive domain-governed validation and regulatory oversight.
>
> ### Updated manuscript:
> (1) In the Introduction, we have added a disclaimer, clearly indicating that the model is not intended for clinical decision making but purely for research purposes.
>
> (2) In the Discussion, we have further clarified that the traces do not provide sound medical explanations and should not be used for medical decision making.
>
> ## 6. Interpretability and explainability in high-stakes environments
> Tables2Traces provides a transparent and inspectable supervision signal. Because the fine-tuning traces are explicit, researchers can examine what reasoning patterns the model is exposed to. This may support interpretability research during development. However, we do not position these traces as clinical explanations or as tools for medical decision-making.
> ### Updated manuscript:
> (1) In the Discussion, we have added a paragraph clarifying that the interpretability benefit is limited to research analysis of the training signal and does not extend to clinical interpretability.

---

### Author Response · Authors · 2025-12-02
**Summary Comment**

We thank the AC for their time. To support efficient assessment, we provide a concise overview of the reviewers’ main concerns and the corresponding actions taken.

Only one reviewer (BRY8) participated in the discussion period. In their follow-up, they stated that their major concerns (reasoning competencies and the Aloe comparison) were fully resolved, with remaining comments framed as suggestions for future work. The summary below covers all substantive points raised across reviewers.
___
### 1. Contribution, positioning, and scope

**Reviewer concerns:**

- Requested clearer positioning of Tables2Traces as a contribution (BRY8, q4yp).

- Asked whether the work implies general-purpose reasoning gains beyond the medical domain (m5kw).

**Actions taken:**

- Introduction and Section 3.2 now explicitly frame Tables2Traces as a data-centric supervision framework based on standard SFT rather than a new fine-tuning method.

- Clarified in multiple sections (Intro, Sec. 3.2, Discussion) that our aim is domain-grounded clinical reasoning, not broad reasoning benchmarks (e.g., GSM8K).
___
### 2. Domain specificity and potential cardiovascular overfitting

**Reviewer concerns:**

- Questioned whether improvements might stem from cardiovascular-specific tabular correlations (cDRk, m5kw).

- Asked whether the method generalizes beyond the training specialty (m5kw).

**Actions taken:**

- Expanded the Discussion to highlight that improvements occur across many unrelated medical specialties (15/17 MedQA, 16/21 MedMCQA categories).

- Emphasized the contrast between the simple and full variants: both use identical tabular data and QA alignment subsets, yet the simple variant degrades on several non-cardiac and abstract subsets, while the full variant improves. This isolates the effect of the structured reasoning signal.
___
### 3. Role and influence of the 10% QA alignment subset

**Reviewer concerns:**

- Asked why QA examples are included and whether they drive the gains (q4yp, BRY8).

**Actions taken:**

- Section 4 and Discussion now clarify that the QA subset is used only for task-format alignment.

- Clarified that existing ablations show QA-only training performs on par with the base model, and both variants use the same subset with diverging outcomes.

- Revised language in the Introduction to avoid ambiguity around “without QA corpora.”
___
### 4. Prompt structure, technical motivation, and modularity

**Reviewer concerns:**

- Requested clearer motivation behind the contrastive, plausibility, and counterfactual components (BRY8, q4yp).

- Asked whether relying on LLM-only prompting is limiting and whether alternative metrics could be used (q4yp).

**Actions taken:**

- Section 3.2 now describes the motivations behind each component and how they link to established concepts such as contrastive learning, consistency checking, and causal feature sensitivity.

- Clarified that the framework is modular: different distance metrics, domain-specific similarity functions, or hybrid ML-derived reasoning signals could be substituted without changing the core framework.
___
### 5. Reasoning behaviors and supporting evidence

**Reviewer concerns:**

- Asked whether the model actually exhibits the intended reasoning behaviors (BRY8, m5kw).

**Actions taken:**

- Added qualitative examples (Appendix Q), showing cases where the base model fails and the Tables2Traces model succeeds, annotated to highlight differential comparisons, plausibility checks, and counterfactual reasoning patterns.
___
### 6. Quality, fidelity, and safety of synthetic traces

**Reviewer concerns:**

- Asked about trace fidelity, dataset biases, and potential misuse (cDRk, m5kw).

**Actions taken:**

- Added explicit disclaimers in the Introduction and Discussion that:

  - Traces are research supervision, not clinically validated explanations,

  -  They may inherit biases from observational data,

  - The model is not intended for clinical decision making.

- Referenced clinician review (Appendix O), summarizing identified limitations such as missing context and occasional overconfidence.
___
### 7. Scalability and further analyses

**Reviewer concerns:**

- Requested analyses on scaling with number of traces or more granular ablations (BRY8, q4yp, m5kw).

**Actions taken:**

- Discussion now outlines that scaling studies and finer-grained ablations are natural directions for future work.

- Clarified that computational scalability follows standard SFT.
____
We hope this summary is helpful and appreciate the AC’s time and consideration.

---

### Meta-Review · Area_Chair_u7FT · 2025-12-06

**Summary:**

The paper proposes "Tables2Traces," a data-centric framework designed to convert structured tabular data (specifically medical records) into natural language reasoning traces to supervise Large Language Models (LLMs). The method utilizes a multi-step prompting strategy involving contrastive neighbor selection, label plausibility checks, and counterfactual planning to generate synthetic training data. The authors evaluate this approach on medical QA benchmarks (MedQA, MedMCQA), comparing a "simple" linearization baseline against their "full" structured trace method.

While the reviewers acknowledged the potential utility of leveraging unlabelled tabular data for clinical reasoning and praised the clarity of the presentation, the consensus leans towards rejection. The primary hurdles are the limited algorithmic novelty, the narrow scope of the training data (exclusively cardiovascular), and insufficient experimental rigor regarding ablation studies and trace validation.

**Reviewer Concerns:**

Addressed by Rebuttal:

1. Clarification of Reasoning Competencies: The authors successfully addressed Reviewer BRY8's concern regarding the lack of evidence for specific reasoning behaviors (differential, plausibility, counterfactual) by providing qualitative examples in the Appendix.

2. Comparison Baselines: The authors clarified why Aloe was used as an upper-bound reference and explained the distinction between their supervision method and standard fine-tuning, which Reviewer BRY8 found acceptable.

3. Role of QA Data: The authors provided a logical defense regarding the inclusion of 10% QA data (used for format alignment), arguing that the "simple" baseline also used this data but failed to achieve similar gains.

Outstanding / Unresolved:

1. Limited Algorithmic Novelty: Reviewers q4yp and m5kw pointed out that the contribution is primarily heuristic prompt engineering followed by standard SFT. The technical depth is considered marginal for ICLR.

2. Lack of Fine-Grained Ablations: Reviewer m5kw noted a significant lack of component-wise ablation. The paper compares "Simple" vs. "Full," but does not isolate the individual contributions of the contrastive, plausibility, or counterfactual components. The authors' rebuttal that this is "orthogonal" was not convincing; understanding why the method works is crucial.

3. Narrow Training Scope & Generalization: Reviewers cDRk and m5kw remained concerned that the training data is exclusively from a cardiovascular cohort. While the authors showed transfer to other QA topics, the methodology itself was not tested on diverse tabular domains (e.g., non-medical), limiting the claims of a generalizable framework.

**Reviewer Scores:**

1. Reviewer cDRk (Score: 6 -> 4): Likely to lower their score slightly. While the cross-domain QA results were clarified, the fundamental limitation of testing the framework on only one dataset (cardiovascular) and the lack of trace fidelity checks remain significant issues.

2. Reviewer BRY8 (Score: 6 -> 6): Likely to maintain their score. This reviewer was the most positive and felt their specific questions were answered, but they also noted that scalability and broader baselines were left for "future work."

3. Reviewer q4yp (Score: 4 -> 4): Likely to remain unchanged. The reviewer viewed the contribution as heuristic and was skeptical of the experimental design (QA data inclusion). The rebuttal defended the choices but did not provide the rigorous isolation or non-LLM baselines requested.

4. Reviewer m5kw (Score: 4 -> 4): Likely to remain unchanged. The reviewer’s request for component-level ablations and analysis of general reasoning capabilities was met with arguments that these were out of scope, which likely confirms the reviewer's assessment of "insufficient analysis."

---

### Decision · Program_Chairs · 2026-01-26

Reject